# ON THE EXPRESSIVE POWER OF GEOMETRIC GRAPH NEURAL NETWORKS

## ABSTRACT

The expressive power of Graph Neural Networks (GNNs) has been studied extensively through the lens of the Weisfeiler-Leman (WL) graph isomorphism test. Yet, many graphs in scientific and engineering applications come embedded in Euclidean space with an additional notion of geometric isomorphism, which is not covered by the WL framework. In this work, we propose a geometric version of the WL test (GWL) for discriminating geometric graphs while respecting the underlying physical symmetries: permutations, rotation, reflection, and translation. We use GWL to characterise the expressive power of GNNs that are invariant or equivariant to physical symmetries in terms of the classes of geometric graphs they can distinguish. This allows us to formalise the advantages of equivariant GNNs over invariant GNNs: equivariant layers have greater expressive power as they enable propagating geometric information beyond local neighbourhoods, while invariant layers cannot distinguish graphs that are locally similar, highlighting their inability to compute global geometric quantities. Finally, we prove the equivalence between the universal approximation properties of geometric GNNs and our more granular discrimination-based perspective.

## 1 INTRODUCTION

Systems in biochemistry (Jamasb et al., 2022), material science (Chanussot et al., 2021), physical simulations (Sanchez-Gonzalez et al., 2020), and multiagent robotics (Li et al., 2020) contain both geometry and relational structure. Such systems can be modelled via *geometric graphs* embedded in Euclidean space. For example, molecules are represented as a set of nodes which contain information about each atom and its 3D spatial coordinates as well as other geometric quantities such as velocity or acceleration. Notably, the geometric attributes transform along with Euclidean transformations of the system, *i.e.* they are equivariant to symmetry groups of rotations, reflections, and translation. Standard Graph Neural Networks (GNNs) which do not take spatial symmetries into account are ill-suited for geometric graphs, as the geometric attributes would no longer retain their physical meaning and transformation behaviour (Bogatskiy et al., 2022; Bronstein et al., 2021).

GNNs specialised for geometric graphs follow the message passing paradigm (Gilmer et al., 2017) where node features are updated in a permutation equivariant manner by aggregating features from local neighbourhoods. Crucially, in addition to permutations, the geometric attributes of the nodes transform along with Euclidean transformations of the system, *i.e.* they are equivariant to the Lie group of rotations ($SO(d)$) or rotations and reflections ($O(d)$). We use $\mathfrak{G}$ as a generic symbol for these Lie groups. We consider two classes of GNNs for geometric graphs: (1) $\mathfrak{G}$-**equivariant models**, where the intermediate features and propagated messages are equivariant geometric quantities such as vectors or tensors (Thomas et al., 2018; Anderson et al., 2019; Jing et al., 2020; Satorras et al., 2021; Brandstetter et al., 2022); and (2) $\mathfrak{G}$-**invariant models**, which only propagate local invariant scalar features such as distances and angles (Schütt et al., 2018; Xie & Grossman, 2018; Gasteiger et al., 2020). Despite promising empirical results for both classes of architectures, key theoretical questions remain unanswered: (1) How to characterise the *expressive power* of geometric GNNs? And (2) what is the tradeoff between $\mathfrak{G}$-equivariant and $\mathfrak{G}$-invariant GNNs?

The graph isomorphism problem (Read & Corneil, 1977) and the Weisfeiler-Leman (WL) (Weisfeiler & Leman, 1968) test for distinguishing non-isomorphic graphs have become a powerful tool for analysing the expressive power of non-geometric GNNs (Xu et al., 2019; Morris et al., 2019).

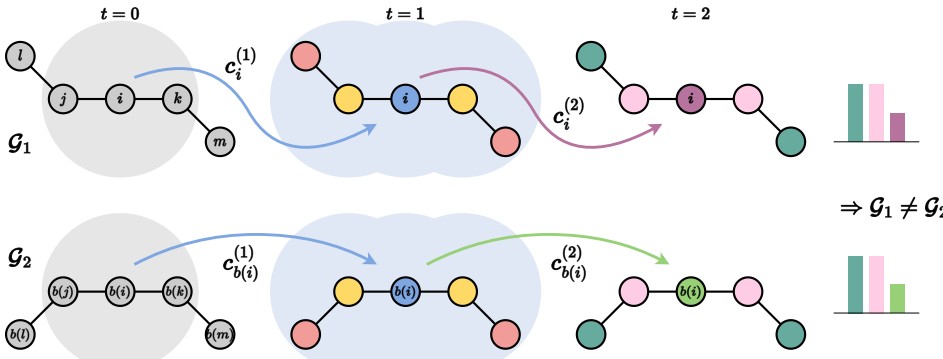

Figure 1: **Geometric Weisfeiler-Leman Test.** GWL distinguishes non-isomorphic geometric graphs $\mathcal{G}_1$ and $\mathcal{G}_2$ by injectively assigning colours to distinct neighbourhood patterns, up to global symmetries (here $\mathfrak{G} = O(d)$). Each iteration expands the neighbourhood from which geometric information can be gathered (shaded for node $i$). Example inspired by Schütt et al. (2021).

The WL framework has been a major driver of progress in graph representation learning (Chen et al., 2019; Maron et al., 2019; Dwivedi et al., 2020; Bodnar et al., 2021b;a). However, the WL framework does not directly apply to geometric graphs as they exhibit a stronger notion of isomorphism that also takes spatial symmetries into account.

**Contributions.** In this work, we study the expressive power of geometric GNNs from the perspective of discriminating non-isomorphic geometric graphs:

- In Section 3, we propose a geometric version of the Weisfeiler-Leman graph isomorphism test, termed GWL. We use GWL to formally characterise classes of graphs that can and cannot be distinguished by $\mathfrak{G}$-invariant and $\mathfrak{G}$-equivariant GNNs. We show how invariant models have limited expressive power as they cannot distinguish graphs where one-hop local neighbourhoods are similar, while equivariant models distinguish a larger class of graphs by propagating geometric vector quantities beyond local neighbourhoods.

- In Section 4, we study the design space of geometric GNNs using GWL, highlighting their theoretical limitations in terms of depth and body order, as well as discussing practical implications. We show that $\mathfrak{G}$-invariant models cannot compute global geometric properties such as volume, area, centroid, etc. Synthetic experiments in Appendix C supplement our theory and highlight practical challenges in building geometric GNNs. ⟶ Addresses R1.4, R2.5, R3.1, R4.1

- In Section 5, we follow Chen et al. (2019) and prove an equivalence between a model's ability to discriminate geometric graphs and its ability to universally approximate $\mathfrak{G}$-invariant functions. While universality is binary, GWL's discrimination-based perspective provides a more granular and practically insightful lens to study geoemtric GNNs.

## 2 BACKGROUND

**Graph Isomorphism and Weisfeiler-Leman.** An attributed graph $\mathcal{G} = (\boldsymbol{A}, \boldsymbol{S})$ with a node set $\mathcal{V}$ of size $n$ consists of an $n \times n$ adjacency matrix $\boldsymbol{A}$ and a matrix of scalar features $\boldsymbol{S} \in \mathbb{R}^{n \times f}$. Two attributed graphs $\mathcal{G}, \mathcal{H}$ are *isomorphic* if there exists an edge-preserving bijection $b : \mathcal{V}(\mathcal{G}) \to \mathcal{V}(\mathcal{H})$ such that $\boldsymbol{s}_i^{(\mathcal{G})} = \boldsymbol{s}_{b(i)}^{(\mathcal{H})}$, where the subscripts index rows and columns in the corresponding matrices. The *Weisfeiler-Leman test* (WL) is an algorithm for testing whether two (attributed) graphs are isomorphic (Weisfeiler & Leman, 1968). At iteration zero the algorithm assigns a *colour* $c_i^{(0)} \in C$ from a countable space of colours $C$ to each node $i$. Nodes are coloured the same if their features are the same, otherwise, they are coloured differently. In subsequent iterations $t$, WL iteratively updates the node colouring by producing a new $\boldsymbol{c}_i^{(t)} \in C$:

$$\boldsymbol{c}_i^{(t)} := \text{HASH}\left(\boldsymbol{c}_i^{(t-1)}, \{\!\{\boldsymbol{c}_j^{(t-1)} \mid j \in \mathcal{N}_i\}\!\}\right), \tag{1}$$

where HASH is an injective map (i.e. a perfect hash map) that assigns a unique colour to each input and $\{\!\{\cdot\}\!\}$ denotes a multiset – a set that allows for repeated elements. The test terminates when the partition of the nodes induced by the colours becomes stable. Given two graphs $\mathcal{G}$ and $\mathcal{H}$, if there exists some iteration $t$ for which $\{\!\{c_i^{(t)} \mid i \in \mathcal{V}(\mathcal{G})\}\!\} \neq \{\!\{c_i^{(t)} \mid i \in \mathcal{V}(\mathcal{H})\}\!\}$, then the graphs are not isomorphic. Otherwise, the WL test is inconclusive, and we say it cannot distinguish the two graphs.

**Group Theory.** We assume basic familiarity with group theory, see Zee (2016) for an overview. We denote the action of the group $\mathfrak{G}$ on a space $X$ by $\mathfrak{g} \cdot x$. If $\mathfrak{G}$ acts on spaces $X$ and $Y$, we say a function $f : X \to Y$ is $\mathfrak{G}$-*equivariant* if $f(\mathfrak{g} \cdot x) = \mathfrak{g} \cdot f(x)$. A function $f : X \to Y$ is $\mathfrak{G}$-*invariant* if $f(\mathfrak{g} \cdot x) = f(x)$. The $\mathfrak{G}$-*orbit* of $x \in X$ is $\mathcal{O}_\mathfrak{G}(x) = \{\mathfrak{g} \cdot x \mid \mathfrak{g} \in \mathfrak{G}\} \subseteq X$. When $x$ and $x'$ are part of the same orbit, we write $x \simeq x'$. We say a function $f : X \to Y$ is $\mathfrak{G}$-*orbit injective* if we have $f(x_1) = f(x_2)$ if and only if $x_1 \simeq x_2$ for any $x_1, x_2 \in X$. Necessarily, such a function is $\mathfrak{G}$-invariant, since $f(\mathfrak{g} \cdot x) = f(x)$. We work with the permutation group over $n$ elements $S_n$ and the Lie groups $\mathfrak{G} = SO(d)$ or $\mathfrak{G} = O(d)$. Invariance to the translation group $T(d)$ is conventionally handled using relative positions. Given one of the standard groups above, for an element $\mathfrak{g}$ we denote by $\boldsymbol{M}_\mathfrak{g}$ (or another capital letter) its standard matrix representation.

Addresses R2.1

**Geometric graphs.** A geometric graph $\mathcal{G} = (\boldsymbol{A}, \boldsymbol{S}, \vec{\boldsymbol{V}}, \vec{\boldsymbol{X}})$ with a node set $\mathcal{V}$ is an attributed graph that is also decorated with geometric attributes: node coordinates $\vec{\boldsymbol{X}} \in \mathbb{R}^{n \times d}$ and (optionally) vector features $\vec{\boldsymbol{V}} \in \mathbb{R}^{n \times d}$ (*e.g.* velocity, acceleration). The geometric attributes transform as follows under the action of the relevant groups: (1) $S_n$ acts on the graph via $\boldsymbol{P}_\sigma \mathcal{G} := (\boldsymbol{P}_\sigma \boldsymbol{A} \boldsymbol{P}_\sigma^\top, \boldsymbol{P}_\sigma \boldsymbol{S}, \boldsymbol{P}_\sigma \vec{\boldsymbol{V}}, \boldsymbol{P}_\sigma \vec{\boldsymbol{X}})$; (2) Orthogonal transformations $\boldsymbol{Q}_\mathfrak{g} \in \mathfrak{G}$ act on $\vec{\boldsymbol{V}}, \vec{\boldsymbol{X}}$ via $\vec{\boldsymbol{V}} \boldsymbol{Q}_\mathfrak{g}, \vec{\boldsymbol{X}} \boldsymbol{Q}_\mathfrak{g}$; and (3) Translations $\vec{\boldsymbol{t}} \in T(d)$ act on the coordinates $\vec{\boldsymbol{X}}$ via $\vec{\boldsymbol{x}}_i + \vec{\boldsymbol{t}}$ for all nodes $i$. Without loss of generality, we work with a single vector feature per node. Our results generalise to multiple vector features or higher-order tensors, in which case we would replace the matrix group representation $\boldsymbol{Q}_\mathfrak{g}$ with a more generic $\rho(\mathfrak{g})$. Two geometric graphs $\mathcal{G}$ and $\mathcal{H}$ are *geometrically isomorphic* (denoted $\mathcal{G} \simeq \mathcal{H}$) if there exists an attributed graph isomorphism $b$ such that the geometric attributes are equivalent, up to global group actions $\boldsymbol{Q}_\mathfrak{g} \in \mathfrak{G}$ and $\vec{\boldsymbol{t}} \in T(d)$:

Addresses R2.2

$$\left(\boldsymbol{s}_i^{(\mathcal{G})}, \vec{\boldsymbol{v}}_i^{(\mathcal{G})}, \vec{\boldsymbol{x}}_i^{(\mathcal{G})}\right) = \left(\boldsymbol{s}_{b(i)}^{(\mathcal{H})}, \boldsymbol{Q}_\mathfrak{g} \vec{\boldsymbol{v}}_{b(i)}^{(\mathcal{H})}, \boldsymbol{Q}_\mathfrak{g} (\vec{\boldsymbol{x}}_{b(i)}^{(\mathcal{H})} + \vec{\boldsymbol{t}})\right) \quad \text{for all } i \in \mathcal{V}(\mathcal{G}). \tag{2}$$

Geometric graph isomorphism and distinguishing (sub-)graph geometries has important practical implications for representation learning. For *e.g.*, in molecular systems, an ideal architecture should map distinct local structural environments around atoms to distinct embeddings in representation space (Bartók et al., 2013; Pozdnyakov et al., 2020).

**Geometric Graph Neural Networks.** We consider two broad classes of geometric GNN architectures. $\mathfrak{G}$-*equivariant GNN layers* update scalar and vector features from iteration $t$ to $t + 1$ via learnable aggregate and update functions, AGG and UPD, respectively:

$$\boldsymbol{s}_i^{(t+1)}, \vec{\boldsymbol{v}}_i^{(t+1)} := \text{UPD}\left(\left(\boldsymbol{s}_i^{(t)}, \vec{\boldsymbol{v}}_i^{(t)}\right), \text{AGG}\left(\{\!\{(\boldsymbol{s}_i^{(t)}, \boldsymbol{s}_j^{(t)}, \vec{\boldsymbol{v}}_i^{(t)}, \vec{\boldsymbol{v}}_j^{(t)}, \vec{\boldsymbol{x}}_{ij}) \mid j \in \mathcal{N}_i\}\!\}\right)\right). \tag{3}$$

where $\vec{\boldsymbol{x}}_{ij} = \vec{\boldsymbol{x}}_i - \vec{\boldsymbol{x}}_j$ denote relative position vectors. Alternatively, $\mathfrak{G}$-*invariant GNN layers* do not update vector features and only aggregate scalar quantities from local neighbourhoods:

Addresses R4.2

$$\boldsymbol{s}_i^{(t+1)} := \text{UPD}\left(\boldsymbol{s}_i^{(t)}, \text{AGG}\left(\{\!\{(\boldsymbol{s}_i^{(t)}, \boldsymbol{s}_j^{(t)}, \vec{\boldsymbol{v}}_i, \vec{\boldsymbol{v}}_j, \vec{\boldsymbol{x}}_{ij}) \mid j \in \mathcal{N}_i\}\!\}\right)\right). \tag{4}$$

For both models, the scalar features at the final iteration are mapped to graph-level predictions via a permutation-invariant readout $f : \mathbb{R}^{n \times f} \to \mathbb{R}^{f'}$. See Appendix B for concrete examples of geometric GNNs covered by our framework.

Addresses R2.2, R4.3

## 3 THE GEOMETRIC WEISFEILER-LEMAN TEST

**Assumptions.** Analogous to the WL test, the geometric and scalar features the nodes are equipped with come from countable subsets $C \subset \mathbb{R}^d$ and $C' \subset \mathbb{R}$, respectively. As a result, when we require functions to be injective, we require them to be injective over the countable set of $\mathfrak{G}$-orbits that are obtained by acting with $\mathfrak{G}$ on the dataset.

**Intuition.** For an intuition of how to generalise WL to geometric graphs, we note that WL uses a local, node-centric, procedure to update the colour of each node $i$ using the colours of its the 1-hop *neighbourhood* $\mathcal{N}_i$. In the geometric setting, $\mathcal{N}_i$ is an attributed point cloud around the central

node $i$. As a result, each neighbourhood carries two types of information: (1) neighbourhood type (invariant to $\mathfrak{G}$) and (2) neighbourhood geometric orientation (equivariant to $\mathfrak{G}$). From an axiomatic point of view, our generalisation of the WL aggregation procedure must meet two properties:

**Property 1: Orbit injectivity of colours.** If two neighbourhoods are the same up to an action of $\mathfrak{G}$ (*e.g.* rotation), then the colours of the corresponding central nodes should be the same. Thus, the colouring must be $\mathfrak{G}$-orbit injective – which also makes it $\mathfrak{G}$-invariant – over the countable set of all orbits of neighbourhoods in our dataset.

**Property 2: Preservation of local geometry.** A key property of WL is that the aggregation is injective. A $\mathfrak{G}$-invariant colouring procedure that purely satisfies Property 1 is not sufficient because, by definition, it loses spatial properties of each neighbourhood such as the relative pose or orientation (Hinton et al., 2011). Thus, we must additionally update auxiliary *geometric information* variables in a way that is $\mathfrak{G}$-equivariant and injective.

**Geometric Weisfeiler-Leman (GWL).** These intuitions motivate the following definition of the GWL test. At initialisation, we assign to each node $i \in \mathcal{V}$ a scalar node colour $c_i \in C'$ and an auxiliary object $\boldsymbol{g}_i$ containing the geometric information associated to it:

$$c_i^{(0)} := \text{HASH}(\boldsymbol{s}_i), \quad \boldsymbol{g}_i^{(0)} := \left( c_i^{(0)}, \vec{\boldsymbol{v}}_i \right), \tag{5}$$

where HASH denotes an injective map over the scalar attributes $\boldsymbol{s}_i$ of node $i$. To define the inductive step, assume we have the colours of the nodes and the associated geometric objects at iteration $t-1$. Then, we can aggregate the geometric information around node $i$ into a new object as follows:

$$\boldsymbol{g}_i^{(t)} := \left( (c_i^{(t-1)}, \boldsymbol{g}_i^{(t-1)}), \; \{\!\{ (c_j^{(t-1)}, \boldsymbol{g}_j^{(t-1)}, \vec{\boldsymbol{x}}_{ij}) \mid j \in \mathcal{N}_i \}\!\} \right), \tag{6}$$

Importantly, the group $\mathfrak{G}$ can act on the geometric objects above inductively by acting on the geometric information inside it. This amounts to rotating (or reflecting) the entire $t$-hop neighbourhood contained inside:

$$\mathfrak{g} \cdot \boldsymbol{g}_i^{(0)} := \left( c_i^{(0)}, \; \boldsymbol{Q}_\mathfrak{g} \vec{\boldsymbol{v}}_i \right), \quad \mathfrak{g} \cdot \boldsymbol{g}_i^{(t)} := \left( (c_i^{(t-1)}, \mathfrak{g} \cdot \boldsymbol{g}_i^{(t-1)}), \; \{\!\{ (c_j^{(t-1)}, \mathfrak{g} \cdot \boldsymbol{g}_j^{(t-1)}, \boldsymbol{Q}_\mathfrak{g} \vec{\boldsymbol{x}}_{ij}) \mid j \in \mathcal{N}_i \}\!\} \right)$$

Clearly, the aggregation building $\boldsymbol{g}_i$ for any time-step $t$ is injective and $\mathfrak{G}$-equivariant. Finally, we can compute the node colours at iteration $t$ for all $i \in \mathcal{V}$ by aggregating the geometric information in the neighbourhood around node $i$:

$$c_i^{(t)} := \text{I-HASH}^{(t)} \left( \boldsymbol{g}_i^{(t)} \right), \tag{7}$$

by using a $\mathfrak{G}$-orbit injective and $\mathfrak{G}$-invariant function that we denote by I-HASH. That is for any geometric objects $\boldsymbol{g}, \boldsymbol{g}'$, I-HASH$(\boldsymbol{g}) = $ I-HASH$(\boldsymbol{g}')$ if and only if there exists $\mathfrak{g} \in \mathfrak{G}$ such that $\boldsymbol{g} = \mathfrak{g} \cdot \boldsymbol{g}'$. Note that I-HASH is an idealised $\mathfrak{G}$-orbit injective function, similar to the HASH function used in WL, which is not necessarily continuous.

Addresses
R4.4

**Overview.** With each iteration, $\boldsymbol{g}_i^{(t)}$ aggregates geometric information in progressively larger $t$-hop subgraph neighbourhoods $\mathcal{N}_i^{(t)}$ around the node $i$. The node colours summarise the structure of these $t$-hops via the $\mathfrak{G}$-invariant aggregation performed by I-HASH. The procedure terminates when the partitions of the nodes induced by the colours do not change from the previous iteration. Finally, given two geometric graphs $\mathcal{G}$ and $\mathcal{H}$, if there exists some iteration $t$ for which $\{\!\{ c_i^{(t)} \mid i \in \mathcal{V}(\mathcal{G}) \}\!\} \neq \{\!\{ c_i^{(t)} \mid i \in \mathcal{V}(\mathcal{H}) \}\!\}$, then GWL deems the two graphs as being geometrically non-isomorphic. Otherwise, we say the test cannot distinguish the two graphs.

**Invariant GWL.** Since we are interested in understanding the role of $\mathfrak{G}$-equivariance, we also consider a more restrictive Invariant GWL (IGWL) that only updates node colours using the $\mathfrak{G}$-orbit injective I-HASH function and does not propagate geometric information:

$$c_i^{(t)} := \text{I-HASH} \left( (c_i^{(t-1)}, \vec{\boldsymbol{v}}_i), \; \{\!\{ (c_j^{(t-1)}, \vec{\boldsymbol{v}}_j, \vec{\boldsymbol{x}}_{ij}) \mid j \in \mathcal{N}_i \}\!\} \right). \tag{8}$$

**IGWL with $k$-body scalars.** In order to further analyse the construction of the node colouring function I-HASH, we consider IGWL$_{(k)}$ based on the maximum number of nodes involved in the computation of $\mathfrak{G}$-invariant scalars (also known as the 'body order' (Batatia et al., 2022a)):

$$c_i^{(t)} := \text{I-HASH}_{(k)} \left( (c_i^{(t-1)}, \vec{\boldsymbol{v}}_i), \; \{\!\{ (c_j^{(t-1)}, \vec{\boldsymbol{v}}_j, \vec{\boldsymbol{x}}_{ij}) \mid j \in \mathcal{N}_i \}\!\} \right), \tag{9}$$

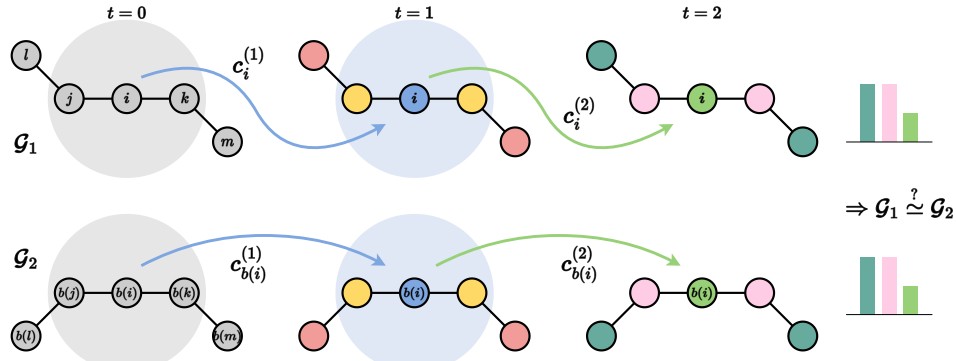

Figure 2: **Invariant GWL Test.** IGWL cannot distinguish $\mathcal{G}_1$ and $\mathcal{G}_2$ as they are 1-hop identical: The $\mathfrak{G}$-orbit of the 1-hop neighbourhood around each node is the same, and IGWL cannot propagate geometric orientation information beyond 1-hop (here $\mathfrak{G} = O(d)$).

and I-HASH$_{(k+1)}$ is defined as:

$$\text{HASH}\left(\{\!\{\text{I-HASH}\left((c_i^{(t-1)}, \vec{v}_i), \{\!\{(c_{j_1}^{(t-1)}, \vec{v}_{j_1}, \vec{x}_{ij_1}), \ldots, (c_{j_k}^{(t-1)}, \vec{v}_{j_k}, \vec{x}_{ij_k})\}\!\}\right) \mid \boldsymbol{j} \in (\mathcal{N}_i)^k\}\!\}\right),$$

where $\boldsymbol{j} = [j_1, \ldots, j_k]$ are all possible $k$-tuples formed of elements of $\mathcal{N}_i$. Therefore, IGWL$_{(k)}$ is now constrained to extract information only from all the possible $k$-sized tuples of nodes (including the central node) in a neighbourhood. For instance, I-HASH$_{(2)}$ can identify neighbourhoods only up to pairwise distances among the central node and any of its neighbours (*i.e.* a 2-body scalar), while I-HASH$_{(3)}$ up to distances and angles formed by any two edges (*i.e.* a 3-body scalar). Notably, distances and angles alone are incomplete descriptors of local geometry (Bartók et al., 2013; Pozdnyakov et al., 2020). Therefore, I-HASH$_{(k)}$ with lower $k$ makes the colouring weaker.

Addresses R4.4

### 3.1 WHAT GEOMETRIC GRAPHS CAN GWL AND IGWL DISTINGUISH?

In order to formalise the expressive power of GWL and IGWL, let us consider what geometric graphs can and cannot be distinguished by the tests. As a simple first observation, we note that when all coordinates and vectors are set equal to zero GWL coincides with the standard WL. In this *edge case*, GWL has the same expressive power as WL.

Next, let us consider consider the simplified setting of two geometric graphs $\mathcal{G}_1 = (\boldsymbol{A}_1, \boldsymbol{S}_1, \vec{\boldsymbol{V}}_1, \vec{\boldsymbol{X}}_1)$ and $\mathcal{G}_2 = (\boldsymbol{A}_2, \boldsymbol{S}_2, \vec{\boldsymbol{V}}_2, \vec{\boldsymbol{X}}_2)$ such that the underlying attributed graphs $(\boldsymbol{A}_1, \boldsymbol{S}_1)$ and $(\boldsymbol{A}_2, \boldsymbol{S}_2)$ are isomorphic. This case frequently occurs in chemistry, where molecules occur in different conformations, but with the same graph topology given by the covalent bonding structure. Recall that each iteration of GWL aggregates geometric information $\boldsymbol{g}_i^{(k)}$ from progressively larger neighbourhoods $\mathcal{N}_i^{(k)}$ around the node $i$, and distinguishes (sub-)graphs via comparing $\mathfrak{G}$-orbit injective colouring of $\boldsymbol{g}_i^{(k)}$. We say $\mathcal{G}_1$ and $\mathcal{G}_2$ are *$k$-hop distinct* if for all graph isomorphisms $b$, there is some node $i \in \mathcal{V}_1, b(i) \in \mathcal{V}_2$ such that the corresponding $k$-hop subgraphs $\mathcal{N}_i^{(k)}$ and $\mathcal{N}_{b(i)}^{(k)}$ are distinct. Otherwise, we say $\mathcal{G}_1$ and $\mathcal{G}_2$ are *$k$-hop identical* if all $\mathcal{N}_i^{(k)}$ and $\mathcal{N}_{b(i)}^{(k)}$ are identical up to group actions.

Addresses R2.3

We can now formalise what geometric graphs can and cannot be distinguished by GWL. Proofs are available in Appendix D.

**Proposition 1.** *GWL can distinguish any $k$-hop distinct geometric graphs $\mathcal{G}_1$ and $\mathcal{G}_2$ where the underlying attributed graphs are isomorphic, and $k$ iterations are sufficient.*

**Proposition 2.** *Up to $k$ iterations, GWL cannot distinguish any $k$-hop identical geometric graphs $\mathcal{G}_1$ and $\mathcal{G}_2$ where the underlying attributed graphs are isomorphic.*

Additionally, we can state the following results about the more constrained IGWL.

**Proposition 3.** *IGWL can distinguish any 1-hop distinct geometric graphs $\mathcal{G}_1$ and $\mathcal{G}_2$ where the underlying attributed graphs are isomorphic, and 1 iteration is sufficient.*

**Proposition 4.** *Any number of iterations of IGWL cannot distinguish any* 1*-hop identical geometric graphs* $\mathcal{G}_1$ *and* $\mathcal{G}_2$ *where the underlying attributed graphs are isomorphic.*

An example illustrating Propositions 1 and 4 is shown in Figures 1 and 2, respectively.

We can now consider the more general case where the underlying attributed graphs for $\mathcal{G}_1 = (\boldsymbol{A}_1, \boldsymbol{S}_1, \vec{\boldsymbol{V}}_1, \vec{\boldsymbol{X}}_1)$ and $\mathcal{G}_2 = (\boldsymbol{A}_2, \boldsymbol{S}_2, \vec{\boldsymbol{V}}_2, \vec{\boldsymbol{X}}_2)$ are non-isomorphic and constructed from point clouds using radial cutoffs, as conventional for biochemistry and material science applications.

**Proposition 5.** *Assuming geometric graphs are constructed from point clouds using radial cutoffs, GWL can distinguish any geometric graphs* $\mathcal{G}_1$ *and* $\mathcal{G}_2$ *where the underlying attributed graphs are non-isomorphic. At most* $k_{Max}$ *iterations are sufficient, where* $k_{Max}$ *is the maximum graph diameter among* $\mathcal{G}_1$ *and* $\mathcal{G}_2$.

These results enable us to compare the expressive powers of GWL and IGWL.

**Theorem 6.** *GWL is strictly more powerful than IGWL.*

This statement formalises the advantage of $\mathfrak{G}$-equivariant intermediate layers for graphs and geometric data, as prescribed in the Geometric Deep Learning blueprint (Bronstein et al., 2021), in addition to echoing similar intuitions in the computer vision community. As remarked by Hinton et al. (2011), translation invariant models do not understand the relationship between the various parts of an image (colloquially called the "Picasso problem"). Similarly, our results point to IGWL failing to understand how the various 1-hops of a graph are stitched together. Finally, we identify a setting where this distinction between the two approaches disappears.

**Proposition 7.** *IGWL has the same expressive power as GWL for fully connected geometric graphs.*

### 3.2 CHARACTERISING THE EXPRESSIVE POWER OF GEOMETRIC GNNs

We would like to characterise the maximum expressive power of geometric GNNs based on the GWL test. Firstly, we show that any message passing $\mathfrak{G}$-equivariant GNN can be at most as powerful as GWL in distinguishing non-isomorphic geometric graphs. Proofs are available in Appendix E.

**Theorem 8.** *Any pair of geometric graphs distinguishable by a* $\mathfrak{G}$-*equivariant GNN is also distinguishable by GWL.*

With a sufficient number of iterations, the output of $\mathfrak{G}$-equivariant GNNs can be equivalent to GWL if certain conditions are met regarding the aggregate, update and readout functions.

**Proposition 9.** $\mathfrak{G}$-*equivariant GNNs have the same expressive power as GWL if the following conditions hold: (1) The aggregation* AGG *is an injective,* $\mathfrak{G}$-*equivariant multiset function. (2) The scalar part of the update* $\text{UPD}_s$ *is a* $\mathfrak{G}$-*orbit injective,* $\mathfrak{G}$-*invariant multiset function. (3) The vector part of the update* $\text{UPD}_v$ *is an injective,* $\mathfrak{G}$-*equivariant multiset function. (4) The graph-level readout* $f$ *is an injective multiset function.*

Similar statements can be made for $\mathfrak{G}$-invariant GNNs and IGWL. Thus, we can directly transfer our results about GWL and IGWL to the class of GNNs bounded by the respective tests. This has several interesting practical implications, discussed subsequently.

Addresses R1.3

## 4 UNDERSTANDING THE DESIGN SPACE OF GEOMETRIC GNNs VIA GWL

**Overview.** We now use the GWL framework to better understand key design choices for building geometric GNNs (Batatia et al., 2022a): (1) Depth or number of layers; and (2) Body order of invariant scalars. In doing so, we formalise theoretical limitations of current architectures and provide practical implications. Proofs are available in Appendix F.

### 4.1 ROLE OF DEPTH: PROPAGATING GEOMETRIC INFORMATION

Each iteration of GWL expands the neighbourhood from which geoemtric information can be gathered. We leveraged this construction in Section 3.1 to formalise the number of GWL iterations required to distinguish classes of geometric graphs.

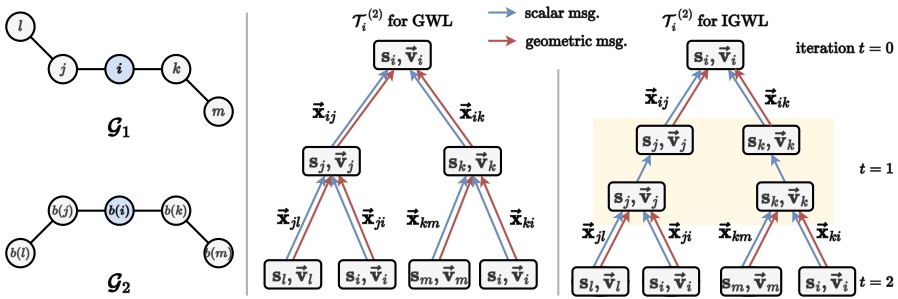

Figure 3: **Geometric Computation Trees for GWL and IGWL.** Unlike GWL, geometric orientation information cannot flow from the leaves to the root in IGWL, restricting its expressive power. IGWL cannot distinguish $\mathcal{G}_1$ and $\mathcal{G}_2$ as all 1-hop neighbourhoods are computationally identical.

Consequently, stacking multiple $\mathfrak{G}$-equivariant GNN layers enables the computation of compositional geometric features. This can be understood via a geometric version of computation trees (Garg et al., 2020). A computation tree $\mathcal{T}_i^{(t)}$ represents the maximum information contained in GWL/IGWL colours or GNN features at iteration $t$ by an 'unrolling' of the message passing procedure. Geometric computation trees are constructed recursively: $\mathcal{T}_i^{(0)} = (\boldsymbol{s}_i, \vec{\boldsymbol{v}}_i)$ for all $i \in \mathcal{V}$. For $t > 0$, we start with a root node $(\boldsymbol{s}_i, \vec{\boldsymbol{v}}_i)$ and add a child subtree $\mathcal{T}_j^{(t-1)}$ for all $j \in \mathcal{N}_i$ along with the relative position $\vec{x}_{ij}$ along the edge, as shown in Figure 3. To obtain the root node's embedding or colour, both scalar and geometric information is propagated from the leaves up to the root. Thus, if two nodes have identical geometric computation trees, they will be mapped to the same node embedding or colour. Critically, geometric orientation information cannot flow from one level to another in the computation trees for IGWL and $\mathfrak{G}$-invariant GNNs, as they only update scalar information. In the recursive construction procedure, we must insert a connector node $(\boldsymbol{s}_j, \vec{\boldsymbol{v}}_j)$ before adding the child subtree $\mathcal{T}_j^{(t-1)}$ for all $j \in \mathcal{N}_i$ and prevent geometric information propagation between them.

Addresses R2.3

As a result, even the most powerful $\mathfrak{G}$-invariant GNNs are restricted in their ability to compute global and non-local geometric properties.

**Proposition 10.** *IGWL and $\mathfrak{G}$-invariant GNNs cannot decide several geometric graph properties: (1) perimeter, surface area, and volume of the bounding box/sphere enclosing the geometric graph; (2) distance from the centroid or centre of mass; and (3) dihedral angles.*

**Practical Implications.** Proposition 10, together with Propositions 1 and 4, highlight critical theoretical limitations of $\mathfrak{G}$-invariant GNNs. Our results suggest that $\mathfrak{G}$-equivariant GNNs should be preferred when working with large geometric graphs such as macromolecules with thousands of nodes, where message passing is restricted to local radial neighbourhoods around each node.

Motivated by these limitations, two straightforward approaches to improving $\mathfrak{G}$-invariant GNNs may be: (1) pre-computing non-local geometric properties as input features, *e.g.* models such as GemNet (Gasteiger et al., 2021) and GearNet (Zhang et al., 2022) already use two-hop dihedral angles. And (2) working with fully connected geometric graphs, as Proposition 7 suggests that $\mathfrak{G}$-equivariant and $\mathfrak{G}$-invariant GNNs can be made equally powerful when performing all-to-all message passing. This is supported by the empirical success of recent $\mathfrak{G}$-invariant 'Graph Transformers' (Joshi, 2020; Shi et al., 2022) for small molecules with tens of nodes, where working with full graphs is tractable.

## 4.2 ROLE OF BODY ORDER: DISTINGUISHING $\mathfrak{G}$-ORBITS

At each iteration of GWL and IGWL, the I-HASH function assigns a $\mathfrak{G}$-invariant colouring to distinct geometric neighbourhood patterns. I-HASH is an idealised $\mathfrak{G}$-orbit injective function which is not necessarily continous. In geometric GNNs, this corresponds to scalarising local geometric information when updating the scalar features; examples are shown in equation 11 and equation 12. We can analyse the construction of the I-HASH function and the scalarisation step in geometric GNNs via the $k$-body variations IGWL$_{(k)}$.

Firstly, we formalise the relationship between the injectivity of I-$\text{HASH}_{(k)}$ and the maximum cardinality of local neighbourhoods in a given dataset.

**Proposition 11.** I-$\text{HASH}_{(m)}$ *is $\mathfrak{G}$-orbit injective for $m = max(\{|\mathcal{N}_i| \mid i \in \mathcal{V}\})$, the maximum cardinality of all local neighbourhoods $\mathcal{N}_i$ in a given dataset.*

**Practical Implications.** While building provably injective I-$\text{HASH}_{(k)}$ functions may require intractably high $k$, the hierarchy of $\text{IGWL}_{(k)}$ tests enable us to study the expressive power of practical $\mathfrak{G}$-invariant aggregators used in current geometric GNN layers, *e.g.* SchNet (Schütt et al., 2018), E-GNN (Satorras et al., 2021), and TFN (Thomas et al., 2018) use distances; DimeNet (Gasteiger et al., 2020) uses distances and angles. Notably, MACE (Batatia et al., 2022b) constructs a *complete* basis of scalars up to arbitrary body order $k$ via Atomic Cluster Expansion (Dusson et al., 2019), which can be $\mathfrak{G}$-orbit injective if the conditions in Proposition 11 are met. We can state the following about the $\text{IGWL}_{(k)}$ hierarchy and the corresponding GNNs.

> Addresses R2.7

> Addresses R4.4

**Proposition 12.** *$\text{IGWL}_{(k)}$ is at least as powerful as $\text{IGWL}_{(k-1)}$. For $k \leq 5$, $\text{IGWL}_{(k)}$ is strictly more powerful than $\text{IGWL}_{(k-1)}$.*

Finally, we show that $\text{IGWL}_{(2)}$ is equivalent to WL when all the pairwise distances between the nodes are the same. A similar observation was recently made by Pozdnyakov & Ceriotti (2022).

**Proposition 13.** *Let $\mathcal{G}_1 = (\boldsymbol{A}_1, \boldsymbol{S}_1, \vec{\boldsymbol{X}}_1)$ and $\mathcal{G}_2 = (\boldsymbol{A}_2, \boldsymbol{S}_2, \vec{\boldsymbol{X}}_2)$ be two geometric graphs with the property that all edges have equal length. Then, $\text{IGWL}_{(2)}$ distinguishes the two graphs if and only if WL can distinguish the attributed graphs $(\boldsymbol{A}_1, \boldsymbol{S}_1)$ and $(\boldsymbol{A}_1, \boldsymbol{S}_1)$.*

This equivalence points to limitations of distance-based $\mathfrak{G}$-invariant models like SchNet (Schütt et al., 2018). These models suffer from all well-known failure cases of WL, *e.g.* they cannot distinguish two equilateral triangles from the regular hexagon (Gasteiger et al., 2020).

**Synthetic Experiments.** Appendix C contains additional synthetic experiments supplementing our results and highlighting practical challenges in building powerful geometric GNNs, *s.a.* oversmoothing and oversquashing with increased depth, as well as designing efficient higher order aggregators.

> Addresses R1.4, R2.5, R3.1, R4.1

## 5 DISCRIMINATION AND UNIVERSALITY

**Overview.** Following Chen et al. (2019), we study the equivalence between the universal approximation capabilities of geometric GNN models (Dym & Maron, 2020) and perspective of discriminating geometric graphs introduced by GWL, generalising their results to any isomorphism induced by a compact Lie group $\mathfrak{G}$. We further study the number of invariant aggregators that are required in a continuous setting to distinguish any two neighbourhoods. Proofs are available in Appendix G.

In the interest of generality, we use a general space $X$ acted upon by a compact group $\mathfrak{G}$ and we are interested in the capacity of $\mathfrak{G}$-invariant functions over $X$ to separate points in $Y$. The restriction to a smaller subset $Y$ is useful because we would like to separately consider the case when $Y$ is countable due to the use of countable features. Therefore, in general, the action of $\mathfrak{G}$ on $Y$ might not be strictly defined since it might yield elements outside $Y$. For our setting, the reader could take $X = (\mathbb{R}^d \times \mathbb{R}^f)^{n \times n}$ to be the space of $n \times n$ geometric graphs and $Y = \mathcal{X}^{n \times n}$, where $\mathcal{X} \subseteq \mathbb{R}^d \times \mathbb{R}^f$.

> Addresses R2.4

**Definition 14.** *Let $\mathfrak{G}$ be a compact group and $\mathcal{C}$ a collection of $\mathfrak{G}$-invariant functions from a set $X$ to $\mathbb{R}$. For a subset $Y \subseteq X$, we say the $\mathcal{C}$ is **pairwise $\boldsymbol{Y}_\mathfrak{G}$ discriminating** if for any $y_1, y_2 \in Y$ such that $y_1 \not\simeq y_2$, there exists a function $h \in \mathcal{C}$ such that $h(y_1) \neq h(y_2)$.*

We note here that $h$ is not necessarily injective, *i.e.* there might be $y_1', y_2'$ for which $h(y_1') = h(y_1')$. Therefore, pairwise discrimination is a weaker notion of discrimination than the one GWL uses.

> Addresses R2.4

**Definition 15.** *Let $\mathfrak{G}$ be a compact group and $\mathcal{C}$ a collection of $\mathfrak{G}$-invariant functions from $X$ to $\mathbb{R}$. For $Y \subseteq X$, we say the $\mathcal{C}$ is **universally approximating** over $Y$ if for all $\mathfrak{G}$-invariant functions $f$ from $X$ to $\mathbb{R}$ and for all $\epsilon > 0$, there exists $h_{\varepsilon,f} \in \mathcal{C}$ such that $\|f - h_{\varepsilon,f}\|_Y := \sup_{y \in Y} |f(y) - h(y)| < \varepsilon$.*

**Countable Features.** We first focus on the countable feature setting, which is also the setting where the GWL test operates. Therefore, we will assume that $Y$ is a countable subset of $X$.

**Theorem 16.** *If $\mathcal{C}$ is universally approximating over $Y$, then $\mathcal{C}$ is also pairwise $Y_\mathfrak{G}$ discriminating.*

This result further motivates the interest in discriminating geometric graphs, since a model that cannot distinguish two non-isomorphic geometric graphs is not universal. By further assuming that $Y$ is finite, we obtain a result in the opposite direction. Given a collection of functions $\mathcal{C}$, we define like in Chen et al. (2019) the class $\mathcal{C}^{+L}$ given by all the functions of the form MLP($[f_1(x), \ldots, f_k(x)]$) with $f_i \in \mathcal{C}$ and finite $k$, where the MLP has $L$ layers with ReLU hidden activations.

**Theorem 17.** *If $\mathcal{C}$ is pairwise $Y_{\mathfrak{G}}$ discriminating, then $\mathcal{C}^{+2}$ is universally approximating over $Y$.*

**Continous Features.** The symmetries characterising geometric graphs are naturally continuous (*e.g.* rotations). Therefore, it is natural to ask how the results above translate to *continuous* $\mathfrak{G}$-invariant functions over a continuous subspace $Y$. Therefore, for the rest of this section, we assume that $(X, d)$ is a metric space, $Y$ is a compact subset of $X$ and $\mathfrak{G}$ acts continuously on $X$.

**Theorem 18.** *If $\mathcal{C}$ can universally approximate any continuous $\mathfrak{G}$-invariant functions on $Y$, then $\mathcal{C}$ is also pairwise $Y_{\mathfrak{G}}$ discriminating.*

With the additional assumption that $X = Y$, we can also prove the converse.

**Theorem 19.** *If $\mathcal{C}$, a class of functions over $Y$, is pairwise $Y_{\mathfrak{G}}$ discriminating, then $\mathcal{C}^{+2}$ can also universally approximate any continuous function over $Y$.*

We now produce an estimate for the number of aggregators needed to learn *continous* orbit-space injective functions on a manifold $X$ based on results from differential geometry (Lee, 2013). A group $\mathfrak{G}$ acts freely on $X$ if $\mathfrak{g}x = x$ implies $\mathfrak{g} = e_{\mathfrak{G}}$, where $e_{\mathfrak{G}}$ is the identity element in $\mathfrak{G}$.

**Theorem 20.** *Let $X$ be a smooth $n$-dim manifold and $\mathfrak{G}$ an $m$-dim compact Lie group acting continuously on $X$. Suppose there exists a smooth submanifold $Y$ of $X$ of the same dimension such that $\mathfrak{G}$ acts freely on it. Then, any $\mathfrak{G}$-orbit injective function $f : X \to \mathbb{R}^d$ requires that $d \geq n - m$.*

We now apply this theorem to the local aggregation operation performed by geometric GNNs. Let $X = \mathbb{R}^{n \times d}$ and $\mathfrak{G} = S_n \times O(d)$ or $S_n \times SO(d)$. Let $\boldsymbol{P}_{\mathfrak{g}}$ and $\boldsymbol{Q}_{\mathfrak{g}}$ be the permutation matrix and the orthogonal matrix associated with the group element $\mathfrak{g} \in \mathfrak{G}$. Then $\mathfrak{g}$ acts on matrices $\boldsymbol{X} \in X$ continuously via $\boldsymbol{P}_{\mathfrak{g}}\boldsymbol{X}\boldsymbol{Q}_{\mathfrak{g}}^{\top}$. Then, $\mathfrak{G}$ orbit-space injective functions on $X$ are functions on point clouds of size $n$ that can distinguish any two different point clouds.

**Theorem 21.** *For $n \geq d - 1 > 0$ or $n = d = 1$, any continuous $S_n \times SO(d)$ orbit-space injective function $f : \mathbb{R}^{n \times d} \to \mathbb{R}^q$ requires that $q \geq nd - d(d-1)/2$.*

We can also generalise this to $O(d)$, with the slightly stronger assumption that $n \geq d$.

**Theorem 22.** *For $n \geq d > 0$, any continuous $S_n \times O(d)$ orbit-space injective function $f : \mathbb{R}^{n \times d} \to \mathbb{R}^q$ requires that $q \geq nd - d(d-1)/2$.*

Overall, these results show that when working with point clouds in $\mathbb{R}^3$ as is common in molecular or physical applications, at least $q = 3(n-1)$ aggregators are required. This result argues why a bigger representational width can help distinguish neighbourhoods. Finally, in the particular case of the zero-dimensional subgroup $S_n \times \{e_{SO(d)}\} \simeq S_n$ we obtain a statement holding for all $n$ and generalising a result from PNA Corso et al. (2020) regarding the aggregators for non-geometric GNNs. The original PNA result considers the case $d = 1$ and here we extend it to arbitrary $d$.

**Proposition 23.** *Any $S_n$-invariant injective function $f : \mathbb{R}^{n \times d} \to \mathbb{R}^q$ requires $q \geq nd$.*

# 6 DISCUSSION

This work proposes a geometric version of the Weisfeiler-Leman graph isomorphism test (GWL) for discriminating geometric graphs while respecting the underlying spatial symmetries. We use GWL to characterise the expressive power of geometric GNNs and connect the universal approximation properties of these models to discriminating geometric graphs.

GWL provides an abstraction to study the limits of geometric GNNs. In practice it is challenging to build maximally powerful GNNs that satisfy the conditions of Proposition 9 as GWL relies on perfect colouring and aggregation functions to identify distinct neighbourhoods and propogate their geometric orientation information, respectively. Based on the intuitions gained from GWL, future work will explore building provably powerful, *practical* geometric GNNs for applications in biochemistry, material science, and multiagent robotics, and better characterise the trade-offs related to practical implementation choices.

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

## A  RELATED WORK

Literature on the *completeness* of atom-centred interatomic potentials has focused on distinguishing 1-hop local neighbourhoods (point clouds) around atoms by building spanning sets for continuous, 𝔊-equivariant multiset functions (Shapeev, 2016; Drautz, 2019; Dusson et al., 2019; Pozdnyakov et al., 2020). Recent theoretical work on geometric GNNs and their universality has shown that architectures such as TFN, GemNet and GVP-GNN (Dym & Maron, 2020; Villar et al., 2021; Gasteiger et al., 2021; Jing et al., 2020) can be universal approximators of continuous, 𝔊-equivariant or 𝔊-invariant multiset functions over point clouds, *i.e.* fully connected graphs. In contrast, the GWL framework studies the expressive power of geometric GNNs operating on sparse graphs from the perspective of discriminating geometric graphs and the graph isomorphism problem. The discrimination lens is potentially more granular and practically insightful than universality. A model may either be universal or not. On the other hand, there could be multiple degrees of discrimination depending on the classes of geometric graphs that can and cannot be distinguished, which our work aims to formalise.

Addresses R2.4

## B    ADDITIONAL BACKGROUND ON GEOMETRIC GNNS

The GWL framework can be used to characterise the expressive power and theoretical limitations of two broad classes of geometric GNNs.

$\mathfrak{G}$-**invariant GNNs.**   $\mathfrak{G}$-invariant GNN layers aggregate scalar quantities from local neighbourhoods via scalarising the geometric information. Scalar features are update from iteration $t$ to $t + 1$ via learnable aggregate and update functions, AGG and UPD, respectively:

$$s_i^{(t+1)} := \text{UPD}\left(s_i^{(t)}\,,\,\text{AGG}\left(\{\!\{(s_i^{(t)}, s_j^{(t)}, \vec{v}_i, \vec{v}_j, \vec{x}_{ij}) \mid j \in \mathcal{N}_i\}\!\}\right)\right). \tag{10}$$

For *e.g.*, SchNet (Schütt et al., 2018) uses relative distances $\|\vec{x}_{ij}\|$ to scalarise local geometric information, while DimeNet (Gasteiger et al., 2020) uses both distances as well as angles $\vec{x}_{ij} \cdot \vec{x}_{ik}$ among triplets, as follows:

$$s_i^{(t+1)} := s_i^{(t)} + \sum_{j \in \mathcal{N}_i} f_1\left(s_j^{(t)}, \|\vec{x}_{ij}\|\right) \tag{SchNet} \tag{11}$$

$$s_i^{(t+1)} := \sum_{j \in \mathcal{N}_i} f_1\left(s_i^{(t)}, s_j^{(t)}, \sum_{k \in \mathcal{N}_i \setminus \{j\}} f_2\left(s_j^{(t)}, s_k^{(t)}, \|\vec{x}_{ij}\|, \vec{x}_{ij} \cdot \vec{x}_{ik}\right)\right) \tag{DimeNet} \tag{12}$$

> Addresses R4.3

$\mathfrak{G}$-**equivariant GNNs.**   $\mathfrak{G}$-equivariant GNN layers update both scalar and vector features by propagating scalar as well as vector messages, $\boldsymbol{m}_i^{(t)}$ and $\vec{\boldsymbol{m}}_i^{(t)}$, respectively:

$$\boldsymbol{m}_i^{(t)}, \vec{\boldsymbol{m}}_i^{(t)} := \text{AGG}\left(\{\!\{(s_i^{(t)}, s_j^{(t)}, \vec{v}_i^{(t)}, \vec{v}_j^{(t)}, \vec{x}_{ij}) \mid j \in \mathcal{N}_i\}\!\}\right) \tag{Aggregate} \tag{13}$$

$$s_i^{(t+1)}, \vec{v}_i^{(t+1)} := \text{UPD}\left((s_i^{(t)}, \vec{v}_i^{(t)})\,,\,(\boldsymbol{m}_i^{(t)}, \vec{\boldsymbol{m}}_i^{(t)})\right) \tag{Update} \tag{14}$$

For *e.g.*, PaiNN (Schütt et al., 2021) interaction layers aggregate scalar and vector features via learnt filters conditioned on the relative distance:

$$\boldsymbol{m}_i^{(t)} := s_i^{(t)} + \sum_{j \in \mathcal{N}_i} f_1\left(s_j^{(t)}, \|\vec{x}_{ij}\|\right) \tag{15}$$

$$\vec{\boldsymbol{m}}_i^{(t)} := \vec{v}_i^{(t)} + \sum_{j \in \mathcal{N}_i} f_2\left(s_j^{(t)}, \|\vec{x}_{ij}\|\right) \odot \vec{v}_j^{(t)} + \sum_{j \in \mathcal{N}_i} f_3\left(s_j^{(t)}, \|\vec{x}_{ij}\|\right) \odot \vec{x}_{ij} \tag{16}$$

E-GNN (Satorras et al., 2021) and GVP-GNN (Jing et al., 2020) use similar operations. The update step applies a gated non-linearity (Weiler et al., 2018) on the vector features, which learns to scale their magnitude using their norm concatenated with the scalar features:

$$s_i^{(t+1)} := \boldsymbol{m}_i^{(t)} + f_4\left(\boldsymbol{m}_i^{(t)}, \|\vec{\boldsymbol{m}}_i^{(t)}\|\right), \qquad \vec{v}_i^{(t+1)} := \vec{\boldsymbol{m}}_i^{(t)} + f_5\left(\boldsymbol{m}_i^{(t)}, \|\vec{\boldsymbol{m}}_i^{(t)}\|\right) \odot \vec{\boldsymbol{m}}_i^{(t)}. \tag{17}$$

The updated scalar features are both $\mathfrak{G}$-invariant and $T(d)$-invariant as the only geometric information used is the relative distances, while the updated vector features are $\mathfrak{G}$-equivariant and $T(d)$-invariant as they aggregate $\mathfrak{G}$-equivariant, $T(d)$-invariant vector quantities from the neighbours.

> Addresses R4.3

Another example of $\mathfrak{G}$-equivariant GNNs is the `e3nn` framework (Geiger & Smidt, 2022), which can be used to instantiate Tensor Field Network (Thomas et al., 2018), Cormorant (Anderson et al., 2019), SEGNN (Brandstetter et al., 2022), and MACE (Batatia et al., 2022b). These models use higher order spherical tensors $\tilde{\boldsymbol{h}}_{i,l} \in \mathbb{R}^{2l+1 \times f}$ as node feature, starting from order $l = 0$ up to arbitrary $l = L$. The first two orders correspond to scalar and vector features, respectively. The higher order tensors $\tilde{\boldsymbol{h}}_i$ are updated via tensor products of neighbourhood features $\tilde{\boldsymbol{h}}_j$ for all $j \in \mathcal{N}_i$ with the higher order spherical harmonic representations $Y$ of the relative displacement $\frac{\vec{x}_{ij}}{\|\vec{x}_{ij}\|} = \hat{\boldsymbol{x}}_{ij}$:

> Addresses R2.2

$$\tilde{\boldsymbol{h}}_i^{(t+1)} := \tilde{\boldsymbol{h}}_i^{(t)} + \sum_{j \in \mathcal{N}_i} Y(\hat{\boldsymbol{x}}_{ij}) \otimes_{\boldsymbol{w}} \tilde{\boldsymbol{h}}_j^{(t)}, \tag{18}$$

where the weights $\boldsymbol{w}$ of the tensor product are computed via a learnt radial basis function of the relative distance, *i.e.* $\boldsymbol{w} = f(\|\vec{x}_{ij}\|)$. To obtain the entry $m_3 \in \{-l_3, \ldots, +l_3\}$ for the order-$l_3$ part

of the updated higher order tensors $\tilde{\boldsymbol{h}}_i^{(t+1)}$, we can expand the tensor product in equation 18 as:

$$\tilde{\boldsymbol{h}}_{i,l_3m_3}^{(t+1)} := \tilde{\boldsymbol{h}}_{i,l_3m_3}^{(t)} + \sum_{l_1m_1,l_2m_2}^{l_3m_3} C_{l_1m_1,l_2m_2}^{l_3m_3} \sum_{j \in \mathcal{N}_i} f_{l_1l_2l_3}\left(\|\vec{\boldsymbol{x}}_{ij}\|\right) Y_{l_1}^{m_1}\left(\hat{\boldsymbol{x}}_{ij}\right) \tilde{\boldsymbol{h}}_{j,l_2m_2}^{(t)}, \qquad (19)$$

where $C_{l_1m_1,l_2m_2}^{l_3m_3}$ are the Clebsch-Gordan coefficients ensuring that the updated features are equivariant. Notably, when restricting the tensor product to only scalars ($l = 0$) and vectors ($l = 1$), we obtain updates of the form similar to equation 15, equation 16 and equation 17.

**Applications.**   Systems in biochemistry, material science, physical simulations, and multiagent robotics can be modelled using geometric GNNs. Invariant GNNs have shown strong performance for protein design (Zhang et al., 2022; Dauparas et al., 2022) and electrocatalysis (Gasteiger et al., 2021; Shi et al., 2022), while equivariant GNNs are being used within learnt interatomic potentials for molecular dynamics (Schütt et al., 2021; Batzner et al., 2022; Batatia et al., 2022b).

> Addressed R4.1

## C   SYNTHETIC EXPERIMENTS FOR GEOMETRIC GNN DESIGN SPACE

> Addresses R1.4, R2.5, R3.1, R4.1

We perform three simple synthetic experiments to highlight the practical challenges of building maximally powerful geometric GNNs. We hope that our synthetic experiments and associated code can be a pedagogical tool for exploring the geometric GNN design space in future work.

**Setup and Hyperparameters.**   We experiment with the following models: (1) SchNet (Schütt et al., 2018) and DimeNet (Gasteiger et al., 2020) as representative $\mathfrak{G}$-invariant GNNs; (2) E-GNN (Satorras et al., 2021) and GVP-GNN (Jing et al., 2020) as representative $\mathfrak{G}$-equivariant GNNs which use scalars and vectors in $\mathbb{R}^3$; and (3) TFN (Thomas et al., 2018) and MACE (Batatia et al., 2022b) to study higher order $\mathfrak{G}$-equivariant GNNs using spherical tensors. For SchNet and DimeNet, we use the implementation from PyTorch Geometric (Fey & Lenssen, 2019). For E-GNN, GVP-GNN, and MACE, we adapt implementations from the respective authors. Our TFN implementation is based on e3nn (Geiger & Smidt, 2022), and we also re-implement MACE by incorporating the `EquivariantProductBasisBlock` from its authors into our TFN layer. We set scalar feature channels to 128 for SchNet, DimeNet, and EGNN. We set scalar/vector/tensor feature channels to 64 for GVP, TFN, MACE. TFN and MACE use order $L = 2$ tensors by default. MACE uses local body order 4 by default. We train all models for 100 epochs using the Adam optimiser, with an initial learning rate $1e-4$, which we reduce by a factor of 0.9 and a patience of 25 epochs when the performance plateaus. All results are averaged across 10 random seeds.

**Identifying neighbourhood fingerprints: counterexamples from Pozdnyakov et al. (2020).** GWL uses a node colouring function I-HASH for distinguishing $\mathfrak{G}$-orbits of neighbourhoods, *i.e.* a neighbourhood fingerprint. In geometric GNNs, this corresponds to a scalarisation step where local geometric information from subsets of neighbours is aggregated to compute $\mathfrak{G}$-invariant scalars (termed the body order).

> Addresses R2.3, R4.4

To demonstrate the practical implications of scalarisation body order, we evaluate current geometric GNN layers on their ability to discriminate counterexamples from Pozdnyakov et al. (2020). Each counterexample consists of a pair of local neighbourhoods that are indistinguishable when comparing their set of $k$-body scalars, *i.e.* I-HASH$_{(k)}$ and geometric GNN layers with body order $k$ cannot distinguish the neighbourhoods. The 3-body counterexample corresponds to Fig.1(b) in (Pozdnyakov et al., 2020), 4-body chiral to Fig.2(e), and 4-body non-chiral to Fig.2(f); the 2-body counterexample is based on the two local neighbourhoods in our running example.

In Table 1, we train single layer geometric GNNs to distinguish the counterexamples using updated scalar features. Unsurprisingly, we find that most layers computing 2 or 3 body scalarisations fail the task. Notably, training higher body order MACE layers to distinguish the chiral and non-chiral 4-body counterexamples should be theoretically possible, but proved challenging in practice. This highlights the difficulty of designing as well as optimising continuous, high body order neighbourhood fingerprints.

**Identifying neighbourhood orientation: rotationally symmetric structures.**   GWL is able to perfectly aggregate $\mathfrak{G}$-equivariant geometric information without losing neighbourhood orientation by making use of an auxiliary nested geometric object $\boldsymbol{g}_i$. On the other hand, $\mathfrak{G}$-equivariant GNNs

| | GNN Layer | Counterexample from Pozdnyakov et al. (2020) | | | |
| --- | --- | --- | --- | --- | --- |
| | | 2-body | 3-body (Fig.1(b)) | 4-body non-chiral (Fig.2(f)) | 4-body chiral (Fig.2(e)) |
| Inv. | SchNet$_{\text{2-body}}$ | $50.0 \pm 0.0$ | $50.0 \pm 0.0$ | $50.0 \pm 0.0$ | $50.0 \pm 0.0$ |
| | DimeNet$_{\text{3-body}}$ | $\textbf{100.0} \pm \textbf{0.0}$ | $50.0 \pm 0.0$ | $50.0 \pm 0.0$ | $50.0 \pm 0.0$ |
| $O(3)$-Equiv. | E-GNN$_{\text{2-body}}$ | $50.0 \pm 0.0$ | $50.0 \pm 0.0$ | $50.0 \pm 0.0$ | $50.0 \pm 0.0$ |
| | GVP-GNN$_{\text{3-body}}$ | $\textbf{100.0} \pm \textbf{0.0}$ | $50.0 \pm 0.0$ | $50.0 \pm 0.0$ | $50.0 \pm 0.0$ |
| | TFN$_{\text{2-body}}$ | $50.0 \pm 0.0$ | $50.0 \pm 0.0$ | $50.0 \pm 0.0$ | $50.0 \pm 0.0$ |
| | MACE$_{\text{3-body}}$ | $\textbf{100.0} \pm \textbf{0.0}$ | $50.0 \pm 0.0$ | $50.0 \pm 0.0$ | $50.0 \pm 0.0$ |
| | MACE$_{\text{4-body}}$ | $\textbf{100.0} \pm \textbf{0.0}$ | $\textbf{100.0} \pm \textbf{0.0}$ | $50.0 \pm 0.0$ | $50.0 \pm 0.0$ |
| | MACE$_{\text{5-body}}$ | $\textbf{100.0} \pm \textbf{0.0}$ | $\textbf{100.0} \pm \textbf{0.0}$ | $50.0 \pm 0.0$ | $50.0 \pm 0.0$ |
| $SO(3)$-Eq. | TFN$_{\text{2-body}}$ | $50.0 \pm 0.0$ | $50.0 \pm 0.0$ | $50.0 \pm 0.0$ | $50.0 \pm 0.0$ |
| | MACE$_{\text{3-body}}$ | $\textbf{100.0} \pm \textbf{0.0}$ | $50.0 \pm 0.0$ | $50.0 \pm 0.0$ | $50.0 \pm 0.0$ |
| | MACE$_{\text{4-body}}$ | $\textbf{100.0} \pm \textbf{0.0}$ | $\textbf{100.0} \pm \textbf{0.0}$ | $50.0 \pm 0.0$ | $50.0 \pm 0.0$ |
| | MACE$_{\text{5-body}}$ | $\textbf{100.0} \pm \textbf{0.0}$ | $\textbf{100.0} \pm \textbf{0.0}$ | $50.0 \pm 0.0$ | $50.0 \pm 0.0$ |

Table 1: *Counterexamples from Pozdnyakov et al. (2020).* We train single layer geometric GNNs to distinguish each counterexample pair of local neighbourhoods that are indistinguishable using $k$-body scalarisation. Most current geometric GNN layers are restricted to body order 2 or 3 and fail the tasks. Distinguishing the 4-body counterexamples should be theoretically possible with higher body order MACE layers, but proved challenging in practice. **This highlights the difficulty of designing as well as optimising high body order neighbourhood fingerprints beyond simple distances and angles.** Anomolous results are marked in red and expected results in green .

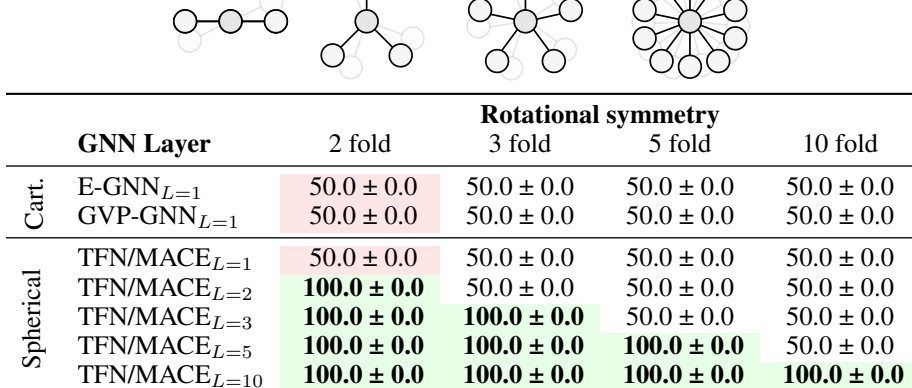

| | GNN Layer | Rotational symmetry | | | |
| --- | --- | --- | --- | --- | --- |
| | | 2 fold | 3 fold | 5 fold | 10 fold |
| Cart. | E-GNN$_{L=1}$ | $50.0 \pm 0.0$ | $50.0 \pm 0.0$ | $50.0 \pm 0.0$ | $50.0 \pm 0.0$ |
| | GVP-GNN$_{L=1}$ | $50.0 \pm 0.0$ | $50.0 \pm 0.0$ | $50.0 \pm 0.0$ | $50.0 \pm 0.0$ |
| Spherical | TFN/MACE$_{L=1}$ | $50.0 \pm 0.0$ | $50.0 \pm 0.0$ | $50.0 \pm 0.0$ | $50.0 \pm 0.0$ |
| | TFN/MACE$_{L=2}$ | $\textbf{100.0} \pm \textbf{0.0}$ | $50.0 \pm 0.0$ | $50.0 \pm 0.0$ | $50.0 \pm 0.0$ |
| | TFN/MACE$_{L=3}$ | $\textbf{100.0} \pm \textbf{0.0}$ | $\textbf{100.0} \pm \textbf{0.0}$ | $50.0 \pm 0.0$ | $50.0 \pm 0.0$ |
| | TFN/MACE$_{L=5}$ | $\textbf{100.0} \pm \textbf{0.0}$ | $\textbf{100.0} \pm \textbf{0.0}$ | $\textbf{100.0} \pm \textbf{0.0}$ | $50.0 \pm 0.0$ |
| | TFN/MACE$_{L=10}$ | $\textbf{100.0} \pm \textbf{0.0}$ | $\textbf{100.0} \pm \textbf{0.0}$ | $\textbf{100.0} \pm \textbf{0.0}$ | $\textbf{100.0} \pm \textbf{0.0}$ |

Table 2: *Rotationally symmetric structures.* We train single layer $\mathfrak{G}$-equivariant GNNs to distinguish two *distinct* rotated versions of each $L$-fold symmetric structure. **We find that layers using order $L$ tensors are unable to identify the orientation of structures with rotation symmetry higher than $L$-fold.** This issue is particularly prevalent for E-GNN and GVP-GNN (tensor order 1).

aggregate geometric information via summing neighbourhood features in fixed dimensional spaces using either cartesian vectors or higher order spherical tensors, which come with tradeoffs between tractability and empirical performance.

In Table 2, we study how rotational symmetries interact with tensor order in $\mathfrak{G}$-equivariant GNNs. We evaluate current $\mathfrak{G}$-equivariant layers on their ability to distinguish the orientation of structures with rotational symmetry. An $L$-fold symmetric structure does not change when rotated by an angle $\frac{2\pi}{L}$ around a point (in 2D) or axis (3D). We consider two *distinct* rotated versions of each $L$-fold symmetric structure and train single layer $\mathfrak{G}$-equivariant GNNs to classify the two orientations using the updated geometric features. We find that layers using order $L$ tensors are unable to identify the orientation of structures with rotation symmetry higher than $L$-fold. This observation can be

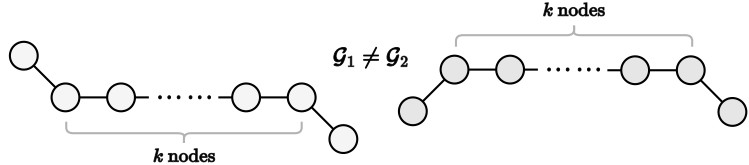

| ($k = 4$-chains) | | | **Number of layers** | | |
| **GNN Layer** | $\lfloor \frac{k}{2} \rfloor$ | $\lfloor \frac{k}{2} \rfloor + 1 = 3$ | $\lfloor \frac{k}{2} \rfloor + 2$ | $\lfloor \frac{k}{2} \rfloor + 3$ | $\lfloor \frac{k}{2} \rfloor + 4$ |
|---|---|---|---|---|---|
| **Inv.** IGWL | 50% | 50% | 50% | 50% | 50% |
| SchNet | $50.0 \pm 0.00$ | $50.0 \pm 0.00$ | $50.0 \pm 0.00$ | $50.0 \pm 0.00$ | $50.0 \pm 0.00$ |
| DimeNet | $50.0 \pm 0.00$ | $50.0 \pm 0.00$ | $50.0 \pm 0.00$ | $50.0 \pm 0.00$ | $50.0 \pm 0.00$ |
| **Equiv.** GWL | 50% | **100%** | **100%** | **100%** | **100%** |
| E-GNN | $50.0 \pm 0.0$ | $50.0 \pm 0.0$ | $50.0 \pm 0.0$ | $50.0 \pm 0.0$ | $\mathbf{100.0 \pm 0.0}$ |
| GVP-GNN | $50.0 \pm 0.0$ | $\mathbf{100.0 \pm 0.0}$ | $\mathbf{100.0 \pm 0.0}$ | $\mathbf{100.0 \pm 0.0}$ | $\mathbf{100.0 \pm 0.0}$ |
| TFN | $50.0 \pm 0.0$ | $50.0 \pm 0.0$ | $50.0 \pm 0.0$ | $\mathbf{80.0 \pm 24.5}$ | $\mathbf{85.0 \pm 22.9}$ |
| MACE | $50.0 \pm 0.0$ | $\mathbf{90.0 \pm 20.0}$ | $\mathbf{90.0 \pm 20.0}$ | $\mathbf{95.0 \pm 15.0}$ | $\mathbf{95.0 \pm 15.0}$ |

Table 3: *$k$-chain geometric graphs.* $k$-chains are $(\lfloor \frac{k}{2} \rfloor + 1)$-hop distinguishable and $(\lfloor \frac{k}{2} \rfloor + 1)$ GWL iterations are theoretically sufficient to distinguish them. We train geometric GNNs with an increasing number of layers to distinguish $k = 4$-chains. **$\mathfrak{G}$-equivariant GNNs may require more iterations that prescribed by GWL, pointing to preliminary evidence of oversmoothing and oversquashing when geometric information is propagated across multiple layers using fixed dimensional feature spaces.** IGWL and $\mathfrak{G}$-invariant GNNs are unable to distinguish $k$-chains for any $k \geq 2$ and $\mathfrak{G} = O(3)$.

attributed to the spherical harmonics which exhibits rotational symmetry themselves and are used as the underlying basis. Layers such as E-GNN and GVP-GNN using cartesian vectors (corresponding to tensor order 1) are popular as working with higher order tensors can be computationally intractable for many applications. However, E-GNN and GVP-GNN are particularly poor at discriminating orientation of rotationally symmetric structures. This may have implications for modelling of periodic materials which naturally exhibit such symmetries (Levine & Steinhardt, 1984).

**Propogating geometric information: $k$-chains.** In addition to perfect aggregation, GWL also assumes perfect propogation of $\mathfrak{G}$-equivariant geometric information, which implies that the test can be run for any number of iterations without loss of information. In geometric GNNs, $\mathfrak{G}$-equivariant information is propogated via summing features from multiple layers in fixed dimensional spaces, which may lead to distortion or loss of information from distant nodes.

Addresses R1.4

To study the practical implications of depth in propagating geometric information beyond local neighbourhoods, we consider $k$-chain geometric graphs which generalise the examples from Schütt et al. (2021). Each pair of $k$-chains consists of $k + 2$ nodes with $k$ nodes arranged in a line and differentiated by the orientation of the 2 end points. Thus, $k$-chain graphs are $(\lfloor \frac{k}{2} \rfloor + 1)$-hop distinguishable, and $(\lfloor \frac{k}{2} \rfloor + 1)$ GWL iterations are theoretically sufficient to distinguish them. In Table 3, we train $\mathfrak{G}$-equivariant and $\mathfrak{G}$-invariant GNNs with an increasing number of layers to distinguish $k$-chains. Despite the supposed simplicity of the task, especially for small chain lengths, we find that popular $\mathfrak{G}$-equivariant GNNs such as E-GNN and TFN may require more iterations that prescribed by GWL. Notably, as the length of the chain gets larger than $k = 4$, all $\mathfrak{G}$-equivariant GNNs tended to lose performance and required more than $(\lfloor \frac{k}{2} \rfloor + 1)$ iterations to solve the task. IGWL and $\mathfrak{G}$-invariant GNNs are unable to distinguish $k$-chains.

Table 3, together with Table 2, points to preliminary evidence of the *oversmoothing* and *oversquashing* phenomenon (Nt & Maehara, 2019; Alon & Yahav, 2021; Topping et al., 2022) for geometric GNNs. These issues are most evident for E-GNN, which uses a single vector feature to aggregate and propagate geometric information. This may have implications in modelling macromolecules where long-range interactions often play important roles. Studying both these issues are exciting avenues for future work towards building provably powerful, *practical* geometric GNNs.

Addresses R1.4

## D  PROOFS FOR WHAT GWL AND IGWL CAN DISTINGUISH

The following results are a consequence of the construction of GWL as well as the definitions of $k$-hop distinct and $k$-hop identical geometric graphs. Note that $k$-hop distinct geometric graphs are also $(k+1)$-hop distinct. Similarly, $k$-hop identical geometric graphs are also $(k-1)$-hop identical, but not necessarily $(k+1)$-hop distinct.

Given two distinct neighbourhoods $\mathcal{N}_1$ and $\mathcal{N}_2$, the $\mathfrak{G}$-orbits of the corresponding geometric multisets $\boldsymbol{g}_1$ and $\boldsymbol{g}_2$ are mutually exclusive, *i.e.* $\mathcal{O}_{\mathfrak{G}}(\boldsymbol{g}_1) \cap \mathcal{O}_{\mathfrak{G}}(\boldsymbol{g}_2) \equiv \emptyset$. By the properties of I-HASH this implies $c_1 \neq c_2$. Conversely, if $\mathcal{N}_1$ and $\mathcal{N}_2$ were identical up to group actions, their $\mathfrak{G}$-orbits would overlap, *i.e.* $\boldsymbol{g}_1 = \mathfrak{g}\, \boldsymbol{g}_2$ for some $\mathfrak{g} \in \mathfrak{G}$ and $\mathcal{O}_{\mathfrak{G}}(\boldsymbol{g}_1) = \mathcal{O}_{\mathfrak{G}}(\boldsymbol{g}_2) \Rightarrow c_1 = c_2$.

**Proposition 1.** *GWL can distinguish any $k$-hop distinct geometric graphs $\mathcal{G}_1$ and $\mathcal{G}_2$ where the underlying attributed graphs are isomorphic, and $k$ iterations are sufficient.*

*Proof of Proposition 1.* The $k$-th iteration of GWL identifies the $\mathfrak{G}$-orbit of the $k$-hop subgraph $\mathcal{N}_i^{(k)}$ at each node $i$ via the geometric multiset $\boldsymbol{g}_i^{(k)}$. $\mathcal{G}_1$ and $\mathcal{G}_2$ being $k$-hop distinct implies that there exists some bijection $b$ and some node $i \in \mathcal{V}_1, b(i) \in \mathcal{V}_2$ such that the corresponding $k$-hop subgraphs $\mathcal{N}_i^{(k)}$ and $\mathcal{N}_{b(i)}^{(k)}$ are distinct. Thus, the $\mathfrak{G}$-orbits of the corresponding geometric multisets $\boldsymbol{g}_i^{(k)}$ and $\boldsymbol{g}_{b(i)}^{(k)}$ are mutually exclusive, *i.e.* $\mathcal{O}_{\mathfrak{G}}(\boldsymbol{g}_i^{(k)}) \cap \mathcal{O}_{\mathfrak{G}}(\boldsymbol{g}_{b(i)}^{(k)}) \equiv \emptyset \Rightarrow c_i^{(k)} \neq c_{b(i)}^{(k)}$. Thus, $k$ iterations of GWL are sufficient to distinguish $\mathcal{G}_1$ and $\mathcal{G}_2$. $\qquad\square$

**Proposition 2.** *Up to $k$ iterations, GWL cannot distinguish any $k$-hop identical geometric graphs $\mathcal{G}_1$ and $\mathcal{G}_2$ where the underlying attributed graphs are isomorphic.*

*Proof of Proposition 2.* The $k$-th iteration of GWL identifies the $\mathfrak{G}$-orbit of the $k$-hop subgraph $\mathcal{N}_i^{(k)}$ at each node $i$ via the geometric multiset $\boldsymbol{g}_i^{(k)}$. $\mathcal{G}_1$ and $\mathcal{G}_2$ being $k$-hop identical implies that for all bijections $b$ and all nodes $i \in \mathcal{V}_1, b(i) \in \mathcal{V}_2$, the corresponding $k$-hop subgraphs $\mathcal{N}_i^{(k)}$ and $\mathcal{N}_{b(i)}^{(k)}$ are identical up to group actions. Thus, the $\mathfrak{G}$-orbits of the corresponding geometric multisets $\boldsymbol{g}_i^{(k)}$ and $\boldsymbol{g}_{b(i)}^{(k)}$ overlap, *i.e.* $\mathcal{O}_{\mathfrak{G}}(\boldsymbol{g}_i^{(k)}) = \mathcal{O}_{\mathfrak{G}}(\boldsymbol{g}_{b(i)}^{(k)}) \Rightarrow c_i^{(k)} = c_{b(i)}^{(k)}$. Thus, up to $k$ iterations of GWL cannot distinguish $\mathcal{G}_1$ and $\mathcal{G}_2$. $\qquad\square$

**Proposition 3.** *IGWL can distinguish any $1$-hop distinct geometric graphs $\mathcal{G}_1$ and $\mathcal{G}_2$ where the underlying attributed graphs are isomorphic, and 1 iteration is sufficient.*

*Proof of Proposition 3.* Each iteration of IGWL identifies the $\mathfrak{G}$-orbit of the $1$-hop local neighbourhood $\mathcal{N}_i^{(k=1)}$ at each node $i$. $\mathcal{G}_1$ and $\mathcal{G}_2$ being $1$-hop distinct implies that there exists some bijection $b$ and some node $i \in \mathcal{V}_1, b(i) \in \mathcal{V}_2$ such that the corresponding $1$-hop local neighbourhoods $\mathcal{N}_i^{(1)}$ and $\mathcal{N}_{b(i)}^{(1)}$ are distinct. Thus, the $\mathfrak{G}$-orbits of the corresponding geometric multisets $\boldsymbol{g}_i^{(1)}$ and $\boldsymbol{g}_{b(i)}^{(1)}$ are mutually exclusive, *i.e.* $\mathcal{O}_{\mathfrak{G}}(\boldsymbol{g}_i^{(1)}) \cap \mathcal{O}_{\mathfrak{G}}(\boldsymbol{g}_{b(i)}^{(1)}) \equiv \emptyset \Rightarrow c_i^{(1)} \neq c_{b(i)}^{(1)}$. Thus, 1 iteration of IGWL is sufficient to distinguish $\mathcal{G}_1$ and $\mathcal{G}_2$. $\qquad\square$

**Proposition 4.** *Any number of iterations of IGWL cannot distinguish any $1$-hop identical geometric graphs $\mathcal{G}_1$ and $\mathcal{G}_2$ where the underlying attributed graphs are isomorphic.*

*Proof of Proposition 4.* Each iteration of IGWL identifies the $\mathfrak{G}$-orbit of the $1$-hop local neighbourhood $\mathcal{N}_i^{(k=1)}$ at each node $i$, but cannot identify $\mathfrak{G}$-orbits beyond $1$-hop by the construction of IGWL as no geometric information is propagated. $\mathcal{G}_1$ and $\mathcal{G}_2$ being $1$-hop identical implies that for all bijections $b$ and all nodes $i \in \mathcal{V}_1, b(i) \in \mathcal{V}_2$, the corresponding $1$-hop local neighbourhoods $\mathcal{N}_i^{(k)}$ and $\mathcal{N}_{b(i)}^{(k)}$ are identical up to group actions. Thus, the $\mathfrak{G}$-orbits of the corresponding geometric multisets $\boldsymbol{g}_i^{(1)}$ and $\boldsymbol{g}_{b(i)}^{(1)}$ overlap, *i.e.* $\mathcal{O}_{\mathfrak{G}}(\boldsymbol{g}_i^{(1)}) = \mathcal{O}_{\mathfrak{G}}(\boldsymbol{g}_{b(i)}^{(1)}) \Rightarrow c_i^{(k)} = c_{b(i)}^{(k)}$. Thus, any number of IGWL iterations cannot distinguish $\mathcal{G}_1$ and $\mathcal{G}_2$. $\qquad\square$

**Proposition 5.** *Assuming geometric graphs are constructed from point clouds using radial cutoffs, GWL can distinguish any geometric graphs $\mathcal{G}_1$ and $\mathcal{G}_2$ where the underlying attributed graphs are non-isomorphic. At most $k_{Max}$ iterations are sufficient, where $k_{Max}$ is the maximum graph diameter among $\mathcal{G}_1$ and $\mathcal{G}_2$.*

*Proof of Proposition 5.* We assume that a geometric graph $\mathcal{G} = (\boldsymbol{A}, \boldsymbol{S}, \vec{\boldsymbol{V}}, \vec{\boldsymbol{X}})$ is constructed from a point cloud $(\boldsymbol{S}, \vec{\boldsymbol{V}}, \vec{\boldsymbol{X}})$ using a predetermined radial cutoff $r$. Thus, the adjacency matrix is defined as $a_{ij} = 1$ if $\|\vec{\boldsymbol{x}}_i - \vec{\boldsymbol{x}}_j\|_2 \leq r$, or 0 otherwise, for all $a_{ij} \in \boldsymbol{A}$. Such construction procedures are conventional for geometric graphs in biochemistry and material science.

Given geometric graphs $\mathcal{G}_1$ and $\mathcal{G}_2$ where the underlying attributed graphs are non-isomorphic, identify $k_{\text{Max}}$ the maximum of the graph diameters of $\mathcal{G}_1$ and $\mathcal{G}_2$, and chose any arbitrary nodes $i \in \mathcal{V}_1, j \in \mathcal{V}_2$. We can define the $k_{\text{Max}}$-hop subgraphs $\mathcal{N}_i^{(k_{\text{Max}})}$ and $\mathcal{N}_j^{(k_{\text{Max}})}$ at $i$ and $j$, respectively. Thus, $\mathcal{N}_i^{(k_{\text{Max}})} = \mathcal{V}_1$ for all $i \in \mathcal{V}_1$, and $\mathcal{N}_j^{(k_{\text{Max}})} = \mathcal{V}_2$ for all $j \in \mathcal{V}_2$. Due to the assumed construction procedure of geometric graphs, $\mathcal{N}_i^{(k_{\text{Max}})}$ and $\mathcal{N}_j^{(k_{\text{Max}})}$ must be distinct. Otherwise, if $\mathcal{N}_i^{(k_{\text{Max}})}$ and $\mathcal{N}_j^{(k_{\text{Max}})}$ were identical up to group actions, the sets $(\boldsymbol{S}_1, \vec{\boldsymbol{V}}_1, \vec{\boldsymbol{X}}_1)$ and $(\boldsymbol{S}_2, \vec{\boldsymbol{V}}_2, \vec{\boldsymbol{X}}_2)$ would have yielded isomorphic graphs.

The $k_{\text{Max}}$-th iteration of GWL identifies the $\mathfrak{G}$-orbit of the $k_{\text{Max}}$-hop subgraph $\mathcal{N}_i^{(k_{\text{Max}})}$ at each node $i$ via the geometric multiset $\boldsymbol{g}_i^{(k_{\text{Max}})}$. As $\mathcal{N}_i^{(k_{\text{Max}})}$ and $\mathcal{N}_j^{(k_{\text{Max}})}$ are distinct for any arbitrary nodes $i \in \mathcal{V}_1, j \in \mathcal{V}_2$, the $\mathfrak{G}$-orbits of the corresponding geometric multisets $\boldsymbol{g}_i^{(k_{\text{Max}})}$ and $\boldsymbol{g}_j^{(k_{\text{Max}})}$ are mutually exclusive, *i.e.* $\mathcal{O}_\mathfrak{G}(\boldsymbol{g}_i^{(k_{\text{Max}})}) \cap \mathcal{O}_\mathfrak{G}(\boldsymbol{g}_j^{(k_{\text{Max}})}) \equiv \emptyset \Rightarrow c_i^{(k_{\text{Max}})} \neq c_j^{(k_{\text{Max}})}$. Thus, $k_{\text{Max}}$ iterations of GWL are sufficient to distinguish $\mathcal{G}_1$ and $\mathcal{G}_2$. $\qquad\square$

**Theorem 6.** *GWL is strictly more powerful than IGWL.*

*Proof of Theorem 6.* Firstly, we can show that the GWL class contains IGWL if GWL can learn the identity when updating $\boldsymbol{g}_i$ for all $i \in \mathcal{V}$, *i.e.* $\boldsymbol{g}_i^{(t)} = \boldsymbol{g}_i^{(t-1)} = \boldsymbol{g}_i^{(0)} \equiv (\boldsymbol{s}_i, \vec{\boldsymbol{v}}_i)$. Thus, GWL is at least as powerful as IGWL, which does not update $\boldsymbol{g}_i$.

Secondly, to show that GWL is strictly more powerful than IGWL, it suffices to show that there exist a pair of geometric graphs that can be distinguished by GWL but not by IGWL. We may consider any $k$-hop distinct geometric graphs for $k > 1$, where the underlying attributed graphs are isomorphic. Proposition 1 states that GWL can distinguish any such graphs, while Proposition 4 states that IGWL cannot distinguish them. An example is the pair of graphs in Figures 1 and 2. $\quad\square$

**Proposition 7.** *IGWL has the same expressive power as GWL for fully connected geometric graphs.*

*Proof of Proposition 7.* We will prove by contradiction. Assume that there exist a pair of fully connected geometric graphs $\mathcal{G}_1$ and $\mathcal{G}_2$ which GWL can distinguish, but IGWL cannot.

If the underlying attributed graphs of $\mathcal{G}_1$ and $\mathcal{G}_2$ are isomorphic, by Proposition 1 and Proposition 4, $\mathcal{G}_1$ and $\mathcal{G}_2$ are 1-hop identical but $k$-hop distinct for some $k > 1$. For all bijections $b$ and all nodes $i \in \mathcal{V}_1, b(i) \in \mathcal{V}_2$, the local neighbourhoods $\mathcal{N}_i^{(1)}$ and $\mathcal{N}_{b(i)}^{(1)}$ are identical up to group actions, and $\mathcal{O}_\mathfrak{G}(\boldsymbol{g}_i^{(1)}) = \mathcal{O}_\mathfrak{G}(\boldsymbol{g}_{b(i)}^{(1)}) \Rightarrow c_i^{(1)} = c_{b(i)}^{(1)}$. Additionally, there exists some bijection $b$ and some nodes $i \in \mathcal{V}_1, b(i) \in \mathcal{V}_2$ such that the $k$-hop subgraphs $\mathcal{N}_i^{(k)}$ and $\mathcal{N}_{b(i)}^{(k)}$ are distinct, and $\mathcal{O}_\mathfrak{G}(\boldsymbol{g}_i^{(k)}) \cap \mathcal{O}_\mathfrak{G}(\boldsymbol{g}_{b(i)}^{(k)}) \equiv \emptyset \Rightarrow c_i^{(k)} \neq c_{b(i)}^{(k)}$. However, as $\mathcal{G}_1$ and $\mathcal{G}_2$ are fully connected, for any $k$, $\mathcal{N}_i^{(1)} = \mathcal{N}_i^{(k)}$ and $\mathcal{N}_{b(i)}^{(1)} = \mathcal{N}_{b(i)}^{(k)}$ are identical up to group actions. Thus, $\mathcal{O}_\mathfrak{G}(\boldsymbol{g}_i^{(1)}) = \mathcal{O}_\mathfrak{G}(\boldsymbol{g}_i^{(k)}) = \mathcal{O}_\mathfrak{G}(\boldsymbol{g}_{b(i)}^{(1)}) = \mathcal{O}_\mathfrak{G}(\boldsymbol{g}_{b(i)}^{(k)}) \Rightarrow c_i^{(1)} = c_i^{(k)} = c_{b(i)}^{(1)} = c_{b(i)}^{(k)}$. This is a contradiction.

If $\mathcal{G}_1$ and $\mathcal{G}_2$ are non-isomorphic and fully connected, for any arbitrary $i \in \mathcal{V}_1, j \in \mathcal{V}_2$ and any $k$-hop neighbourhood, we know that $\mathcal{N}_i^{(1)} = \mathcal{N}_i^{(k)}$ and $\mathcal{N}_j^{(1)} = \mathcal{N}_j^{(k)}$. Thus, a single iteration of GWL and IGWL identify the same $\mathfrak{G}$-orbits and assign the same node colours, *i.e.* $\mathcal{O}_\mathfrak{G}(\boldsymbol{g}_i^{(1)}) = \mathcal{O}_\mathfrak{G}(\boldsymbol{g}_i^{(k)}) \Rightarrow c_i^{(1)} = c_i^{(k)}$ and $\mathcal{O}_\mathfrak{G}(\boldsymbol{g}_j^{(1)}) = \mathcal{O}_\mathfrak{G}(\boldsymbol{g}_j^{(k)}) \Rightarrow c_j^{(1)} = c_j^{(k)}$. This is a contradiction. $\quad\square$

# E    PROOFS FOR EQUIVALENCE BETWEEN GWL AND GEOMETRIC GNNS

Our proofs adapt the techniques used in (Xu et al., 2019; Morris et al., 2019) for connecting WL with GNNs. Note that we omit including the relative position vectors $\vec{x}_{ij}$ in GWL and geometric GNN updates for brevity, as relative positions vectors can be merged into the vector features.

**Theorem 8.** *Any pair of geometric graphs distinguishable by a $\mathfrak{G}$-equivariant GNN is also distinguishable by GWL.*

***Proof of Theorem 8.*** Consider two geometric graphs $\mathcal{G}$ and $\mathcal{H}$. The theorem implies that if the GNN graph-level readout outputs $f(\mathcal{G}) \neq f(\mathcal{H})$, then the GWL test will always determine $\mathcal{G}$ and $\mathcal{H}$ to be non-isomorphic, *i.e.* $\mathcal{G} \neq \mathcal{H}$.

We will prove by contradiction. Suppose after $T$ iterations, a GNN graph-level readout outputs $f(\mathcal{G}) \neq f(\mathcal{H})$, but the GWL test cannot decide $\mathcal{G}$ and $\mathcal{H}$ are non-isomorphic, *i.e.* $\mathcal{G}$ and $\mathcal{H}$ always have the same collection of node colours for iterations 0 to $T$. Thus, for iteration $t$ and $t + 1$ for any $t = 0 \ldots T - 1$, $\mathcal{G}$ and $\mathcal{H}$ have the same collection of node colours $\{c_i^{(t)}\}$ as well as the same collection of neighbourhood geometric multisets $\left\{ (c_i^{(t)}, \boldsymbol{g}_i^{(t)}), \ \{\!\!\{ (c_j^{(t)}, \boldsymbol{g}_j^{(t)}) \mid j \in \mathcal{N}_i \}\!\!\} \right\}$ up to group actions. Otherwise, the GWL test would have produced different node colours at iteration $t + 1$ for $\mathcal{G}$ and $\mathcal{H}$ as different geometric multisets get unique new colours.

We will show that on the same graph for nodes $i$ and $k$, if $(c_i^{(t)}, \boldsymbol{g}_i^{(t)}) = (c_k^{(t)}, \mathfrak{g} \cdot \boldsymbol{g}_k^{(t)})$, we always have GNN features $(\boldsymbol{s}_i^{(t)}, \vec{\boldsymbol{v}}_i^{(t)}) = (\boldsymbol{s}_k^{(t)}, \boldsymbol{Q}_\mathfrak{g} \vec{\boldsymbol{v}}_k^{(t)})$ for any iteration $t$. This holds for $t = 0$ because GWL and the GNN start with the same initialisation. Suppose this holds for iteration $t$. At iteration $t + 1$, if for any $i$ and $k$, $(c_i^{(t+1)}, \boldsymbol{g}_i^{(t+1)}) = (c_k^{(t+1)}, \mathfrak{g} \cdot \boldsymbol{g}_k^{(t+1)})$, then:

$$\left\{ (c_i^{(t)}, \boldsymbol{g}_i^{(t)}), \ \{\!\!\{ (c_j^{(t)}, \boldsymbol{g}_j^{(t)}) \mid j \in \mathcal{N}_i \}\!\!\} \right\} = \left\{ (c_k^{(t)}, \mathfrak{g} \cdot \boldsymbol{g}_k^{(t)}), \ \{\!\!\{ (c_j^{(t)}, \mathfrak{g} \cdot \boldsymbol{g}_j^{(t)}) \mid j \in \mathcal{N}_k \}\!\!\} \right\} \quad (20)$$

By our assumption on iteration $t$,

$$\left\{ (\boldsymbol{s}_i^{(t)}, \vec{\boldsymbol{v}}_i^{(t)}), \ \{\!\!\{ (\boldsymbol{s}_j^{(t)}, \vec{\boldsymbol{v}}_j^{(t)}) \mid j \in \mathcal{N}_i \}\!\!\} \right\} = \left\{ (\boldsymbol{s}_k^{(t)}, \boldsymbol{Q}_\mathfrak{g} \vec{\boldsymbol{v}}_k^{(t)}), \ \{\!\!\{ (\boldsymbol{s}_j^{(t)}, \boldsymbol{Q}_\mathfrak{g} \vec{\boldsymbol{v}}_j^{(t)}) \mid j \in \mathcal{N}_k \}\!\!\} \right\} \quad (21)$$

As the same aggregate and update operations are applied at each node within the GNN, the same inputs, *i.e.* neighbourhood features, are mapped to the same output. Thus, $(\boldsymbol{s}_i^{(t+1)}, \vec{\boldsymbol{v}}_i^{(t+1)}) = (\boldsymbol{s}_k^{(t+1)}, \boldsymbol{Q}_\mathfrak{g} \vec{\boldsymbol{v}}_k^{(t+1)})$. By induction, if $(c_i^{(t)}, \boldsymbol{g}_i^{(t)}) = (c_k^{(t)}, \mathfrak{g} \cdot \boldsymbol{g}_k^{(t)})$, we always have GNN node features $(\boldsymbol{s}_i^{(t)}, \vec{\boldsymbol{v}}_i^{(t)}) = (\boldsymbol{s}_k^{(t)}, \boldsymbol{Q}_\mathfrak{g} \vec{\boldsymbol{v}}_k^{(t)})$ for any iteration $t$. This creates valid mappings $\phi_s, \phi_v$ such that $\boldsymbol{s}_i^{(t)} = \phi_s(c_i^{(t)})$ and $\vec{\boldsymbol{v}}_i^{(t)} = \phi_v(c_i^{(t)}, \boldsymbol{g}_i^{(t)})$ for any $i \in \mathcal{V}$.

Thus, if $\mathcal{G}$ and $\mathcal{H}$ have the same collection of node colours and geometric multisets, then $\mathcal{G}$ and $\mathcal{H}$ also have the same collection of GNN neighbourhood features

$$\left\{ (\boldsymbol{s}_i^{(t)}, \vec{\boldsymbol{v}}_i^{(t)}), \ \{\!\!\{ (\boldsymbol{s}_j^{(t)}, \vec{\boldsymbol{v}}_j^{(t)}) \mid j \in \mathcal{N}_i \}\!\!\} \right\} = \left\{ (\phi_s(c_i^{(t)}), \phi_v(c_i^{(t)}, \boldsymbol{g}_i^{(t)})), \ \{\!\!\{ (\phi_s(c_j^{(t)}), \phi_v(c_i^{(t)}, \boldsymbol{g}_i^{(t)})) \mid j \in \mathcal{N}_i \}\!\!\} \right\}$$

Thus, the GNN will output the same collection of node scalar features $\{\boldsymbol{s}_i^{(T)}\}$ for $\mathcal{G}$ and $\mathcal{H}$ and the permutation-invariant graph-level readout will output $f(\mathcal{G}) = f(\mathcal{H})$. This is a contradiction.    $\square$

Similarly, $\mathfrak{G}$-invariant GNNs of the form in Equation 4 can be at most as powerful as IGWL.

**Theorem 24.** *Any pair of geometric graphs distinguishable by a $\mathfrak{G}$-invariant GNN is also distinguishable by IGWL.*

*Proof.* The proof follows similarly to the proof for Theorem 8.    $\square$

**Proposition 9.** *$\mathfrak{G}$-equivariant GNNs have the same expressive power as GWL if the following conditions hold: (1) The aggregation $\mathrm{AGG}$ is an injective, $\mathfrak{G}$-equivariant multiset function. (2) The scalar part of the update $\mathrm{UPD}_s$ is a $\mathfrak{G}$-orbit injective, $\mathfrak{G}$-invariant multiset function. (3) The vector part of the update $\mathrm{UPD}_v$ is an injective, $\mathfrak{G}$-equivariant multiset function. (4) The graph-level readout $f$ is an injective multiset function.*

**Proof of Theorem 9**. Consider a GNN where the conditions hold. We will show that, with a sufficient number of iterations $t$, the output of this GNN is equivalent to GWL, *i.e.* $\boldsymbol{s}^{(t)} \equiv c^{(t)}$.

Let $\mathcal{G}$ and $\mathcal{H}$ be any geometric graphs which the GWL test decides as non-isomorphic at iteration $T$. Because the graph-level readout function is injective, *i.e.* it maps distinct multiset of node scalar features into unique embeddings, it suffices to show that the GNN's neighbourhood aggregation process, with sufficient iterations, embeds $\mathcal{G}$ and $\mathcal{H}$ into different multisets of node features.

For this proof, we replace $\mathfrak{G}$-orbit injective functions with injective functions over the equivalence class generated by the actions of $\mathfrak{G}$. Thus, all elements belonging to the same $\mathfrak{G}$-orbit will first be mapped to the same representative of the equivalence class, denoted by the square brackets $[\dots]$, followed by an injective map. The result is $\mathfrak{G}$-orbit injective.

Let us assume the GNN updates node scalar and vector features as:

$$\boldsymbol{s}_i^{(t)} = \mathrm{UPD}_s\left(\left[(\boldsymbol{s}_i^{(t-1)}, \vec{\boldsymbol{v}}_i^{(t-1)}),\ \mathrm{AGG}\left(\{\!\{(\boldsymbol{s}_i^{(t-1)}, \boldsymbol{s}_j^{(t-1)}, \vec{\boldsymbol{v}}_i^{(t-1)}, \vec{\boldsymbol{v}}_j^{(t-1)}) \mid j \in \mathcal{N}_i\}\!\}\right)\right]\right) \quad (22)$$

$$\vec{\boldsymbol{v}}_i^{(t)} = \mathrm{UPD}_v\left((\boldsymbol{s}_i^{(t-1)}, \vec{\boldsymbol{v}}_i^{(t-1)}),\ \mathrm{AGG}\left(\{\!\{(\boldsymbol{s}_i^{(t-1)}, \boldsymbol{s}_j^{(t-1)}, \vec{\boldsymbol{v}}_i^{(t-1)}, \vec{\boldsymbol{v}}_j^{(t-1)}) \mid j \in \mathcal{N}_i\}\!\}\right)\right) \quad (23)$$

with the aggregation function $\mathrm{AGG}$ being $\mathfrak{G}$-equivariant and injective, the scalar update function $\mathrm{UPD}_s$ being $\mathfrak{G}$-invariant and injective, and the vector update function $\mathrm{UPD}_v$ being $\mathfrak{G}$-equivariant and injective.

The GWL test updates the node colour $c_i^{(t)}$ and geometric multiset $\boldsymbol{g}_i^{(t)}$ as:

$$c_i^{(t)} = h_s\left(\left[(c_i^{(t-1)}, \boldsymbol{g}_i^{(t-1)}),\ \{\!\{(c_j^{(t-1)}, \boldsymbol{g}_j^{(t-1)}) \mid j \in \mathcal{N}_i\}\!\}\right]\right), \quad (24)$$

$$\boldsymbol{g}_i^{(t)} = h_v\left((c_i^{(t-1)}, \boldsymbol{g}_i^{(t-1)}),\ \{\!\{(c_j^{(t-1)}, \boldsymbol{g}_j^{(t-1)}) \mid j \in \mathcal{N}_i\}\!\}\right), \quad (25)$$

where $h_s$ is a $\mathfrak{G}$-invariant and injective map, and $h_v$ is a $\mathfrak{G}$-equivariant and injective operation (e.g. in equation 6, expanding the geometric multiset by copying).

We will show by induction that at any iteration $t$, there always exist injective functions $\varphi_s$ and $\varphi_v$ such that $\boldsymbol{s}_i^{(t)} = \varphi_s(c_i^{(t)})$ and $\vec{\boldsymbol{v}}_i^{(t)} = \varphi_v(c_i^{(t)}, \boldsymbol{g}_i^{(t)})$. This holds for $t = 0$ because the initial node features are the same for GWL and GNN, $c_i^{(0)} \equiv \boldsymbol{s}_i^{(0)}$ and $\boldsymbol{g}_i^{(0)} \equiv (\boldsymbol{s}_i^{(0)}, \vec{\boldsymbol{v}}_i^{(0)})$ for all $i \in \mathcal{V}(\mathcal{G}), \mathcal{V}(\mathcal{H})$. Suppose this holds for iteration $t$. At iteration $t+1$, substituting $\boldsymbol{s}_i^{(t)}$ with $\varphi_s(c_i^{(t)})$, and $\vec{\boldsymbol{v}}_i^{(t)}$ with $\varphi_v(c_i^{(t)}, \boldsymbol{g}_i^{(t)})$ gives us

$$\boldsymbol{s}_i^{(t+1)} = \mathrm{UPD}_s\left(\left[(\varphi_s(c_i^{(t)}), \varphi_v(c_i^{(t)}, \boldsymbol{g}_i^{(t)})),\ \mathrm{AGG}\left(\{\!\{(\varphi_s(c_i^{(t)}), \varphi_s(c_j^{(t)}), \varphi_v(c_i^{(t)}, \boldsymbol{g}_i^{(t)}), \varphi_v(c_j^{(t)}, \boldsymbol{g}_j^{(t)})) \mid j \in \mathcal{N}_i\}\!\}\right)\right]\right)$$

$$\vec{\boldsymbol{v}}_i^{(t+1)} = \mathrm{UPD}_v\left((\varphi_s(c_i^{(t)}), \varphi_v(c_i^{(t)}, \boldsymbol{g}_i^{(t)})),\ \mathrm{AGG}\left(\{\!\{(\varphi_s(c_i^{(t)}), \varphi_s(c_j^{(t)}), \varphi_v(c_i^{(t)}, \boldsymbol{g}_i^{(t)}), \varphi_v(c_j^{(t)}, \boldsymbol{g}_j^{(t)})) \mid j \in \mathcal{N}_i\}\!\}\right)\right)$$

The composition of multiple injective functions is injective. Therefore, there exist some injective functions $g_s$ and $g_v$ such that:

$$\boldsymbol{s}_i^{(t+1)} = g_s\left(\left[(c_i^{(t)}, \boldsymbol{g}_i^{(t)}),\ \{\!\{(c_j^{(t)}, \boldsymbol{g}_j^{(t)}) \mid j \in \mathcal{N}_i\}\!\}\right]\right), \quad (26)$$

$$\vec{\boldsymbol{v}}_i^{(t+1)} = g_v\left((c_i^{(t)}, \boldsymbol{g}_i^{(t)}),\ \{\!\{(c_j^{(t)}, \boldsymbol{g}_j^{(t)}) \mid j \in \mathcal{N}_i\}\!\}\right), \quad (27)$$

We can then consider:

$$\boldsymbol{s}_i^{(t+1)} = g_s \circ h_s^{-1}\, h_s\left(\left[(c_i^{(t)}, \boldsymbol{g}_i^{(t)}),\ \{\!\{(c_j^{(t)}, \boldsymbol{g}_j^{(t)}) \mid j \in \mathcal{N}_i\}\!\}\right]\right), \quad (28)$$

$$\vec{\boldsymbol{v}}_i^{(t+1)} = g_v \circ h_v^{-1}\, h_v\left((c_i^{(t)}, \boldsymbol{g}_i^{(t)}),\ \{\!\{(c_j^{(t)}, \boldsymbol{g}_j^{(t)}) \mid j \in \mathcal{N}_i\}\!\}\right), \quad (29)$$

Then, we can denote $\varphi_s = g_s \circ h_s^{-1}$ and $\varphi_v = g_v \circ h_v^{-1}$ as injective functions because the composition of injective functions is injective. Hence, for any iteration $t+1$, there exist injective functions $\varphi_s$ and $\varphi_v$ such that $\boldsymbol{s}_i^{(t+1)} = \varphi_s\left(c_i^{(t+1)}\right)$ and $\vec{\boldsymbol{v}}_i^{(t+1)} = \varphi_v\left(c_i^{(t+1)}, \boldsymbol{g}_i^{(t+1)}\right)$.

At the $T$-th iteration, the GWL test decides that $\mathcal{G}$ and $\mathcal{H}$ are non-isomorphic, which means the multisets of node colours $\{c_i^{(T)}\}$ are different for $\mathcal{G}$ and $\mathcal{H}$. The GNN's node scalar features $\{\boldsymbol{s}_i^{(T)}\} = \{\varphi_s(c_i^{(T)})\}$ must also be different for $\mathcal{G}$ and $\mathcal{H}$ because of the injectivity of $\varphi_s$.

$\square$

A weaker set of conditions is sufficient for a $\mathfrak{G}$-invariant GNN to be at least as expressive as IGWL.

**Proposition 25.** *$\mathfrak{G}$-invariant GNNs have the same expressive power as IGWL if the following conditions hold: (1) The aggregation $\psi$ and update $\phi$ are $\mathfrak{G}$-orbit injective, $\mathfrak{G}$-invariant multiset functions. (2) The graph-level readout $f$ is an injective multiset function.*

*Proof.* The proof follows similarly to the proof for Theorem 9. $\square$

## F GEOMETRIC GNN DESIGN SPACE PROOFS

**Proposition 10.** *IGWL and $\mathfrak{G}$-invariant GNNs cannot decide several geometric graph properties: (1) perimeter, surface area, and volume of the bounding box/sphere enclosing the geometric graph; (2) distance from the centroid or centre of mass; and (3) dihedral angles.*

*Proof of Proposition 10.* Following Garg et al. (2020), we say that a class of models *decides* a geometric graph property if there exists a model belonging to this class such that for any two geometric graphs that differ in the property, the model is able to distinguish the two geometric graphs.

In Figure 4 we provide an example of two geometric graphs that demonstrate the proposition. $\mathcal{G}_1$ and $\mathcal{G}_2$ differ in the following geometric graph properties:

- Perimeter, surface area, and volume of the bounding box enclosing the geometric graph[1]: (32 units, 40 units$^2$, 16 units$^3$) vs. (28 units, 24 units$^2$, 8 units$^3$).

- Multiset of distances from the centroid or centre of mass: $\{0.00, 1.00, 1.00, 2.45, 2.45\}$ vs. $\{0.40, 1.08, 1.08, 2.32, 2.32\}$.

- Dihedral angles: $\angle(ljkm) = \frac{(\vec{\boldsymbol{x}}_{jk} \times \vec{\boldsymbol{x}}_{lj}) \cdot (\vec{\boldsymbol{x}}_{jk} \times \vec{\boldsymbol{x}}_{mk})}{|\vec{\boldsymbol{x}}_{jk} \times \vec{\boldsymbol{x}}_{lj}||\vec{\boldsymbol{x}}_{jk} \times \vec{\boldsymbol{x}}_{mk}|}$ are clearly different for the two graphs.

However, according to Proposition 4 and Theorem 24, both IGWL and $\mathfrak{G}$-invariant GNNs cannot distinguish these two geometric graphs, and therefore, cannot decide all these properties.

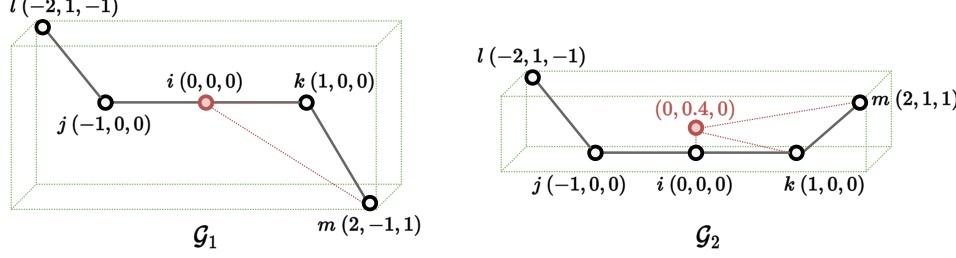

Figure 4: Two geometric graphs for which IGWL and $\mathfrak{G}$-invariant GNNs cannot distinguish their perimeter, surface area, volume of the bounding box/sphere, distance from the centroid, and dihedral angles. The centroid is denoted by a red point and distances from it are denoted by dotted red lines. The bounding box enclosing the geometric graph is denoted by the dotted green lines.

We can also show this by constructing geometric computation trees for any number of IGWL or $\mathfrak{G}$-invariant GNN iterations, as illustrated in Figure 3. We observe that the geometric computation trees of any pair of isomorphic nodes are identical, as all 1-hop neighbourhoods are computationally identical. Therefore, the set of node colours or node scalar features will also be identical, which implies that $\mathcal{G}_1$ and $\mathcal{G}_2$ cannot be distinguished. $\square$

**Proposition 11.** *I-HASH$_{(m)}$ is $\mathfrak{G}$-orbit injective for $m = max(\{|\mathcal{N}_i| \mid i \in \mathcal{V}\})$, the maximum cardinality of all local neighbourhoods $\mathcal{N}_i$ in a given dataset.*

---

[1]The same result applies for the bounding sphere, not shown in the figure.

*Proof of Proposition 11.* As $m$ is the maximum cardinality of all local neighbourhoods $\mathcal{N}_i$ under consideration, any distinct neighbourhoods $\mathcal{N}_1$ and $\mathcal{N}_2$ must have distinct multisets of $m$-body scalars. As I-HASH$_{(m)}$ computes scalars involving up to $m$ nodes, it will be able to distinguish any such $\mathcal{N}_1$ and $\mathcal{N}_2$. Thus, I-HASH$_{(m)}$ is $\mathfrak{G}$-orbit injective. $\qquad\square$

**Proposition 12.** *IGWL$_{(k)}$ is at least as powerful as IGWL$_{(k-1)}$. For $k \leq 5$, IGWL$_{(k)}$ is strictly more powerful than IGWL$_{(k-1)}$.*

*Proof of Proposition 12.* By construction, I-HASH$_{(k)}$ computes $\mathfrak{G}$-invariant scalars from all possible tuples of up to $k$ nodes formed by the elements of a neighbourhood and the central node. Thus, the I-HASH$_{(k)}$ class contains I-HASH$_{(k-1)}$, and I-HASH$_{(k)}$ is at least as powerful as I-HASH$_{(k-1)}$. Thus, the corresponding test IGWL$_{(k)}$ is at least as powerful as IGWL$_{(k-1)}$.

Secondly, to show that IGWL$_{(k)}$ is strictly more powerful than IGWL$_{(k-1)}$ for $k \leq 5$, it suffices to show that there exist a pair of geometric neighbourhoods that can be distinguished by IGWL$_{(k)}$ but not by IGWL$_{(k-1)}$:

- For $k = 3$ and $\mathfrak{G} = O(3)$ or $SO(3)$, for the local neighbourhood from Figure 1 in Schütt et al. (2021), two configurations with different angles between the neighbouring nodes can be distinguished by IGWL$_{(3)}$ but not by IGWL$_{(2)}$.

- For $k = 4$ and $\mathfrak{G} = O(3)$ or $SO(3)$, the pair of local neighbourhoods from Figure 1 in Pozdnyakov et al. (2020) can be distinguished by IGWL$_{(4)}$ but not by IGWL$_{(3)}$.

- For $k = 5$ and $\mathfrak{G} = O(3)$, the pair of local neighbourhoods from Figure 2(e) in Pozdnyakov et al. (2020) can be distinguished by IGWL$_{(5)}$ but not by IGWL$_{(4)}$.

- For $k = 5$ and $\mathfrak{G} = SO(3)$, the pair of local neighbourhoods from Figure 2(f) in Pozdnyakov et al. (2020) can be distinguished by IGWL$_{(5)}$ but not by IGWL$_{(4)}$.

$\qquad\square$

**Proposition 13.** *Let $\mathcal{G}_1 = (\boldsymbol{A}_1, \boldsymbol{S}_1, \vec{\boldsymbol{X}}_1)$ and $\mathcal{G}_2 = (\boldsymbol{A}_2, \boldsymbol{S}_2, \vec{\boldsymbol{X}}_2)$ be two geometric graphs with the property that all edges have equal length. Then, IGWL$_{(2)}$ distinguishes the two graphs if and only if WL can distinguish the attributed graphs $(\boldsymbol{A}_1, \boldsymbol{S}_1)$ and $(\boldsymbol{A}_1, \boldsymbol{S}_1)$.*

*Proof of Proposition 13.* Let $c$ and $k$ the colours produced by IGWL$_{(2)}$ and WL, respectively, and let $i$ and $j$ be two nodes belonging to any two graphs like in the statement of the result. We prove the statement inductively.

Clearly, $c_i^{(0)} = k_i^{(0)}$ for all nodes $i$ and $c_i^{(0)} = c_j^{(0)}$ if and only if $k_i^{(0)} = k_j^{(0)}$. Now, assume that the statement holds for iteration $t$. That is $c_i^{(t)} = c_j^{(t)}$ if and only if $k_i^{(t)} = k_j^{(t)}$ holds for all $i$. Note that $c_i^{(t+1)} = c_j^{(t+1)}$ if and only if $c_i^{(t)} = c_j^{(t)}$ and $\{\!\!\{(c_p^{(t)}, \|\vec{\boldsymbol{x}}_{ip}\|) \mid p \in \mathcal{N}_i\}\!\!\} = \{\!\!\{(c_p^{(t)}, \|\vec{\boldsymbol{x}}_{jp}\|) \mid p \in \mathcal{N}_j\}\!\!\}$, since the norm of the relative vectors is the only injective invariant that IGWL$_{(2)}$ can compute (up to a scaling). Since all the norms are equal, by the induction hypothesis, this is equivalent to $k_i^{(t)} = k_j^{(t)}$ and $\{\!\!\{k_p^{(t)} \mid p \in \mathcal{N}_i\}\!\!\} = \{\!\!\{k^{(t)} \mid p \in \mathcal{N}_j\}\!\!\}$. Therefore, this is equivalent to $k_i^{(t+1)} = k_j^{(t+1)}$ $\qquad\square$

# G  UNIVERSALITY AND DISCRIMINATION PROOFS

## G.1  EQUIVALENCE BETWEEN UNIVERSALITY AND DISCRIMINATION

The results in this subsection use the proofs from Chen et al. (2019) with minor adaptations.

**Theorem 16.** *If $\mathcal{C}$ is universally approximating over $Y$, then $\mathcal{C}$ is also pairwise $Y_{\mathfrak{G}}$ discriminating.*

*Proof.* Given any $y \in Y$, we can construct the $\mathfrak{G}$-invariant function over $X$, $\delta_y(x) = 0$ if $y \simeq x$ and 1 otherwise. Therefore, $\delta_y$ can be approximated with some $\epsilon < 0.5$ over $Y$ by some function $h \in \mathcal{C}$. Hence, $h(y) \neq h(y')$ for any $y, y' \in Y$ and $\mathcal{C}$ is pairwise $Y_{\mathfrak{G}}$ discriminating. $\qquad\square$

The following two Lemmas follow from Chen et al. (2019) with minor adaptations.

**Lemma 26.** *If $\mathcal{C}$ is pairwise $Y_{\mathfrak{G}}$ discriminating, then for all $y \in Y$, there exists a function $\delta_y \in \mathcal{C}^{+1}$ such that for all $y'$, $\delta_y(y') = 0$ if and only if $y \simeq y'$.*

*Proof.* For any $y_1, y_2 \in Y$ such that $y_1 \not\simeq y_2$, let $\delta_{y_1,y_2}$ be the function that distinguishes $y_1, y_2$. That is $\delta_{y_1,y_2}(y_1) \neq \delta_{y_1,y_2}(y_2)$. Then, we can define a function $\overline{\delta}_{y,y'} \in \mathcal{C}$:

$$\overline{\delta}_{y,y'}(x) = |\delta_{y,y'}(x) - \delta_{y,y'}(y)| \rightarrow \begin{cases} = 0 & \text{if } x \simeq y \\ > 0 & \text{if } x \simeq y' \\ \geq 0 & \text{otherwise} \end{cases} \tag{30}$$

This function is already similar to the $\delta_y$ function whose existence we want to prove. To obtain a function that is strictly positive over all the $x \in Y$ with $x \not\simeq y$, we can construct $\delta_y$ as a sum over all the $\overline{\delta}_{y,y'}$:

$$\delta_y(x) = \sum_{y' \in Y, y' \not\simeq y} \overline{\delta}_{y,y'}(x) \rightarrow \begin{cases} = 0 & \text{if } x \simeq y \\ > 0 & \text{if } x \not\simeq y \text{ and } x \in \mathcal{O}_{\mathfrak{G}}(Y) \supseteq Y \\ \geq 0 & \text{otherwise} \end{cases} \tag{31}$$

Given the finite set of functions $\{\delta_{y,y'}\}$, notice that

$$\overline{\delta}_{y,y'}(x) = \text{ReLU}\big(\delta_{y,y'}(x) - \delta_{y,y'}(y)\big) + \text{ReLU}\big(\delta_{y,y'}(y) - \delta_{y,y'}(x)\big).$$

Then $\delta_y$ is obtained by summing all these functions over $y' \in Y$ with $y' \not\simeq y$, so $\delta_y \in \mathcal{C}^{+1}$. $\qquad \square$

**Lemma 27.** *Let $\mathcal{C}$ is a class of $\mathfrak{G}$-invariant functions from $X \rightarrow \mathbb{R}$ such that for any $y, y' \in Y \subseteq X$, where $Y$ is finite, there is a $\delta_y \in \mathcal{C}$ with the property $\delta_y(y') = 0$ if and only if $y \simeq y'$. Then $\mathcal{C}^{+1}$ is universally approximating over $Y$.*

*Proof.* For $y \in Y$, define $r_y := \frac{1}{2} \min_{y' \in Y, y' \not\simeq y} \delta_y(y')$. Define the bump function with radius $r > 0$, $b_r : \mathbb{R} \rightarrow \mathbb{R}$ as $b_r(s) = \psi(\frac{s}{r})$, where

$$\psi(z) = \text{ReLU}(z+1) + \text{ReLU}(1-z) - 2\text{ReLU}(z).$$

Define $k_y := |Y \cap \mathcal{O}_{\mathfrak{G}}(y)|^{-1}$. Since $Y$ is finite and the intersection with the orbit of $y$ contains $y$, $k_y$ is finite and well-defined. We can define the $\mathfrak{G}$-invariant function $h$ from $X$ to $\mathbb{R}$ as:

$$h(x) = \sum_{y \in Y} k_y f(y) b_{r_y}(\delta_y(x)) \tag{32}$$

Notice that $h|_Y = f|_Y$ and $h \in \mathcal{C}^{+1}$. Therefore, $\mathcal{C}^{+1}$ is universally approximating. $\qquad \square$

**Theorem 17.** *If $\mathcal{C}$ is pairwise $Y_{\mathfrak{G}}$ discriminating, then $\mathcal{C}^{+2}$ is universally approximating over $Y$.*

*Proof.* Result follows directly from the two Lemmas above. $\qquad \square$

**Lemma 28.** *Let $X, Y$ be topological spaces and $h : X \times Y \rightarrow \mathbb{R}$ a continuous function. Then, if $Y$ is compact, $f(x) = \inf_{y \in Y} h(x,y)$ is continuous.*

*Proof.* The open sets $(-\infty, a)$ and $(b, \infty)$ form a basis for the topology of $\mathbb{R}$. Thus, we show that their preimage under $f$ is open. First, notice $x \in f^{-1}((-\infty, a))$ if and only if $(x, y) \in h^{-1}((-\infty, a))$ for some $y \in Y$. Therefore, $f^{-1}((-\infty, a)) = p_X(h^{-1}((-\infty, a)))$, where $p_X : X \times Y \rightarrow X$ is the function projecting in the first argument. Since $p_X$ is continuous and open, it follows $p_X(h^{-1}((-\infty, a)))$ is open.

When $x \in f^{-1}((b, \infty))$, it implies that for all $y \in Y, h(x,y) > b$. This means that for all $x \in f^{-1}((b, \infty))$ and $y \in Y$, we have $(x, y) \in h^{-1}((b, \infty))$. Since $h^{-1}((b, \infty)$ is open, then there exists an open box $U_{x,y} \times V_{x,y} \subseteq h^{-1}((b, \infty)$ containing $(x, y)$. Then, the union $\cup_{y \in Y}\big(U_{x,y} \times V_{x,y}\big)$ covers $\{x\} \times Y$. Since $Y$ is compact, there exists a finite subcover $\cup_{y_k}^{K_x}\big(U_{x,y_k} \times V_{x,y_k}\big)$ of size $K_x$. Then notice that the open set $A_x := \cap_{y_k}^{K_x} U_{x,y_k}$ is a neighbourhood around $x$ and $A_x \times Y \subseteq h^{-1}((b, \infty))$. Therefore, $A_x \subseteq f^{-1}((b, \infty))$ and since $x \in f^{-1}((b, \infty))$ was chosen arbitrarily, $f^{-1}((b, \infty))$ is open. $\qquad \square$

**Theorem 18.** *If $\mathcal{C}$ can universally approximate any continuous $\mathfrak{G}$-invariant functions on $Y$, then $\mathcal{C}$ is also pairwise $Y_{\mathfrak{G}}$ discriminating.*

*Proof.* Consider $y, y' \in Y$ such that $y \not\simeq y'$. Then, the function $\delta_y(x) = \inf_{\mathfrak{g} \in \mathfrak{G}} d(y, \mathfrak{g}x) = \min_{\mathfrak{g} \in \mathfrak{G}} d(y, \mathfrak{g}x) > 0$, where the second equality follows from the compactness of $\mathfrak{G}$. This function is $\mathfrak{G}$-invariant. To show that it is continuous, notice that the function $h(x, \mathfrak{g}) = d(y, \mathfrak{g}x)$ is given by the composition $d_y \circ a$, where $a : X \times \mathfrak{G} \to X$ is the continuous group action and $d_y : X \to \mathbb{R}$ is given by $d_y(x) = d(y, x)$, which is also continuous. Since composition of continuous functions is continuous and $\delta_y(x) = \inf_{\mathfrak{g} \in \mathfrak{G}} h(x, \mathfrak{G})$, it follows from Lemma 28 that $\delta_y$ is a continuous function.

Given a universally approximating class of functions $\mathcal{C}$, we can find a function $f$ approximating $\delta_y$ with precision $\epsilon < \frac{\delta_y(y')}{2}$ and, therefore, $f(y) \neq f(y')$. $\square$

**Definition 29.** *Let $\mathcal{C}$ be a class of functions $X \to \mathbb{R}$ and $Y \subseteq X$. We say that $\mathcal{C}$ can locate every orbit over $Y$ if for any $y \in Y$ and any $\varepsilon > 0$ there exists $\delta_y \in \mathcal{C}$ such that:*

1. *For all $y' \in Y$, $\delta_y(y') \geq 0$.*

2. *For all $y' \in Y$, if $y \simeq y'$, then $\delta_y(y') = 0$.*

3. *There exists $r_y > 0$ such that if $\delta_y(y') < r_y$ for any $y' \in Y$, then there is a $\mathfrak{g} \in \mathfrak{G}$ such that $d(y', \mathfrak{g} \cdot y) < \varepsilon$.*

Notice that since $\delta_y \in \mathcal{C}$, it is $\mathfrak{G}$-invariant and then for any $y^* \in \mathcal{O}_{\mathfrak{G}}(y')$, $\delta_y(y') = \delta_y(y^*)$ and there exists $\mathfrak{g} \in \mathfrak{G}$ such that $d(y^*, \mathfrak{g} \cdot y) < \varepsilon$. Therefore, intuitively one should see $\delta_y$ as some sort of "distance function" measuring how far all $y^* \in \mathcal{O}_{\mathfrak{G}}(y')$ are from the orbit of $y$. In other words, when $\delta_y(y^*)$ is low, it means that the entire orbit of $y^*$ is close to the orbit of $y$.

**Lemma 30.** *If $\mathcal{C}$ is a collection of continuous $\mathfrak{G}$-invariant functions from $X \to \mathbb{R}$ that is pairwise $Y_{\mathfrak{G}}$-discriminating, then $\mathcal{C}^{+1}$ is able to locate every orbit over $Y$.*

*Proof.* Select an arbitrary $y \in Y$. For $y' \not\simeq y$, let $\delta_{y,y'}$ be the function in $\mathcal{C}$ separating $y$ and $y'$. Consider the radius $r_{y,y'} := \frac{1}{2}|\delta_{y,y'}(y) - \delta_{y,y'}(y')| > 0$ and define the set

$$A_{y'} := \delta_{y,y'}^{-1}\left(\delta_{y,y'}(y') - r_y, \delta_{y,y'}(y') + r_y\right)$$

Since $\delta_{y,y'}$ is continuous, $A_{y'}$ is open. If $y' \simeq y$, then we define $A_y = B(y, \varepsilon)$, the open ball in $X$ centred at $y$ with radius $\varepsilon$. Clearly, $\bigcup_{y' \in Y} A_{y'}$ forms a cover for $Y$. Since $Y$ is compact, then there exists a finite subcover given by a finite subset $Y_0 \subseteq Y$ such that $\bigcup_{y' \in Y_0} A_{y'}$.

We construct the function $\delta_y(y')$ over $X$ as $\delta_y(y') := \sum_{y' \in Y_0 \setminus \mathcal{O}_{\mathfrak{G}}(y^*)} \bar{\delta}_{y,y'}(y^*)$, where $\bar{\delta}_{y,y'}(y^*) := \max(\frac{4}{3}r_{y,y'} - |\delta_{y,y'}(y^*) - \delta_{y,y'}(y')|, 0)$. Since $\delta_{y,y'}$ is continuous and $\mathfrak{G}$-invariant, so is $\delta_y$. Finally, it can be shown that $\delta_y$ can indeed locate the orbit of $y$ over $Y$.

1. Clearly, $\delta_y(x) \geq 0$ for any $x \in X$.

2. For any $y^* \in Y$, if $y \simeq y^*$, then because $\delta_{y,y'}$ is $\mathfrak{G}$-invariant, we have $\delta_{y,y'}(y^*) = \delta_{y,y'}(y)$, so $\delta_y(y^*) = 0$.

3. First, consider $y^* \in Y$ such that $\forall \mathfrak{g} \in \mathfrak{G}$, $d(\mathfrak{g} \cdot y^*, y) \geq \varepsilon$. Then, $y^* \in Y \setminus \bigcup_{y' \in \mathcal{O}_{\mathfrak{G}}(y)} A_{y'}$. Then, there must be a $y' \in Y_0 \setminus \mathcal{O}_{\mathfrak{G}}(y)$, such that $y^* \in A_{y'}$. Therefore, $|\delta_{y,y'}(y^*) - \delta_{y,y'}(y')| < r_{y,y'} < \frac{4}{3}r_{y,y'}$. Then, we have $\frac{4}{3}r_{y,y'} - |\delta_{y,y'}(y^*) - \delta_{y,y'}(y')| > \frac{4}{3}r_{y,y'} - r_{y,y'} = \frac{1}{3}r_{y,y'} > 0 \Rightarrow \bar{\delta}_{y,y'}(y^*) > \frac{1}{3}r_{y,y'}$. Therefore, we can set $r_y := \frac{1}{3}\min_{y' \in Y_0 \setminus \mathcal{O}_{\mathfrak{G}}(y)} r_{y,y'} > 0$. If $\delta_y(y^*) < r_y$ it follows that for all $y' \in Y_0 \setminus \mathcal{O}_{\mathfrak{G}}(y)$, $\bar{\delta}_{y,y'}(y^*) < \frac{1}{3}r_{y,y'}$, which implies $y^* \in \bigcup_{\mathfrak{g} \in \mathfrak{G}} B(\mathfrak{g} \cdot y, \varepsilon)$. Finally, this proves there is a $\mathfrak{g} \in \mathfrak{G}$ such that $d(y^*, \mathfrak{g} \cdot y) < \varepsilon$.

Since the absolute value function can be realised using ReLU activations, it is easy to see that $\delta_y \in \mathcal{C}^{+1}$. $\square$

**Lemma 31.** *If $\mathcal{C}$, a class of functions over a compact set $Y$, can locate every isomorphism class, then $\mathcal{C}^{+2}$ is universal approximating over $Y$.*

*Proof.* Consider any continuous and $\mathfrak{G}$-invariant function on $X$. Since $Y \subseteq X$ is compact, then $f$ is uniformly continuous when restricted to $Y$. In other words, for all $\varepsilon > 0$, there exists $r > 0$, such that for all $y_1, y_2 \in Y$, if $d(y_1, y_2) < r$, then $|f(y_1) - f(y_2)| < \varepsilon$.

Let $y \in Y$ and define $B_{\mathfrak{G}}(y, r) := \bigcup_{y' \in \mathcal{O}_{\mathfrak{G}}(y)} B(y', r)$ to be the union of all the open balls of radius $r$ on the orbit of $y$. Using the function $\delta_y$ from Definition 29, there exists $r_y$ such that $\delta_y^{-1}([0, r_y)) \subseteq B_{\mathfrak{G}}(y, r)$ for any $y \in Y$. Since $\delta_y$ is continuous, $\delta_y^{-1}([0, r_y))$ is open. Therefore, $\{\delta_y^{-1}([0, r_y))\}_{y \in Y}$ is an open cover for $Y$. Since $Y$ is compact, we can find a finite subcover $\{\delta_y^{-1}([0, r_y))\}_{y \in Y_0}$, where $Y_0$ is a finite subset of $Y$.

We can now use the functions $\delta_y$ to construct a set of continuous $\mathfrak{G}$-invariant functions that forms a partition of unity for this finite cover. For $y_0 \in Y_0$ we construct the function $\phi_{y_0}(y') = \max(r_{y_0} - \delta_{y_0}(y'), 0)$ and the function $\phi(y') = \sum_{y^* \in Y_0} \phi_{y^*}(y')$, both of which are continuous. Noticing that $\mathrm{supp}(\phi_{y_0}) = \delta_{y_0}^{-1}([0, r_{y_0}))$ and that $\phi_{y^*}(y') > 0$ for any $y' \in Y$, the set of functions $\psi_{y_0}(y') = \frac{\phi_{y_0}(y')}{\phi(y')}$ form a partition of unity with $\sum_{y_0 \in Y_0} \psi_{y_0}(y') = 1$ for all $y' \in Y$.

Notice that we can write any $\mathfrak{G}$-invariant function $f$ as:

$$f(y') = f(y') \sum_{y_0 \in Y_0} \psi_{y_0}(y') = \sum_{y_0 \in Y_0 \,:\, y' \in \delta_{y_0}^{-1}([0, r_{y_0}))} f(y') \psi_{y_0}(y') \tag{33}$$

The intuition is that because $f$ is continuous, we can approximate $f(y')$ in the expression above by the value of $f(y_0)$ since $y'$ is in the neighbourhood of some $y_0$. Thus, the function that approximates $f$ is $h(y') = \sum_{y_0 \in Y_0} f(y_0) \psi_{y_0}(y')$. We now show that $h$ can approximate $f$ with arbitrary accuracy.

If $y' \in \delta_{y_0}^{-1}([0, r_{y_0}))$, then there exists $\mathfrak{g} \in \mathfrak{G}$ such that $d(y', \mathfrak{g} \cdot y_0) < r$. Using the fact that $f$ is continous, this implies $|f(y') - f(\mathfrak{g} \cdot y_0)| < \varepsilon$. Because $f$ is invariant, $f(y_0) = f(\mathfrak{g} \cdot y_0)$, which implies $|f(y') - f(y_0)| < \varepsilon$. Then we have:

$$\Big| f(y') - \sum_{y_0 \in Y_0} f(y_0) \psi_{y_0}(y') \Big| = \Big| f(y') - \sum_{y_0 \in Y_0 \,:\, y' \in \delta_{y_0}^{-1}([0, r_{y_0}))} f(y_0) \psi_{y_0}(y') \Big| \tag{34}$$

$$= \sum_{y_0 \in Y_0 \,:\, y' \in \delta_{y_0}^{-1}([0, r_{y_0}))} |f(y') - f(y_0)| \, \psi_{y_0}(y') < \varepsilon \tag{35}$$

Finally, to see that $h$ is in $\mathcal{C} + 2$, we can use an MLP with one hidden layer to approximate $\psi_{y_0}$ followed by one final layer to compute the linear combination of the $\psi_{y_0}$. $\qquad\square$

**Theorem 19.** *If $\mathcal{C}$, a class of functions over $Y$, is pairwise $Y_{\mathfrak{G}}$ discriminating, then $\mathcal{C}^{+2}$ can also universally approximate any continuous function over $Y$.*

*Proof.* The proof follows from the two Lemmas above. $\qquad\square$

## G.2 NUMBER OF AGGREGATORS IN CONTINOUS SETTING

**Theorem 20.** *Let $X$ be a smooth $n$-dim manifold and $\mathfrak{G}$ an $m$-dim compact Lie group acting continuously on $X$. Suppose there exists a smooth submanifold $Y$ of $X$ of the same dimension such that $\mathfrak{G}$ acts freely on it. Then, any $\mathfrak{G}$-orbit injective function $f : X \to \mathbb{R}^d$ requires that $d \geq n - m$.*

*Proof.* Suppose for the sake of contradiction that there exists an orbit-space injective and continuous function $f : X \to \mathbb{R}^m$, with $m < d$. Since $Y$ is a submanifold of the same dimension as $X$, then $f$ must also be injective over $Y$. By the Quotient Manifold Theorem (Lee, 2013), $Y/\mathfrak{G}$ is a topological manifold of dimension $d = \dim X - \dim \mathfrak{G}$. The map $f$ induces an injective function $g : Y/\mathfrak{G} \to \mathbb{R}^m$. This map is also continuous because for an open set $V \in \mathbb{R}^m$, $g^{-1}(V) = \pi_Y(f^{-1}(V))$. Because $f$ is continuous and $\pi_Y$ is an open map, this set is open.

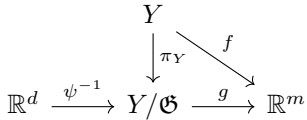

Because $Y/\mathfrak{G}$ is a manifold, there exist an open set $U \subseteq Y/\mathfrak{G}$ and a homeomorphism $\psi : U \to \mathbb{R}^d$. Then the composition $h = g \circ \psi^{-1}$ is a continuous and injective map from $\psi(U) \subseteq \mathbb{R}^m$ to $\mathbb{R}^m$. By the Invariance of Domain Theorem (Bredon, 2013, Corollary 19.9), $h$ is open and it is a homeomorphism onto its image $h(\psi(U)) \subseteq \mathbb{R}^m$. By the Invariance of Dimension Theorem (Bredon, 2013, Corollary 19.10), $d = m$. □

**Theorem 21.** *For $n \geq d - 1 > 0$ or $n = d = 1$, any continuous $S_n \times SO(d)$ orbit-space injective function $f : \mathbb{R}^{n \times d} \to \mathbb{R}^q$ requires that $q \geq nd - d(d-1)/2$.*

*Proof.* We now consider the case when $\mathfrak{G} = S_n \times SO(d)$. First, notice that the proof above also holds for this group since it is a subgroup of $S_n \times O(d)$. However, we can obtain a stronger result and show the result holds for $n \geq d - 1$. In what follows, we reuse the notation from the proof above.

We define the set

$$Z' = \{\boldsymbol{X} \in X \mid \exists\, 1 \leq i_1 < \ldots < i_{d-1} \leq n \text{ s.t } \boldsymbol{x}_{i_1}, \ldots, \boldsymbol{x}_{i_{d-1}} \text{ are linearly independent }\},$$

containing $d - 1$ row-vectors that are linearly independent. Define $M_{\boldsymbol{X}}$ to be the set of all $(d - 1) \times (d - 1)$ minors of the matrix $\boldsymbol{X}$. Then, we can construct a continuous function $h(\boldsymbol{X}) = \max_{m \in M_{\boldsymbol{X}}} |m|$ and notice that $Z'$ coincides with the open set $h^{-1}((0, \infty))$. Then, the set $V = Y \cap Z'$ is also open and non-empty. Therefore, $V$ is a submanifold of $X$ of the same dimension and the action of $\mathfrak{G}$ is well-defined and continuous on $W$. We can show again this action is free.

As in the proof above, we have that $\boldsymbol{P}_{\mathfrak{g}} = \boldsymbol{I}_n$, so it remains to inspect the case $\boldsymbol{X} = \boldsymbol{X}\boldsymbol{Q}_{\mathfrak{g}}$. Any non-trivial rotation $\boldsymbol{Q}_{\mathfrak{g}}$ must rotate at least a two-dimensional subspace of $\mathbb{R}^d$. Since the rows of the matrix $\boldsymbol{X}$ span a $(d-1)$-dimensional subspace of $\mathbb{R}^d$, then $\boldsymbol{Q}_{\mathfrak{g}}$ cannot leave $\boldsymbol{X}$ invariant unless $\boldsymbol{Q}_{\mathfrak{g}} = \boldsymbol{I}_d$. Applying Theorem 20 again yields the result.

For $n = d = 1$, $\|\cdot\| : \mathbb{R}^{1 \times d} \to \mathbb{R}$ is, as before, $\mathfrak{G}$ orbit-space injective. □

**Theorem 22.** *For $n \geq d > 0$, any continuous $S_n \times O(d)$ orbit-space injective function $f : \mathbb{R}^{n \times d} \to \mathbb{R}^q$ requires that $q \geq nd - d(d-1)/2$.*

*Proof.* First, suppose that $n \geq d > 1$. Consider the subspace $Y = \{\boldsymbol{X} \in X \mid \|\boldsymbol{x}_i\| \neq \|\boldsymbol{x}_j\|, \forall i < j\}$, where the norm is just the standard Euclidean norm. Consider the function $g : X \to \mathbb{R}$ given by $g(\boldsymbol{X}) = \min_{i < j} |\|\boldsymbol{x}_i\| - \|\boldsymbol{x}_j\||$. By standard analysis, this function is continuous and notice that $Y = g^{-1}((0, \infty))$, which means that $Y$ is open in $X$. We also define the set

$$Z = \{\boldsymbol{X} \in X \mid \exists\, i_1 < \ldots < i_d < n, |\det(\boldsymbol{x}_{i_1}, \ldots, \boldsymbol{x}_{i_d})| > 0\},$$

containing row-vectors that span $\mathbb{R}^d$. As above, this set is the preimage of the absolute determinant over $(0, \infty)$, which makes $Z$ open in $X$. Then, the set $W = Y \cap Z$ is also open and non-empty. Therefore, $W$ is a submanifold of $X$ of the same dimension and the action of $\mathfrak{G}$ is well-defined and continuous on $W$. We can show this action is free.

We investigate the solutions of the equation $\boldsymbol{P}_{\mathfrak{g}}\boldsymbol{X}\boldsymbol{Q}_{\mathfrak{g}}^\top = \boldsymbol{X} \iff \boldsymbol{P}_{\mathfrak{g}}\boldsymbol{X} = \boldsymbol{X}\boldsymbol{Q}_{\mathfrak{g}}$ for $\boldsymbol{X} \in W$. Since orthogonal transformations preserve norms and the rows of $\boldsymbol{X}$ have different norms, it follows that $\boldsymbol{P}_{\mathfrak{g}} = \boldsymbol{I}_n \Rightarrow \boldsymbol{X} = \boldsymbol{X}\boldsymbol{Q}_{\mathfrak{g}}$. We know that a subset of the rows of $\boldsymbol{X}$ span the whole of $\mathbb{R}^d$. Define the sub-matrix of $\boldsymbol{X}$ containing these rows by $\boldsymbol{X}^* \in \mathbb{R}^{d \times d}$. Then, we have $\boldsymbol{X}^*\boldsymbol{Q}_{\mathfrak{g}} = \boldsymbol{X}^* \Rightarrow \boldsymbol{Q}_{\mathfrak{g}} = (\boldsymbol{X}^*)^{-1}\boldsymbol{X}^* = \boldsymbol{I}_d$. This proves that the action is free and applying Theorem 20, yields the result.

For the trivial case when $n = 1$, notice that $\|\cdot\| : \mathbb{R}^{1 \times d} \to \mathbb{R}$ is $\mathfrak{G}$ orbit-space injective. □

**Proposition 23.** *Any $S_n$-invariant injective function $f : \mathbb{R}^{n \times d} \to \mathbb{R}^q$ requires $q \geq nd$.*

*Proof.* Reusing the notation from above, notice that for all $n \geq 1$, $S_n$ acts freely on the sub-manifold $Y$ as shown above. Seeing $S_n$ as a zero-dimensional Lie group and applying Theorem 20 yields the result. □

# H   ALTERNATIVE GWL FORMULATION FOR $SO(2)$

The version of GWL presented in the main text makes use of a geometric neighbourhood object $\boldsymbol{g}$ that keeps increasing in size at each iteration. A natural question to ask is whether a more compact encoding of the orientation of a neighbourhood can be achieved. In this section, we show this is possible for $SO(2)$, which has some particularly convenient properties: it is commutative and its (discrete) subgroups are particularly simple. First, we point out why when working with the standard representation of $SO(d)$ acting on vectors in $\mathbb{R}^d$, the orientation of a neighbourhood cannot be (injectively) encoded into another vector obtained from permutation invariant aggregation.

**Definition 32** (Representation). *A representation of a group $\mathfrak{G}$ on a vector space $V$ over is a group homomorphism from $\mathfrak{G}$ to $GL(V)$, the general linear group on $V$. That is, a representation is a map $\rho : \mathfrak{G} \to GL(V)$, such that $\rho(\mathfrak{g}_1 \mathfrak{g}_2) = \rho(\mathfrak{g}_1)\rho(\mathfrak{g}_2)$.*

Denote by $\rho^0$ the usual representation of $O(d)$ on $\mathbb{R}^d$. Let $\mathcal{X} \subset \mathbb{R}^d$ be a countable feature space, $X = \{\boldsymbol{x}_1, \ldots, \boldsymbol{x}_n\} \subset \mathcal{X}$ a multi-set of features and associated representations $\rho_1, \ldots, \rho_n : \mathfrak{G} \to SO(d)$. Furthermore assume that $\mathfrak{G}$ acts on $\mathbb{R}^{n \times d}$ via $gX := [\rho^1(\boldsymbol{x}_1), \ldots, \rho^n(\boldsymbol{x}_n)]$. We want to find a multi-set function $f : \mathcal{X}^n \to \mathbb{R}^d$ that is $O(d)$-equivariant and injective over the orbit of $X$, denoted by $\mathcal{O}_{\mathfrak{G}}(X)$. A first observation is that if $O(d)$ acts (faithfully) via $\rho^0$ on $\mathbb{R}^d$, then there is no such function due to the presence of discrete rotational symmetries in certain sets of input vectors (i.e. snowflake like structures).

**Proposition 33** (The Discrete Symmetry Problem). *Let $\boldsymbol{X} \in \mathbb{R}^{n \times 2}$ be a point cloud with $n$ points having (discrete) rotational symmetry, i.e. there exists $\pi \in S_n$ and $\mathfrak{g} \neq \mathrm{id} \in O(2)$ s.t. $\pi \boldsymbol{X} = \mathfrak{g} \boldsymbol{X}$. Then, any permutation-invariant and $O(d)$-equivariant function $f : \mathbb{R}^{n \times 2} \to \mathbb{R}^2$ has the property that $f(\boldsymbol{X}) = f(\mathfrak{g}\boldsymbol{X}) = 0$ for all $\mathfrak{g}$.*

*Proof.* $\pi \boldsymbol{X} = \mathfrak{g} \boldsymbol{X} \implies f(\pi \boldsymbol{X}) = f(\mathfrak{g} \boldsymbol{X}) \implies f(\boldsymbol{X}) = \rho^0(\mathfrak{g}) f(\boldsymbol{X}) \implies f(\boldsymbol{X}) = 0$ ☐

This result easily extends to any dimension $d$ by finding such a set of vectors that span a two dimensional plane and performing a rotation in that plane. The proposition above shows that we need to find an alternative representation $\rho$. We show that this is possible for $\mathfrak{G} = SO(2)$ and for the rest of this section assume $d = 2$.

**Theorem 34.** *There exists a representation $\rho : \mathfrak{G} \to SO(\mathbb{R}^2)$ and a multi-set function $\mathrm{HASH}_v$ sending $X$ to $\mathbb{R}^2 \setminus \{0\}$ that is unique for each multi-set $X \subset \mathcal{X}$ of bounded size and $\mathrm{HASH}_v(\mathfrak{g} X) = \rho(\mathfrak{g})\mathrm{HASH}_v(X)$. Furthermore, for any two $X_1, X_2 \subset \mathcal{X}$ we have that $\|\mathrm{HASH}_v(X_1)\| = \|\mathrm{HASH}_v(X_2)\|$ iff there is $\mathfrak{g} \in \mathfrak{G}$ such that $gX_1 = X_2$.*

*Proof.* Let $\mathcal{O}_{\mathfrak{G}}(X)$ be an orbit of $X$ generated by the action of the group $\mathfrak{G} = SO(2)$ acting on $\mathbb{R}^{n \times d}/S_n$ (i.e. the space of multi-sets of 2D vectors) as in the statement of Theorem 34. Then, $SO(2)$ acts transitively on $\mathcal{O}_{\mathfrak{G}}(X)$ by construction.

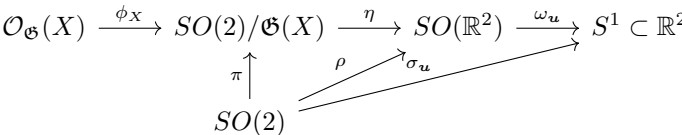

Denoting by $\mathfrak{G}(X)$ its isotropy/stabilizer group of $X$, the function $\phi_X : \mathcal{O}(X) \to SO(2)/\mathfrak{G}(X)$ given by $\phi_X(Y) = \phi_X(\mathfrak{g} X) = \mathfrak{G}(X)$ with $g \in SO(2)$ is well-defined and, moreover, it is an equivariant homeomorphism of $\mathfrak{G}$-spaces (*e.g.* see Proposition 4.1 in Bredon (1972)).

Notice that because $SO(2)$ is abelian, $\mathfrak{G}(X)$ is a normal subgroup and $SO(2)/\mathfrak{G}(X)$ is not only a manifold, but also a (Lie) group. We will show that we can find a faithful orthogonal representation of this quotient group. Because we are working with finite neighbourhoods, the group $\mathfrak{G}(X)$ is a finite cyclic subgroup (standard result) if $X \neq 0$. Otherwise, $\mathfrak{G}(X) = SO(2)$.

Therefore, let us denote by $\alpha(\mathfrak{g}\mathfrak{G}(X))$ the smallest angle of a rotation in the equivalence class $\mathfrak{g}\mathfrak{G}(X)$. If $\mathfrak{G}(X)$ is a finite cyclic subgroup generated by rotations of an angle $\theta$, then $\alpha(SO(2)/\mathfrak{G}(X)) = [0, \theta)$. If $\mathfrak{G}(X) = SO(2)$, then $\alpha(SO(2)/\mathfrak{G}(X)) = 0$. Thus, we can

construct the orthogonal representation $\eta : SO(2)/\mathfrak{G}(X) \to SO(\mathbb{R}^d)$ given by $\eta(\mathfrak{g}\mathfrak{G}(X)) = \boldsymbol{R}_{\alpha(SO(2)/\mathfrak{G}(X))2\pi/\theta}$ if $\mathfrak{G}(X)$ is finite and $\eta(\mathfrak{g}\mathfrak{G}(X)) = \boldsymbol{I}$, otherwise. It can be checked that in either case, $\eta$ is an injective group homomorphism and, therefore, a faithful representation of the group.

Given $\boldsymbol{u} = [1,0] \in S^1 \subset R^2$, we can define a map $\omega_{\boldsymbol{u}} : SO(\mathbb{R}^d) \to S^1$ given by $\omega_{\boldsymbol{u}}(\boldsymbol{R}_\alpha) = \boldsymbol{R}_\alpha \boldsymbol{u}$, which is another homeomorphism. Thus, the composition is $\psi = \omega_{\boldsymbol{u}} \circ \eta \circ \phi_X : \mathcal{O}_{\mathfrak{G}}(X) \to \mathbb{R}^2$ is an injective function. Moreover, because the quotient map $\pi : SO(2) \to SO(2)/\mathfrak{G}(X)$ is a group homomorphism, we can compose it with $\eta$ to lift the representation of $SO(2)/\mathfrak{G}(X)$ to a representation $\rho = \eta \circ \pi$ of $SO(2)$. Therefore, it follows that $\psi$ is also equivariant with respect to this representation of $SO(2)$.

Finally, we can extend $\psi$ to a function that is injective for any multi-set by constructing a $\psi_{\mathcal{O}_{\mathfrak{G}}(X)}$ for each orbit $\mathcal{O}_{\mathfrak{G}}(X)$, where $X$ is a representative of the orbit, by using a map $\omega_{\mathcal{O}_{\mathfrak{G}}(X)}$ where $\boldsymbol{u} = [\text{I-HASH}(X), 0]$ (i.e. the norm encodes the orbit). $\qquad\square$

**Alternative $SO(2)$ Geometric Weisfeiler-Leman Test**. Given a geometric graph $(\mathcal{G}, \boldsymbol{X}, \boldsymbol{H})$, the $SO(2)$-GWL algorithm consists of three steps:

1. For each node $v \in \mathcal{G}$ and $u \in \mathcal{N}(v)$, initialise:

$$\boldsymbol{m}_{v,u}^0 := \text{HASH}(\boldsymbol{h}_v, \boldsymbol{h}_u, \|\text{HASH}_v(\boldsymbol{x}_v - \boldsymbol{x}_u)\|) \frac{\text{HASH}_v(\boldsymbol{x}_v - \boldsymbol{x}_u)}{\|\text{HASH}_v(\boldsymbol{x}_v - \boldsymbol{x}_u)\|}$$

$$\boldsymbol{m}_v^0 := \text{HASH}_v(\boldsymbol{m}_{v,u_1}^0, \ldots, \boldsymbol{m}_{v,u_{d(v)}}^0)$$

2. For each node $v \in \mathcal{G}$, update $\boldsymbol{m}_v^{l+1} := \text{HASH}_v(\boldsymbol{m}_{u_1}^l, \ldots, \boldsymbol{m}_{u_{d(v)}}^l)$.

3. Go back to step 2 until the partition induced by $\{\!\{\|\boldsymbol{m}_v^l\| : v \in \mathcal{G}\}\!\}$ becomes stable.

4. Return the colours of the nodes $\{\!\{\|\boldsymbol{m}_v^L\| : v \in \mathcal{G}\}\!\}$ at the last iteration $L$.

