# OpenReview forum: "On the Expressive Power of Geometric Graph Neural Networks"
_ICLR.cc/2023/Conference — Submitted to ICLR 2023_

### Official Review · Reviewer_J5ce · 2022-10-25

**Confidence:** 3
**Correctness:** 2
**Technical Novelty And Significance:** 3
**Empirical Novelty And Significance:** 3
**Recommendation:** 3

**Clarity, Quality, Novelty And Reproducibility:**

There are several problems that need to be clarified:
- I do not think that x_{ij} appearing in equations (3) (4) and (6) is defined anywhere. Node coordinates are defined as a matrix in R^{nxd} but here indices i and j are in [n]. I guess x_{ij} is the vector obtained from the coordinates of nodes i and j?
- The definition of geometric GNN needs to be clarified. In particular the aggregate function \psi needs to be invariant with respect to the action of the group considered and the permutation group. How do you ensure that?
- Similarly, the aggregation step in the geometric WL is not well defined. How do you define the final hash in equation (7). Please be more explicit. The authors write: "by using aG-orbit injective andG-invariant function that we denote by I-HASH." but clearly a constant function will be invariant but useless here. So what function do you take exactly?

**Strength And Weaknesses:**

The theoretical analysis extends known results connecting WL to the expressive power of GNN.

I would have liked to get a better motivation for this paper. It is not clear at all to me that this extension to geometric GNN will be of any use. What would be a learning task where these GNNs will produce better results than standard GNNs? It is fine to present theoretical results but I think that a clear connection to machine learning is missing here.


**Summary Of The Paper:**

This paper introduces equivariant and invariant geometric graph neural networks. The authors provide a theoretical analysis of these architectures by introducing a geometric version of the Weisfeiler-Leman (WL) test. In addition to the graph structure, nodes have now geometric attributes (like position, velocity...) in geometric graphs. Geometric graphs will be said to be geometrically isomorphic, if their underlying graphs are isomorphic and their geometric attributes can be obtained through rotations (or reflections). In this setting, a natural extension of WL is proposed and connected to the expressive power of geometric GNN.

**Summary Of The Review:**

This paper is not ready for publication and need some major revisions.

---

> ### Author Response · Authors · 2022-11-13
> **(Part 1/2) Point-by-point response to Reviewer 4**
>
> >R4.1. I would have liked to get a better motivation for this paper. It is not clear at all to me that this extension to geometric GNN will be of any use. What would be a learning task where these GNNs will produce better results than standard GNNs? It is fine to present theoretical results but I think that a clear connection to machine learning is missing here.
>
> Thank you for the honest feedback. In the revised **Introduction**, we have taken much care to better motivate why geometric graphs are interesting and practically relevant, as well as why we cannot apply standard GNNs in these applications. Briefly, geometric graphs come with both **relational structure** as well as **geometric attributes** acted upon by physical symmetries. Feeding the geometric attributes into standard GNNs and associated theoretical toolkits would no longer retain their physical meaning and transformation behaviour. However, [retaining these semantics is critical](https://arxiv.org/abs/2203.06153) for geometric graph applications such as protein design, electrocatalyst discovery and molecular dynamics simulations.
>
> Our work thus fills a **key research gap** – we propose the Geometric Weisfeiler-Leman test (GWL), a novel theoretical tool to characterise the expressive power of geometric GNNs from the perspective of graph isomorphism. GWL helps understand the workings and theoretical limitations of geometric GNNs.
>
> Thank you for encouraging us to better emphasise our practical implications. Towards making clearer connections between theory and practice, we have expanded **Section 4** on probing the design space of current geometric GNNs via GWL, as well as synthetic experiments highlighting practical challenges in **Appendix E**. Please refer to the revised manuscript for more, we highlighted the main changes to addressing this point with the boxes labelled “R4.1”.
>
> >R4.2. I do not think that x_{ij} appearing in equations (3) (4) and (6) is defined anywhere. Node coordinates are defined as a matrix in R^{nxd} but here indices i and j are in [n]. I guess x_{ij} is the vector obtained from the coordinates of nodes i and j?
>
> Thank you for highlighting the lack of clarity. You are correct in your understanding that x_{ij} = x_i - x_j denotes the relative position vector. This is the conventional notation in the literature for geometric GNNs, and using relative positions ensures translation invariance in GWL as well as geometric GNN layers. We have addressed this in our revision.
>
> >R4.3. The definition of geometric GNN needs to be clarified. In particular the aggregate function \psi needs to be invariant with respect to the action of the group considered and the permutation group. How do you ensure that?
>
> Thank you for bringing this up. We agree that a more extensive background on which geometric GNNs are covered under the GWL framework is helpful. In **Appendix B**, we now provide concrete examples of invariant GNNs (SchNet and DimeNet), equivariant GNNs using cartesian vectors (PaiNN, E-GNN, GVP-GNN), and equivariant GNNs using spherical tensors (TFN, Cormorant, MACE).
>
> Indeed the aggregation step for geometric GNNs needs to be permutation invariant and $\mathfrak{G}$-invariant or $\mathfrak{G}$-equivariant, depending on the class of architectures under consideration. A typical example of $\mathfrak{G}$-invariant aggregation is the SchNet layer in eq.11 - SchNet ensures invariance to rotations, reflections and translations by using only the **relative distances** between the central node and each neighbour to aggregate their features. The relative distances always remain the same no matter how the geometric graph may be transformed in space.
>
> For $\mathfrak{G}$-equivariant aggregation, PaiNN in eq. 16 upgrades SchNet to include neighbourhood vector features and relative displacements, both of which are $\mathfrak{G}$-equivariant and translation invariant.
>
> We have elaborated on this in Appendix B and highlighted the main changes to addressing this point with the boxes labelled “R4.3”.

---

> > ### Author Response · Authors · 2022-11-13
> > **(Part 2/2) Point-by-point response to Reviewer 4**
> >
> > >R4.4. Similarly, the aggregation step in the geometric WL is not well defined. How do you define the final hash in equation (7). Please be more explicit. The authors write: "by using a G-orbit injective and G-invariant function that we denote by I-HASH." but clearly a constant function will be invariant but useless here. So what function do you take exactly?
> >
> > Thank you for bringing the lack of detail to our notice; we agree that we can be more precise about the definition of $\text{I-HASH}$ and its connection with neighbourhood aggregation/scalarisation in geometric GNNs.
> >
> > As you noted, the node colouring function $\text{I-HASH}$ is G-orbit injective and G-invariant. $\text{I-HASH}$ distinguishes $\mathfrak{G}$-orbits of neighbourhoods, i.e. it is a **neighbourhood fingerprint** which returns the same colour for rotated versions of the same neighbourhood. $\text{I-HASH}$ is thus an idealised or perfect $\mathfrak{G}$-orbit injective function, similar to the $\text{HASH}$ function used in WL, which means it is not necessarily continuous. Thus, GWL and $\text{I-HASH}$ are meant to serve as abstract mathematical tools to uncover practical results about geometric GNN models, and may not be implementable.
> >
> > In geometric GNNs, the node colouring operation corresponds to the **scalarisation step** where local geometric information from subsets of neighbours is aggregated to compute $\mathfrak{G}$-invariant scalar quantities. The number of neighbours involved is termed the body order of scalarisation. For e.g. distances among the central node and a neighbour are 2 body scalars, angles among triplets of neighbour - central node - neighbour are 3 body scalars, and so on. These are possible examples of G-invariant aggregators used in geometric GNNs, but distances and angles are also well known to be incomplete descriptors of neighbourhoods, i.e. not $\mathfrak{G}$-orbit injective even for some [seemingly simple counterexamples](https://journals.aps.org/prl/abstract/10.1103/PhysRevLett.125.166001).
> >
> > Notably, the recently proposed [MACE layer](https://arxiv.org/abs/2206.07697) constructs a complete basis of scalars up to arbitrary body order $k$ via Atomic Cluster Expansion. In the revised **Section 4** studying the geometric GNN design space via GWL, we have emphasised how MACE can be a possible candidate for defining a continuous $\mathfrak{G}$-orbit injective function if the conditions in Proposition 11 are met.
> >
> > Interestingly, our synthetic experiments in Table 1 from **Appendix C** indicate that, while MACE may be sufficient in theory, it remains practically challenging to design and optimise such higher order neighbourhood fingerprints.
> >
> > >R4.5. This paper is not ready for publication and need some major revisions.
> >
> > Thank you for your actionable feedback which helped significantly improve the manuscript. We believe we have addressed all the concerns raised. Please let us know if you are satisfied with our responses and have any further queries that we can address.

---

> ### Author Response · Authors · 2022-11-16
> **We are happy to discuss further**
>
> Dear reviewer, thank you very much once again for taking the time to review our manuscript and for your actionable feedback so far.
>
> We have worked hard to address the concerns raised by the original review in the revised manuscript, including better motivating why our work is relevant and important, as well as adding synthetic experiments and strengthening connections with practice. We have clarified the concerns around use of relative positions, geometric GNNs covered under our framework, and definitions of $\text{I-HASH}$ and the corresponding scalarisation in geometric GNNs.
>
> As we are nearing the Reviewer-Author Discussions deadline (18 November), please let us know if you are satisfied with our responses and have any further queries that we can address. If we have addressed your concerns, please consider revising your score.

---

> ### Author Response · Authors · 2022-12-05
> **Discussion phase 2 ending soon; please let us know if your concerns have been addressed**
>
> Dear reviewer,
>
> Apologies for the repeated message, and thank you once again for your insightful review and actionable feedback.
>
> We have worked very hard to improve our manuscript and address the concerns raised by yourself and other reviewers during the revision period. As a quick summary, we addressed your main concerns by:
> - Better motivating why our work is relevant and important.
> - Adding synthetic experiments and strengthening connections with practice.
> - Clarifying the concerns around use of relative positions and the various classes of geometric GNNs covered under our framework.
> - Better elaborating on the definitions of $\text{I-HASH}$ and the corresponding scalarisation in geometric GNNs.
>
> We would like to re-iterate that this is the **first work to study the expressive power of geometric GNNs** from the perspective of graph isomorphism. We have highlighted how this is non-trivial and fills a key research gap, as theoretical tools for standard GNNs are not applicable to the increasingly relevant special case of geometric graphs in drug discovery, biochemistry, material science, robotics, etc.
>
> As there is less than one week till the end of Discussions Phase 2 (**12 December**), please let us know if you are satisfied with our revised manuscript and have any further queries that we would be happy to address. If we have addressed your concerns, please consider revising your score.
>
> Soliciting actionable feedback from expert reviewers such as yourself to craft the best version of our paper is our top priority here.
>
> Best regards,
>
> Authors

---

> > ### Comment · Reviewer_J5ce · 2022-12-06
> > **still not convinced**
> >
> > Thank you for your answer but I still do not understand the motivation for this paper. The problem of geometric attributes being invariant to a group of transformation is already present for point clouds. There you can get rid of this problem by centering the points and aligning them thanks to a simple SVD. This is a very simple pre-processing task on the data and then in your case where there is an additional graph feature, you are left with the standard graph isomorphism problem. This seems a much simpler approach than yours and I do not see where your approach will be more efficient.

---

> > > ### Author Response · Authors · 2022-12-07
> > > **(Part 1/2) Post-rebuttal response to Reviewer 4**
> > >
> > > > Thank you for your answer but I still do not understand the motivation for this paper.
> > >
> > > Thank you for taking the time to review our paper, again. We have tried our best in the revised manuscript and in our point-by-point responses to provide technical reasons and motivations around the rising importance of geometric GNNs, why they are different from normal GNNs, and the significance of our work.
> > >
> > > **We have addressed all of your remaining concerns** and we do hope that you will take this into account in your evaluation and consider revising your score.
> > >
> > > If your primary ground for strong rejection is that you are unconvinced by the need for geometric GNNs altogether, as stated in your review and response, we would like to sincerely request you to reconsider this position. Let us elaborate below.
> > >
> > > ## What is our main motivation
> > >
> > > Our main motivation is to study the theoretical properties and **expressive power** of **geometric GNNs**.
> > > - Geometric GNNs use an iterative, **local message passing** procedure where the messages are rotation-invariant/equivariant.
> > > - Our theoretical framework, the Geometric WL test, mirrors this message passing procedure of geometric GNNs. This **alignment** is critical as it allows us to precisely characterise the expressive power of geometric GNNs via GWL.
> > > - GWL can subsequently be used to answer questions around the theoretical limits of geometric GNNs: What is the **maximally powerful geometric GNN** capable of in the best case scenario? What are the limitations and tradeoffs between various classes of geometric GNNs?
> > >
> > > ## What is not our motivation
> > >
> > > It was **not** our goal to provide the most efficient algorithm for geometric graph isomorphism.
> > > - We used a geometric GNN’s ability to solve geometric graph isomorphism as a **measure of expressivity**. There could be other measures of expressive power, too.
> > > - GWL is **one possible approach** to geometric graph isomorphism, which is motivated by the Weisfeler-Leman graph isomorphism test for normal graphs. It may not be the most efficient approach.
> > > - As you noted, there could well be other approaches for this problem, e.g. based on SVD. However, these approaches **may not enable us to study the expressive power of geometric GNNs** in the same way as GWL does. The main motivation of our work is to study geometric GNN expressivity and GWL is a tool to do so.
> > >
> > > ## Why are geometric GNNs exciting
> > >
> > > In the revised manuscript and responses, we have provided technical reasons and motivations around the importance of geometric GNNs, why they are different from normal GNNs, and the significance of our work. Let us supplement this with more ‘soft’ reasons which make us excited about this emerging field:
> > > - Methodological and technical advances in geometric GNNs are regularly published at ICLR as well as NeurIPS, ICML, etc. (see references in the manuscript).
> > > - Geometric GNNs have now become the core neural network toolkit for recent advances and breakthroughs across **scientific discovery** applications, including but not limited to:
> > >     - [Designing novel proteins](https://www.science.org/doi/10.1126/science.add2187) (Science, 2022)
> > >     - [RNA folding prediction](https://www.science.org/doi/10.1126/science.abe5650) (Science, 2021, [featured on cover page](https://images-cdn.dashdigital.com/sciencemagazine/27_august_2021/data/imgpages/mobile3/0001_mmmnek.jpg?lm=1630003718000))
> > >     - [Molecular dynamics simulations](https://www.nature.com/articles/s41467-022-29939-5) (Nature Communications, 2022)
> > > - These advances have attracted significant investment from [large](https://www.microsoft.com/en-us/research/lab/microsoft-research-ai4science/) [corporations](https://www.isomorphiclabs.com/) as well as [startup](https://generatebiomedicines.com/) [companies](https://atomic.ai/) into translating geometric GNN-driven research into real-world technologies.
> > >
> > > In the light of all this evidence towards the rising importance of geometric GNNs, we feel that **new theoretical tools to advance our understanding of geometric GNN fundamentals** are much needed.
> > >
> > > Please do note that all the other reviewers agree with our position on the significance of geometric GNNs and our motivation:
> > > - R1: “Learning on geometric graphs becomes more and more significant these days, and theoretical analysis of geometric GNNs is somehow limited. This work picks a good field.”
> > > - R2: “The idea to extrapolate the WL hierarchy to geometric graphs is important given the recent advances and interesting”
> > > - R3: “The paper has a very clear motivation, and attacks a problem of interest, given the recent successes of geometric GNNs in various key applications in the physical sciences.”

---

> > > > ### Author Response · Authors · 2022-12-07
> > > > **(Part 2/2) Post-rebuttal response to Reviewer 4**
> > > >
> > > > > The problem of geometric attributes being invariant to a group of transformation is already present for point clouds. There you can get rid of this problem by centering the points and aligning them thanks to a simple SVD. This is a very simple pre-processing task on the data and then in your case where there is an additional graph feature, you are left with the standard graph isomorphism problem. This seems a much simpler approach than yours and I do not see where your approach will be more efficient.
> > > >
> > > > We would like to note that this is a new concern regarding the efficiency of the proposed approach as well as comparing it to an alternative approach based on SVD. This concern was not raised in the initial review, nor during the author-reviewer discussion phase, when we could have addressed it directly within the manuscript.
> > > >
> > > > It may well be the case that applying SVD followed by normal GNNs is a **viable alternative** to the current paradigm of geometric GNNs which follows the [Geometric Deep Learning](https://arxiv.org/abs/2104.13478) blueprint. However, to the best of our knowledge, we are **not aware of such an approach being proposed** in any recent papers in our application areas of interest.
> > > >
> > > > Additionally, as we previously stated, such an SVD-based approach **may not enable us to study the expressive power of geometric GNNs** in the same way as the proposed GWL framework does. GWL mirrors this message passing procedure of geometric GNNs. This alignment is critical as it allows us to precisely characterise the expressive power of geometric GNNs via GWL.
> > > >
> > > > That being said, we would like to look into this idea more deeply, because it is interesting and relevant - thank you for the pointer!
> > > >
> > > > At this late stage, however, we are unsure whether we can validate this idea and compare SVD-based GNNs to geometric GNNs in a timely and rigorous manner. We will try our best. As a final request, if you have any suggested experimental setup, we can try to follow it.
> > > >
> > > > ---
> > > >
> > > > **tl;dr** Geometric GNNs are an emerging research area and are driving impactful applications in drug discovery, biochemistry, material science, and engineering. This paper proposes a novel theoretical framework to advance our fundamental understanding of geometric GNNs. Our theoretical framework may not be the most efficient tool in a practical sense, but we believe it does offer interesting new insight and points the community towards a useful direction.

---

> > > > > ### Comment · Reviewer_XpRF · 2022-12-10
> > > > > **Clarification**
> > > > >
> > > > > How will this SVD approach work with covariant information attached to the nodes? In point clouds, this is standard pre-processing. But here we don't just have coordinates and we are not just working with point clouds. We want the attached features to be steerable. I am not sure I follow this. I would appreciate it if the authors or the reviewer can clarify how might be handled. I am currently a bit pressed for time to work this out.

---

> > > > > > ### Author Response · Authors · 2022-12-10
> > > > > > **We agree that more information is required**
> > > > > >
> > > > > > > How will this SVD approach work with covariant information attached to the nodes? In point clouds, this is standard pre-processing. But here we don't just have coordinates and we are not just working with point clouds. We want the attached features to be steerable. I am not sure I follow this. I would appreciate it if the authors or the reviewer can clarify how might be handled. I am currently a bit pressed for time to work this out.
> > > > > >
> > > > > > Thank you to R3 for your diligence in raising this counterpoint!
> > > > > >
> > > > > > We agree with you that R4's new proposition of SVD+GNN as an alternative to geometric GNNs and GWL is missing full details, and we would like to give them a chance to substantiate. In our opinion, we also agree that the proposed SVD+GNN **does not cover all the use cases of geometric graphs**, and this may explain why such approaches are not seen in biochemistry, material science, and engineering applications (to the best of our knowledge).
> > > > > >
> > > > > > From our perspective, we would like to add that:
> > > > > > 1. There could be one possible but cumbersome way of handling steerable vector features in the SVD-GNN (based on our understanding of the proposal). One could perform SVD independently for each input vector feature channel, but this may not be efficient depending on large we set the channel dimension.
> > > > > > 2. One other issue with SVD-GNN may be generalisation.
> > > > > >     - In preliminary experiments on QM9 molecular regression, we tried implementing a model where we do PCA on the centerred input node coordinates and pass them to a normal (non-geometric GNN). **Unfortunately, we could not get such an approach to train.** (This could also be due to coding errors/bugs).
> > > > > >     - Each geometric graph may have different principle components of variation and the ones observed during training could be different from those in the evaluation set. Essentially, the way we understand it is that the SVD based approach tries to pre-compute a canonical orientation of each geometric graph, but the types of orientations may vary across training and evaluation data. Instead, geometric GNNs take the view of localised one-hop environments around each node and perform geometric message passing to represent local geometry. Looking at geometry locally may be better from a generalisation/learnability perspective as compared to finding global canonical orientations.

---

> > > > > > > ### Comment · Reviewer_XpRF · 2022-12-11
> > > > > > > **Not sure**
> > > > > > >
> > > > > > > My impression is that the reviewer suggests standard alignment. I briefly tried to convert the paper of Brandstetter et al. (quoted in appendix B) into an SVD GNN -- I might be missing something, but I am not sure this can be done in a meaningful way and circumvent the need for steerable message passing there. Perhaps the reviewer can clarify.

---

> > > > > ### Author Response · Authors · 2022-12-11
> > > > > **Final day of phase 2 discussions; please let us know if your concerns have been addressed**
> > > > >
> > > > > Dear reviewer,
> > > > >
> > > > > We wanted to send another reminder that the discussion period for phase 2 is coming to an end today - **12 December 2022**.
> > > > >
> > > > > Please let us know if we have sufficiently addressed your concern.
> > > > >
> > > > > We would like to **request you to revisit your position** of strongly rejecting this work on the grounds of **lack of motivation**. We have tried to highlight evidence as to why our work and research area is interesting, significant and practically useful. This is supported by the recommendations of all the other reviewers, as well.
> > > > >
> > > > > Please also note that we have addressed all of your remaining concerns.
> > > > >
> > > > > Best regards,
> > > > >
> > > > > Authors

---

### Official Review · Reviewer_XpRF · 2022-10-25

**Confidence:** 5
**Correctness:** 3
**Technical Novelty And Significance:** 3
**Empirical Novelty And Significance:** 3
**Recommendation:** 8

**Clarity, Quality, Novelty And Reproducibility:**

The paper's presentation is top-notch and quite crisp. The motivation and the general formulation is quite clear, partly because of how well-motivated and almost ubiquitous the use of geometric GNNs now is. The results presented, to the best of my knowledge, are original and will serve to inform further development of practical geometric GNNs and the study of their theoretical properties.

**Strength And Weaknesses:**

- The paper has a very clear motivation, and attacks a problem of interest, given the recent successes of geometric GNNs in various key applications in the physical sciences.
- The formulation of the geometric WL is straightforward and clean, and also interleaves nicely with existing work using the WL test.
- The procedure proposed seems to indicate some directions for building provably powerful geometric GNNs, which are also practical. The current formulation is not exactly practical, and is kind of abstract.
- The connections between universality and expressivity discussed are also quite interesting.
- Lastly, the paper is quite well written and is a pleasure to read.

**Summary Of The Paper:**

The paper considers the problem of isomorphism, in the spirit of the WL test, but for geometric graphs. Such graphs have become prominent in recent times e.g. in applications in chemistry, and have seen notable empirical success. In addition to the usual combinatorial (graph) structure, the geometric nature of such graphs implies that GNNs designed to work on them must respect additional physical symmetries (rotation, reflection, translation). The usual WL test is not sufficient to cover such graphs, and a similar, but more general, geometric analog is needed. This paper fills the gap by proposing such a "geometric WL test", which can start to answer questions about the expressive power of geometric GNNs. The proposed procedure is quite straightforward, and the attendant theorems characterize some classes of geometric graphs that can be distinguished by the proposed test. Further, it also manages to quantify questions such as the tradeoffs between G-equivariant GNNs and G-invariant ones. Lastly, the paper also provides an equivalence between a model's ability to discriminate geometric graphs and their universality properties (vis-a-vis to approximate G-invariant functions).



**Summary Of The Review:**

See above.

---

> ### Author Response · Authors · 2022-11-13
> **Point-by-point response to Reviewer 3**
>
> >R3.1. The procedure proposed seems to indicate some directions for building provably powerful geometric GNNs, which are also practical. The current formulation is not exactly practical, and is kind of abstract.
>
> Thank you for encouraging us to better emphasise the practical implications of our work. Towards this end, we have included in **Section 4** an updated section discussing how to use GWL to understand the design space of geometric GNNs. We highlight theoretical limitations of current geometric GNNs in terms of depth (new results proving invariant GNNs cannot compute global geometric properties) and body order of local aggregation (updated presentation).
>
> Additionally, in **Appendix C**, we include synthetic experiments supplementing our results from Section 4 and highlighting practical challenges in building powerful geometric GNNs: oversmoothing and oversquashing with increased depth, as well as difficulty of designing efficient higher order aggregators. We are actively working on translating our theoretical insights into provably powerful and practical geometric GNNs.
>
> Lastly, we would like to state that the GWL test is indeed meant to serve as an **abstract mathematical tool** to uncover practical results about geometric GNNs, and not an efficiently implementable algorithm. Both GWL and WL assume perfect hash functions, which may require infinite space complexity in generality, for example if we used an infinite order spherical harmonic expansion.
>
> >R3.2. The paper's presentation is top-notch and quite crisp. The motivation and the general formulation is quite clear, partly because of how well-motivated and almost ubiquitous the use of geometric GNNs now is. The results presented, to the best of my knowledge, are original and will serve to inform further development of practical geometric GNNs and the study of their theoretical properties.
>
> Thank you very much for your positive comments championing our work! Thank you for your actionable feedback; do let us know if you have any further suggestions on improving the paper.

---

> > ### Comment · Reviewer_XpRF · 2022-11-14
> > **Thanks for the updates**
> >
> > I went over the responses to the other reviewers and the updates (thanks for highlighting them in different colours). I think these improve the paper and clarify some points further. Overall my assessment of the paper comes from the fact that it studies an important problem, and there is value in analogues of the WL criterion for geometric graphs -- which are becoming important. Besides, since the study of expressivity of such graphs is open there is value to the paper even if it seems, on the surface an extension of the usual WL criterion. I like the paper and would keep my rating. I appreciate your efforts in updating the article.

---

> > > ### Author Response · Authors · 2022-11-16
> > > **Thank you for the response**
> > >
> > > > Overall my assessment of the paper comes from the fact that it studies an important problem, and there is value in analogues of the WL criterion for geometric graphs -- which are becoming important.
> > >
> > > Thank you for acknowledging our revisions, and for your positive comments championing our work, once again!
> > >
> > > > Besides, since the study of expressivity of such graphs is open there is value to the paper even if it seems, on the surface an extension of the usual WL criterion. I like the paper and would keep my rating. I appreciate your efforts in updating the article.
> > >
> > > We agree that it is important to highlight why this work is relevant and what research gap it fills. We have attempted to do so through our revision as well as responses to the other reviewers, as you have already noted. (Thank you for reading them!)
> > >
> > > For completeness, we would like to reiterate once again why GWL for geometric graphs is a non-trivial extension of the usual WL test for normal graphs:
> > >
> > > 1. **Geometric features are more than simple feature vectors:** We agree that simply adding feature vectors to each node would constitute a straightforward extension to WL. However, in contrast to adding generic feature vectors (which are essentially lists of scalars), geometric features must satisfy a well-defined transformation behaviour under spatial symmetries to retain their physical meaning. This introduces a notion of “geometric isomorphism” to the problem, which interplays with the “graph isomorphism” notion in the standard WL setting.
> > > 2. **Standard WL is not enough to study the increasingly relevant geometric graph setting:** The standard WL framework has [recently been applied](https://arxiv.org/abs/2201.07136) to study geometric GNNs which are purely based on distance cut-offs (e.g. SchNet like models). Yet, it would not be possible to perform this analysis for more recent equivariant GNNs such as MACE or PaiNN, because they contain feature types with symmetries, something not treated in standard WL. Our framework extends WL in a way that allows us to reason about the full range of geometric GNN models from invariant SchNet and DimeNet to equivariant MACE and PaiNN, and therefore constitutes a useful extension.
> > >
> > > Thus, GWL fills a key research gap as theoretical tools for standard GNNs are not applicable to the increasingly relevant special case of geometric graphs and geometric GNNs. The GWL test is an abstract mathematical tool which we use to answer practical questions about the design space of geometric GNNs, e.g. What is the role of depth? How to design powerful local fingerprints/aggregators? Do we need higher order tensors? We hope to release our accompanying synthetic experiments and pedagogical resources for beginners and experts to explore the geometric GNN design space in a principled and theoretically grounded manner.

---

> > > > ### Comment · Reviewer_XpRF · 2022-12-10
> > > > **Thanks**
> > > >
> > > > Thanks for the detailed clarifications. I am going to raise my confidence score by one notch as I would like to at least make a point that the paper is decent enough to be published. I disagree with some assessments of other reviewers that this is "not well motivated" (geometric graphs are everywhere now and have a clear performance benefit compared to standard GNNs). The case (for the sake of argument) can indeed be made that geometric WL is not practically useful, but really, one could argue the original WL wasn't very useful either in that case. I in fact think much of the work on WL expressivity itself in the GNN literature is incomplete without connecting to learning -- expressivity alone isn't very useful -- but nevertheless, it is a focus of much attention. However, despite this, there is a gap in the literature, which this paper does make attempt to fill -- which can prove to be useful later. I think there is enough value in that. In my opinion, the case for rejecting this paper would somehow circle back to making a broader point about the usefulness of WL-based papers themselves. But I don't think this is the right place to do that.

---

> > > > > ### Author Response · Authors · 2022-12-10
> > > > > **Thank you for raising your confidence**
> > > > >
> > > > > > Thanks for the detailed clarifications. I am going to raise my confidence score by one notch as I would like to at least make a point that the paper is decent enough to be published.
> > > > >
> > > > > Thank you very much, as always, for your feedback and for raising your confidence score.
> > > > >
> > > > > > I disagree with some assessments of other reviewers that this is "not well motivated" (geometric graphs are everywhere now and have a clear performance benefit compared to standard GNNs).
> > > > >
> > > > > **Thank you for pointing this out very clearly.** We would like to note that R4's comments regarding geometric GNNs and GWL not being well motivated are in direct contradiction with the comments of yourself as well as R1 and R2.
> > > > >
> > > > > > The case (for the sake of argument) can indeed be made that geometric WL is not practically useful, but really, one could argue the original WL wasn't very useful either in that case. I in fact think much of the work on WL expressivity itself in the GNN literature is incomplete without connecting to learning -- expressivity alone isn't very useful -- but nevertheless, it is a focus of much attention. However, despite this, there is a gap in the literature, which this paper does make attempt to fill -- which can prove to be useful later. I think there is enough value in that. In my opinion, the case for rejecting this paper would somehow circle back to making a broader point about the usefulness of WL-based papers themselves. But I don't think this is the right place to do that.
> > > > >
> > > > > We like debating, so let us try to defend the line of research on GNN expressivity and WL, briefly.
> > > > >
> > > > > First of all, we are in complete agreement with you that **expressivity is only one layer of the painting that is theoretical understanding of GNNs** and deep learning architectures, in general. We agree that real-world generalisation is probably more important, practically. We are actively working on studying generalisation for geometric GNNs in future work.
> > > > >
> > > > > However, we believe that studying the expressivity of normal GNNs and geometric GNNs is an important first step in deeper theoretical understanding of our tools. GNN expressivity research from the WL lens is significant and practically relevant for two broad reasons:
> > > > >
> > > > > 1. Characterising the expressive power of architectures **tells us their maximum capabilities** or **best-case scenario** performance, as well as their theoretical limitations. In turn, this gives us a **recipe for improving our models**. For e.g.
> > > > >     - In our work, we show that rotation-invariant GNNs have theoretical limitations in not being able to compute non-local and global geometric properties. How to improve these models? Add non-local/global properties are pre-computed inputs to them, or work with a fully-connected graph. Such findings can lead to practical performance improvements and better geometric GNN design.
> > > > >     - The above point mirrors work in normal GNN expressivity: vanilla MPNNs are limited by 1-WL, and several successful proposals improve the theoretical capabilities and practical performance of MPNNs by higher order message passing ([SWL, CWL](http://proceedings.mlr.press/v139/bodnar21a.html)), multiple aggregators ([PNA](https://proceedings.neurips.cc/paper/2020/hash/99cad265a1768cc2dd013f0e740300ae-Abstract.html)), and positional encodings ([Graph Transformers](https://arxiv.org/abs/2012.09699)). These works are SOTA and impactful across a range of GNN benchmarks and applications (we can have a separate debate about GNN benchmarks).
> > > > >
> > > > > 2. Characterising the expressive power of architectures also gives us **basic design principles for architecture design** and helps **formalise architectural choices in a less hand-wavy manner**. For e.g.
> > > > >     - In our work, we can formally show that geometric GNNs can distinguish an increasing class of geometric graphs by: (a) increasing body order; (b) increasing tensor order; (c) increasing depth. Thus, we can formally characterise what makes a more powerful or maximally powerful geometric GNN, which is useful.
> > > > >     - The above point mirrors frameworks from normal GNNs, such as [k-GNNs](https://ojs.aaai.org/index.php/AAAI/article/view/4384) or [k-IGNs](https://proceedings.neurips.cc/paper/2019/hash/bb04af0f7ecaee4aae62035497da1387-Abstract.html), which help us reason about what basic design choices practitioners can make in order to build more powerful models in a principled manner, as opposed to throwing a zoo of models into a grid search.

---

> ### Author Response · Authors · 2022-12-07
> **Minor query on Correctness rating**
>
> > Correctness: 3: Some of the paper’s claims have minor issues. A few statements are not well-supported, or require small changes to be made correct.
>
> Dear Reviewer,
>
> We noticed that you rated the paper's Correctness as a '3' (although we felt that your review did not reflect this position). Please let us know if there are any remaining claims or statements that you feel have minor issues or require changes to be made correct. We would be happy to address them.
>
> Best regards,
>
> Authors

---

### Official Review · Reviewer_Kinq · 2022-10-30

**Confidence:** 4
**Correctness:** 4
**Technical Novelty And Significance:** 3
**Empirical Novelty And Significance:** Not applicable
**Recommendation:** 5

**Clarity, Quality, Novelty And Reproducibility:**

In my opinion, the novelty is limited (however it is not insignificant) - purely because the theorem statements and proofs in Section 4 are very similar to the ones in the k-WL analogue. Please see weakness 3 for statements corresponding to section 5.

No experiments - so reproducibility is not a cause of concern.

**Strength And Weaknesses:**

**Strengths**:
1. The idea to extrapolate the WL hierarchy to geometric graphs is important given the recent advances and interesting
2. The background work and the connections to recent works on geometric graphs is very well presented.

**Weakness, corresponding questions and suggestions:**
1. In my opinion, based on the statements on preserving the relative positions and using appropriate group actions $-$ the paper appears to incorrectly convey that all Lie groups preserve isometry - which is not the case. Also paper in current state cannot directly handle something like quadrapole-quadrapole interactions (e.g. Anderson, et al. 2019 - Comormant: Covariant molecular neural networks). Would suggest the authors rather use group representations rather than presenting them just as group actions (this would avoid any confusion to readers who may assume all lie groups are matrix lie groups for instance)
2. To me, it appears like the expressive power of IGWL appears to be misrepresented - am I misunderstanding something? For example, in Figure 2, the authors claim IGWL cannot distinguish the two geometric graphs - however the multiset (Eq 9) does include information about vectors - and a powerful and expressive multiset function, can allow for interactions between the vectors in the multiset from different neighbors - which can help distinguishing the graphs.This raises concern about the statements in section 4 - for example - Proposition 4, Theorem 6.
3. There appears to be a lack of a connection from moving from the statements in Section 4 to the statements in Section 5. For instance, until Section 5, the authors assume countable number of orbits -- which could be violated in Section 5 e.g. Theorem 21 (unless in cases for example where you explicitly show that the function is a proper map or via alternative mechanisms - currently I am unfortunately unable to locate a statement which conveys the same) and hence can't directly see its extension to the geometric graph application
4. Lack of experiments (please add at least on synthetic geometric graphs) which show the authors claims that equivariant layers are more powerful than invariant layers - all else remaining constant.
5. Some of the lemmas and theorem statements in the main paper and appendix related to section 5 - are well known results available in most differential geometry books (e.g. John Lee's Smooth Manifolds) - Would suggest the authors clearly indicate this)

**Summary Of The Paper:**

In this work, the authors propose a hierarchy (analogous to the WL hierarchy for graphs) $-$ for graphs with geometric features (Typically node features with coordinate information, velocity, etc) which can then be used to characterize the expressive power of equivariant and invariant geometric graph neural network layers to distinguish geometric graphs. The authors then subsequently provide theoretical statements - which link the discriminative power of models on geometric graphs to its universality.

**Summary Of The Review:**

In current state, the weaknesses associated with the paper out-weigh the strengths. However, I would be happy to increase the scores if the authors provide appropriate clarifications.


**Updates:**

27th Nov:

Thank you very much for the detailed clarifications to reviewers questions as well as the extensive updates to the paper based on reviewer suggestions. I will increase my score to an 6 and would like to see the paper accepted.


After reading through the other reviews, I have one follow up question regarding the motivation - and would be happy to increase my score further.

Consider the case where geometric graphs are not rigid - for example molecules typically tend to have multiple conformations - where the relative positions of atoms in the molecule change -- and potentially a different contact graph (say based on 3nm distances) is obtained . In such a case, combining 1WL with equivariant/ invariant layers could be too restrictive -- i.e. different conformers of the same molecule would end up with different representations (while ideally we want them in the same equivalence class). Given, the above scenario, why do we need to study the expressive power of 1WL with Lie group equivariant/ invariant layers?


11th Dec:
After discussions with other reviewers, reducing score to 5 (From 6)

---

> ### Author Response · Authors · 2022-11-13
> **(Part 1/3) Point-by-point response to Reviewer 2**
>
> >R2.1. In my opinion, based on the statements on preserving the relative positions and using appropriate group actions − the paper appears to incorrectly convey that all Lie groups preserve isometry - which is not the case…Would suggest the authors rather use group representations rather than presenting them just as group actions (this would avoid any confusion to readers who may assume all lie groups are matrix lie groups for instance)
>
> Thank you for highlighting the lack of clarity around which Lie groups we work with and the use of group actions vs. group representations. We entirely agree that all Lie groups do not preserve isometry. We have addressed this notational issue in the updated manuscript by stating that $\mathfrak{G}$ denotes the Lie group of **either rotations** or **rotations and reflections**, only.
>
> To clarify, we work with group actions $\mathfrak{g} \in \mathfrak{G}$ as well as their group representations $\mathbf{Q}_\mathfrak{g}$, using the appropriate notation depending on the context. For e.g. consider eq.2 or the un-numbered equation between eq.6 and eq.7 - the group action $\mathfrak{g}$ acts on the geometric object $\mathbf{g}_i$ via the group representation $\mathbf{Q}_\mathfrak{g}$ transforming the vector quantities.
>
> Please refer to the revised manuscript for more, we highlighted the main changes to addressing these points with the boxes labelled “R2.1”.
>
> >R2.2. Also paper in current state cannot directly handle something like quadrapole-quadrapole interactions (e.g. Anderson, et al. 2019 - Comormant: Covariant molecular neural networks).
>
> Without loss of generality, and for retaining a simple presentation, we have currently worked with geometric graphs having a single vector feature per node. Our results generalise to multiple vector features or **higher-order geometric tensors** per node.
>
> This works because GWL injectively aggregates geometric information into the $\mathbf{g}_i$ object as shown in eq.5 and eq.6. Thus, using spherical tensors of higher order as the choice of basis for the vector features would allow GWL to handle quadrapole-quadrapole interactions. In this case, we would replace the matrix group representation $\mathbf{Q}_\mathfrak{g}$ with a more generic $\rho(\mathfrak{g})$. We highlighted the main changes to addressing this point with the boxes labelled “R2.2”.
>
> To make it clear what geometric GNNs are covered under the GWL framework, in **Appendix B**, we provide more detailed background and unified equations for invariant GNNs (SchNet and DimeNet), equivariant GNNs using cartesian vectors (PaiNN, E-GNN, GVP-GNN), and equivariant GNNs using spherical tensors (TFN, Cormorant, MACE). We hope this addresses your concerns, together with R2.1.

---

> > ### Author Response · Authors · 2022-11-13
> > **(Part 2/3) Point-by-point response to Reviewer 2**
> >
> > >R2.3. To me, it appears like the expressive power of IGWL appears to be misrepresented - am I misunderstanding something? For example, in Figure 2, the authors claim IGWL cannot distinguish the two geometric graphs - however the multiset (Eq 9) does include information about vectors - and a powerful and expressive multiset function, can allow for interactions between the vectors in the multiset from different neighbors - which can help distinguishing the graphs. This raises concern about the statements in section 4 - for example - Proposition 4, Theorem 6.
> >
> > We agree that we should be more precise here; thank you for bringing this up.
> >
> > Firstly, note that the node colouring function $\text{I-HASH}$ in IGWL is G-orbit injective and G-invariant. $\text{I-HASH}$ distinguishes $\mathfrak{G}$-orbits of neighbourhoods, i.e. it is a neighbourhood fingerprint which returns the same colour for rotated versions of the same neighbourhood. As you correctly noted, $\text{I-HASH}$ does allow for interactions between the vector quantities in the input multiset. But it is **axiomatically restricted** to performing a scalarisation of these vectors. In geometric GNNs, this is termed the body order of scalarisation, e.g. distances from central node - neighbour are 2 body, angles among triplets of neighbour - central node - neighbour are 3 body, and so on.
> >
> > Now, focus on only the one hop neighbourhoods around each node and its corresponding bijection in the running example. For each neighbourhood, we can superimpose onto it the bijection via some rotation. E.g. $l$ and $b(l)$: 90°, $j$ and $b(j)$: 135°, $i$ and $b(i)$: 0°. Thus, all neighbourhoods have the same set of $\mathfrak{G}$-invariant scalars as their bijection (in this case, the set of distances and angles, as the maximum possible body order is 3). As a consequence, the $\text{I-HASH}$ function will assign the same colour to each node and its bijection.
> >
> > Finally, note that IGWL does not propagate geometric quantities beyond the one hop neighbourhood. Thus, each iteration repeatedly assigns the same colours to each node as local scalarisations are the same. As a consequence, IGWL cannot distinguish the two graphs.
> >
> > An alternative way to concisely present this is via **geometric computation trees**. IGWL cannot distinguish the two graphs as all one hop neighbourhoods are computationally identical, i.e. output the same scalars. We have elaborated on how computation trees are constructed and discussed their consequences in the updated Section 4.
> >
> > Note that this simple running example is specific to $\mathfrak{G} = O(d)$ so cross products are not permitted. The two chains can be distinguished by  $SO(d)$-IGWL as the sign of the cross products at the two ends would differ. It remains possible to construct similar counterexamples for $SO(d)$ based on the work of Pozdnyakov et al. We explored the practical implications of these counterexamples in the synthetic experiment in Table 1 of **Appendix C**.
> >
> > >R2.4. There appears to be a lack of a connection from moving from the statements in Section 4 to the statements in Section 5. For instance, until Section 5, the authors assume countable number of orbits -- which could be violated in Section 5 e.g. Theorem 21 (unless in cases for example where you explicitly show that the function is a proper map or via alternative mechanisms - currently I am unfortunately unable to locate a statement which conveys the same) and hence can't directly see its extension to the geometric graph application
> >
> > Thank you for bringing up the connection between the discrimination-based perspective of GWL and the section on universal approximation. Firstly, in the Related Work section, we have further emphasised their relationship as two theoretical tools to study geometric GNNs. Discrimination is a potentially **more granular lens** than universality. A model may either be universal or not. On the other hand, there could be multiple degrees of discrimination depending on the classes of geometric graphs that can and cannot be distinguished.
> >
> > Additionally, we would like to note that results on universal approximation in Section 5 considers two settings: **the countable case** like in GWL, and subsequently, **the continuous case**. Importantly, the notion of discriminating two geometric graphs is weaker than the one used in GWL (discriminating **all** non-isomorphic graphs). We have highlighted statements emphasising this with the boxes labelled “R2.4”.
> >
> > In Theorems 21-23, we then reconnect the continuous setting with injective or perfect aggregation assumed in GWL. We provide lower bounds on the representational width needed to define continuous injective functions for identifying all unique neighbourhoods up to group actions, i.e. **neighbourhood fingerprints**. Such lower bounds are important because, while GWL is an abstract mathematical tool operating in countable settings, real-world geometric GNN models operate on continuous feature spaces.

---

> > > ### Author Response · Authors · 2022-11-13
> > > **(Part 3/3) Point-by-point response to Reviewer 2**
> > >
> > > >R2.5. Lack of experiments (please add at least on synthetic geometric graphs) which show the authors claims that equivariant layers are more powerful than invariant layers - all else remaining constant.
> > >
> > > We agree that experiments supporting our theoretical results would significantly strengthen our message. Towards this end, we have included in **Section 4** an updated section discussing how to use GWL to understand the design space of geometric GNNs. Additionally, in **Appendix C**, we include synthetic experiments supplementing our results from Section 4.
> > >
> > > Both theory and synthetic experiments now point to clear limitations of invariant GNNs. This helps us **highlight practical challenges** in building powerful geometric GNNs, such as oversmoothing and oversquashing with increased depth, as well as designing efficient higher order aggregators. These synthetic experiments have led to exciting avenues for future work towards building provably powerful, practical geometric GNNs – thank you for your suggestion!
> > >
> > > >R2.6. Some of the lemmas and theorem statements in the main paper and appendix related to section 5 - are well known results available in most differential geometry books (e.g. John Lee's Smooth Manifolds) - Would suggest the authors clearly indicate this)
> > >
> > > Thank you for bringing this to our notice; this was unintended. We highlight this in the revised version of the manuscript and reference classic results (e.g. the Quotient Manifold Theorem) in the proof. Please let us know if we missed something.
> > >
> > > >R2.7. In my opinion, the novelty is limited (however it is not insignificant) - purely because the theorem statements and proofs in Section 4 are very similar to the ones in the k-WL analogue.
> > >
> > > Thank you for this critical assessment. We would like to reiterate our main novelty and contributions briefly. We have also revised the Abstract and Introduction to better highlight why our work is significant:
> > > 1. This is the **first work to study the expressive power of geometric GNNs** from the perspective of geometric graph isomorphism. We believe this is non-trivial because geometric graphs come with both relational structure and geometric attributes. Our work fills a key research gap as theoretical tools for standard GNNs are not applicable to the increasingly relevant special case of geometric graphs due to the stronger notion of physical symmetries and geometric isomorphism.
> > > 2. Our main contribution is a simple yet precise definition of a Geometric WL test, which is a theoretical upper bound on the expressive power of geometric GNNs. We use GWL to provide novel insights into the workings and theoretical limitations of geometric GNNs.
> > > 3. We study the design space of geometric GNNs through the lens of GWL and highlight practical challenges in building probably powerful models. We supplement our theory with new synthetic experiments which can be a pedagogical tool for the community.
> > > 4. Finally, we connect our discrimination-based perspective in GWL to previously studied universal approximation capabilities of geometric GNNs. This is important because discrimination is a more granular lens into geometric GNNs than universality.
> > >
> > > We agree that the results in Section 3 (revised presentation) seem easy to derive from the definitions. This may be particularly evident for the statements and proof techniques in Section 3.2 (revised presentation) which largely follow Xu et al., Morris et al. We believe these are the **fruits of proposing a suitable formalisation** and **framework**, including the GWL test itself as well as the notion of $k$-hop distinct and $k$-hop identical geometric sub-graphs. The results are nonetheless significant because they help understand the workings and theoretical limitations of the two classes of geometric GNNs. We have expanded on this in Section 4 on probing the geometric GNN design space via GWL, as well as synthetic experiments highlighting practical challenges in Appendix E.
> > >
> > > Lastly, do note that GWL is a generalisation of **1-WL** and is different from the k-WL hierarchy of tests for standard graph isomorphism. k-WL maintains colourings for and propagates information among all **k-tuples** of nodes, including non-local tuples. This is not directly possible for geometric graphs as the geometric attributes correspond to real-world physical quantities such as velocity, acceleration, etc. Thus, there is no notion of geometric features associated with (non-local) k-tuples. Such a generalisation could be an interesting future extension of GWL.
> > >
> > > >R2.8. In current state, the weaknesses associated with the paper out-weigh the strengths. However, I would be happy to increase the scores if the authors provide appropriate clarifications.
> > >
> > > Thank you for your actionable feedback which helped significantly improve the manuscript. We believe we have addressed all the concerns raised. Please let us know if you are satisfied with our responses and have any further queries that we can address.

---

> > > > ### Author Response · Authors · 2022-11-16
> > > > **Quick update on R2.6**
> > > >
> > > > >R2.6. Some of the lemmas and theorem statements in the main paper and appendix related to section 5 - are well known results available in most differential geometry books (e.g. John Lee's Smooth Manifolds) - Would suggest the authors clearly indicate this)
> > > >
> > > > Thank you for bringing this to our notice; this was completely unintended. We have remedied this in the updated revision and cited appropriately. In particular, we have stated that "We now produce an estimate for the number of aggregators needed to learn continous orbit-space injective functions on a manifold X **based on results from differential geometry (Lee, 2013)**." We have also updated the appendix with the same.
> > > >
> > > > Please let us know if there is anything we have missed - we will immediately remedy this and make appropriate citations.

---

> > > > > ### Comment · Reviewer_Kinq · 2022-11-28
> > > > > **Response to authors**
> > > > >
> > > > > Dear Authors,
> > > > >
> > > > > Thank you very much for the detailed clarifications to reviewers questions as well as the extensive updates to the paper based on reviewer suggestions. I will increase my score to an 6 and would like to see the paper accepted.
> > > > >
> > > > >
> > > > > After reading through the other reviews, I have one follow up question regarding the motivation - and would be happy to increase my score further.
> > > > >
> > > > > Consider the case where geometric graphs are not rigid - for example molecules typically tend to have multiple conformations - where the relative positions of atoms in the molecule change -- and potentially a different contact graph (say based on 3nm distances) is obtained . In such a case, combining 1WL with equivariant/ invariant layers could be too restrictive -- i.e. different conformers of the same molecule would end up with different representations (while ideally we want them in the same equivalence class). Given, the above scenario, why do we need to study the expressive power of 1WL with Lie group equivariant/ invariant layers?

---

> ### Author Response · Authors · 2022-11-16
> **We are happy to discuss further**
>
> Dear reviewer, thank you very much once again for taking the time to review our manuscript and for your actionable feedback so far.
>
> We have worked hard to address the concerns raised by the original review in the revised manuscript, including better highlighting our novelty as well as adding synthetic experiments and strengthening connections with practice. We have clarified the concerns around group actions/representations, geometric GNNs covered under our framework, capabilities of IGWL, and connections between GWL and universal approximation. We have also added the requested citations.
>
> As we are nearing the Reviewer-Author Discussions deadline (18 November), please let us know if you are satisfied with our responses and have any further queries that we can address. If we have addressed your concerns, please consider revising your score.

---

> ### Author Response · Authors · 2022-11-30
> **(Part 1/2) Post-rebuttal response to Reviewer 2**
>
> > Thank you very much for the detailed clarifications to reviewers questions as well as the extensive updates to the paper based on reviewer suggestions. I will increase my score to an 6 and would like to see the paper accepted.
>
> Thank you once again for your actionable feedback and for acknowledging our revisions. We are glad to read your positive comment supporting our work!
>
> > After reading through the other reviews, I have one follow up question regarding the motivation - and would be happy to increase my score further.
>
> We have provided a detailed response below. Please let us know if you are satisfied with our response and have any further queries that we would be happy to discuss. We will update our manuscript based on your question at the earliest opportunity. For e.g., through preparing this response, we can now elaborate better on why geometric graph isomorphism is practically relevant when we introduce its definition in the Background in Section 2 (page 3).
>
> > Consider the case where geometric graphs are not rigid - for example molecules typically tend to have multiple conformations - where the relative positions of atoms in the molecule change -- and potentially a different contact graph (say based on 3nm distances) is obtained . In such a case, combining 1WL with equivariant/ invariant layers could be too restrictive -- i.e. **different conformers of the same molecule** would end up with **different representations** (while ideally we want them in the same equivalence class). Given the above scenario, why do we need to study the expressive power of 1WL with Lie group equivariant/ invariant layers?
>
> Our main motivation for proposing the Geometric WL test is as an **abstract theoretical tool** to study the expressive power and representational limits of geometric GNNs, a class of **practical GNN models** specialised for geometric graphs with [rigid symmetries](https://en.wikipedia.org/wiki/Rigid_transformation) (rotations, translations, reflections). These architectures have shown impactful real-world applications in [drug discovery](https://www.science.org/doi/full/10.1126/science.abe5650), [molecular simulations](https://www.nature.com/articles/s41467-022-29939-5), [robotics](https://ieeexplore.ieee.org/abstract/document/9341668/), etc.
>
> Let us elaborate on our motivation in the context of the scenario described by the reviewer, discussing how the GWL framework and geometric GNNs are relevant for the case of both rigid as well as non-rigid symmetries.

---

> > ### Author Response · Authors · 2022-11-30
> > **(Part 2/2) Post-rebuttal response to Reviewer 2**
> >
> > ## Non-rigid and local symmetries
> > While our work focuses on geometric graphs with rigid symmetries, we agree that additionally considering non-rigid symmetries is very interesting and practically relevant. To the best of our knowledge, and as noted by the reviewer, it is not possible to model non-rigid symmetries and local flexibility as Lie groups or as exact symmetries in the first place. One possible approach would be to consider approximate symmetries (e.g. distortion) in a similar fashion as described in the [Geometric Deep Learning](https://arxiv.org/abs/2104.13478) textbook.
> >
> > Our current setup for geometric GNNs and GWL is based on seminal works such as [SchNet](https://aip.scitation.org/doi/10.1063/1.5019779) and [Tensor Field Networks](https://arxiv.org/abs/1802.08219). Thus, the rigid symmetries are enforced by the equivariant or invariant architecture, but the "flexibility symmetries" are learned from data approximately, because it is hard to formalise them. This does not detract from the significance of GWL or of this line of work, though – it just highlights an exciting area of further research on geometric graph and molecular modelling.
> >
> > Next, we describe how conformers and structurally similar molecules are handled in our setup.
> >
> > ## Rigid symmetries and conformers
> > In the case of molecular geometric graphs with rigid symmetries, GWL can distinguish any unique geometric graphs, as stated in Proposition 5. Essentially, GWL is an injective or one-to-one input preserving function. The maximally expressive geometric GNN model, i.e. one with the same expressive power as GWL, can thus map each unique input in a given dataset of geometric graphs to a unique output representation.
> >
> > In practical molecular applications, as **input**, we are often given different conformers of the same molecule, or molecules that may be structurally similar but functionally different. The desired **output** in such cases is highly application dependent.
> >
> > Consider three cases of significance in drug discovery:
> > 1. Molecular property prediction for high-throughput virtual screening, e.g. in-vivo toxicity or antibiotic activity of potential drug targets ([Bender-Glen, 2014](https://pubs.rsc.org/en/content/articlehtml/2004/ob/b409813g)). In this case, the **output for different conformers** is usually **the same**, i.e. the model should map them to the same functional properties or equivalence class.
> > 2. Potential energy prediction for molecular dynamics simulations, e.g. given a molecular geometry, predict the overall energy or per-atom forces which are used to simulate the molecules trajectory ([Bartok-Kondor-Csanyi, 2013](https://journals.aps.org/prb/abstract/10.1103/PhysRevB.87.184115)). In this case, the **output for different conformers** will be **different**, i.e. the model should map them to unique output energies or forces.
> > 3. Binding affinity prediction, e.g. given an antibody and a protein binding site such as the sars-cov2 spike receptor, predict their binding. In this case, the **output for different conformers** will usually be **different**, as some antibodies can only bind to spike receptors in one particular conformation (called the *closed* conformer) and not in the *open* ones.
> >
> > Critically, the ideal model for cases 2 and 3 would already encapsulate case 1, i.e. if our model can do well on cases 2 and 3, we would expect it to also do well on case 1. If we have an injective, input preserving map to an intermediate representation space (as in cases 2 and 3), we can then map from these intermediate representations to any other output space (e.g. any desired output space in case 1).
> >
> > This line of reasoning is also the basis of seminal work on the expressive power of neural networks for sets ([DeepSets](https://arxiv.org/abs/1703.06114)) as well as standard non-geometric graphs ([GIN](https://openreview.net/forum?id=ryGs6iA5Km)). In essence, if your model can injectively map unique inputs to unique outputs, in theory, it can then learn any other practically relevant and domain-specific output map on top of the output of this injective map.
> >
> > The interesting question that your comment raises is whether such an injective, input preserving model is possible? If it is possible, how will it generalise to out-of-distribution input? Is there a tradeoff between expressivity, which allows you to perfectly fit training data, and generalisation to unseen data? We are excited to explore the generalisation properties of geometric GNNs in future work.

---

> > > ### Author Response · Authors · 2022-12-06
> > > **Update on post-rebuttal response**
> > >
> > > Dear reviewer,
> > >
> > > Please let us know if our response sufficiently addresses your concern regarding the motivation for our work when considering non-rigid symmetries.
> > >
> > > As a brief summary, our main motivation is to study the expressive power of GNNs for geometric graphs with **rigid symmetries**, which are practically important in drug discovery, molecular modelling, robotics, etc. To the best of our knowledge, it is not yet possible to formulate non-rigid symmetries as Lie groups (or it has not been done in deep learning literature yet). This means that the geometric GNNs under consideration will only be able to *approximately* learn non-rigid symmetries from data, while the rigid symmetries are 'baked in' to the architecture in an *exact* manner.
> > >
> > > In the specific case of different conformers of the same molecule, we may or may not want to map different conformers to the same output, **depending on the task**. GWL is a theoretical tool which can discriminate any two unique molecular geometric graphs -- GWL and the corresponding geometric GNNs will always map different conformers to different outputs. Thus, GWL studies idealised, **injective, input-preserving** geometric GNNs which can be further composed with any task-specific output transformation.
> > >
> > > Best regards,
> > >
> > > Authors

---

> > > > ### Author Response · Authors · 2022-12-09
> > > > **Gentle reminder**
> > > >
> > > > Dear reviewer,
> > > >
> > > > We wanted to send a gentle reminder about this matter. Please let us know if our response sufficiently addresses your concern regarding the motivation for geometric GNNs and GWL when considering non-rigid symmetries.
> > > >
> > > > Have a great weekend ahead,
> > > >
> > > > Authors

---

> > > > > ### Author Response · Authors · 2022-12-11
> > > > > **Final day of phase 2 discussions; please let us know if your concerns have been addressed**
> > > > >
> > > > > Dear reviewer,
> > > > >
> > > > > We wanted to send another reminder that the discussion period for phase 2 is coming to an end today - **12 December 2022**.
> > > > >
> > > > > Please let us know if we have sufficiently addressed your concern. Briefly, non-rigid symmetries have **not been studied** in geometric GNN literature to the best of our knowledge, and no models can handle them exactly at present. We have elaborated on how expressive geometric GNNs and GWL are **still relevant** when considering non-rigid symmetries and molecular conformers, nonetheless.
> > > > >
> > > > > Please let us know if you have any further queries that we would be happy to address. If we have addressed your concerns, please consider revising your score, as originally promised.
> > > > >
> > > > > Best regards,
> > > > >
> > > > > Authors

---

> > > > > > ### Comment · Reviewer_Kinq · 2022-12-11
> > > > > > **Update**
> > > > > >
> > > > > > Dear Authors,
> > > > > >
> > > > > > Thank you very much for the response and presenting information about non rigid symmetries.
> > > > > > However, after discussions with other reviewers, I am reducing my score to a 5 (from 6)

---

> > > > > > > ### Author Response · Authors · 2022-12-12
> > > > > > > **Could you elaborate/provide feedback**
> > > > > > >
> > > > > > > Thank you for your consideration.
> > > > > > >
> > > > > > > We have worked very hard to address all the concerns raised in your original review (thank you for the actionable comments, which have improved our paper considerably), and tried our best to initiate discussions throughout the past weeks.
> > > > > > >
> > > > > > > We find this negative change on the final day to be very disappointing, since you had already acknowledged that we have addressed all your concerns. Additionally, you had stated that you would like to see the paper accepted and would be willing to increase the score further.
> > > > > > >
> > > > > > > **Could you please elaborate on your concerns based on which you have changed your recommendation to rejection?**
> > > > > > >
> > > > > > > We will try our very best to address them as soon as we can.

---

> > > > > > > > ### Comment · Reviewer_Kinq · 2022-12-12
> > > > > > > > **Response**
> > > > > > > >
> > > > > > > > Dear Authors,
> > > > > > > >
> > > > > > > > Thank you again for your response and all the efforts. I am really sorry for the change, extremely late in the process - but I believe this change is more reflective of the my score, post discussions with other reviewers
> > > > > > > >
> > > > > > > > Some of the points which resulted in my change:
> > > > > > > >
> > > > > > > > Pros:
> > > > > > > > 1. Are geometric graphs important - absolutely.
> > > > > > > > 2. Presentation of the paper
> > > > > > > >
> > > > > > > > Cons:
> > > > > > > > 1. Is WL the right choice/ procedure to study expressivity for geometric graphs?
> > > > > > > > 2. Incremental nature of theoretical contribution
> > > > > > > > 3. Concerns raised via comparison with SVD - theoretically and empirically [appears to have concerns about steer-ability not possible]

---

> > > > > > > > > ### Author Response · Authors · 2022-12-12
> > > > > > > > > **Please reconsider your position, especially the last point which has not been substantiated upon**
> > > > > > > > >
> > > > > > > > > Dear Reviewer,
> > > > > > > > >
> > > > > > > > > > Is WL the right choice/ procedure to study expressivity for geometric graphs?
> > > > > > > > >
> > > > > > > > > We would like to request you to please judge the paper on the precise claims. Here are our precise claims:
> > > > > > > > > - The graph isomorphism problem and WL are **one possible tool** to study normal GNN expressivity. Arguably, WL has proven to be the leading theoretical tool for normal GNNs, leading to practical insights. (We have emphasised this in our response to R3, as well: https://openreview.net/forum?id=Rkxj1GXn9_&noteId=sNH9EtGCeXs.)
> > > > > > > > > - Geometric GNNs are becoming important, but **no theoretical tools exist** to study their expressive power and understand their workings.
> > > > > > > > > - In this paper, we have attempted to generalise graph isomorphism to geometric graph isomophism, and WL to Geometric WL. We have highlighted why this is **novel, non-trivial, fills an important research gap,** and **leads to interesting practical insights** for geometric GNN design.
> > > > > > > > >
> > > > > > > > > We feel that your concern on whether WL is the 'right' procedure is subjective. There could be other, potentially better, measures of geometric GNN expressivity.
> > > > > > > > >
> > > > > > > > > However, our goal was to point the community in what we hope was **one useful direction**.
> > > > > > > > >
> > > > > > > > > ---
> > > > > > > > >
> > > > > > > > > > Incremental nature of theoretical contribution
> > > > > > > > >
> > > > > > > > > We respectfully disagree with this position.
> > > > > > > > >
> > > > > > > > > - We proposed to study geometric GNN expressivity through the lens of graph isomorphism for geometric graphs. To the best of our knowledge, this is the **first definition of geometric graph isomorphism** with node coordinates and steerable attributes, at least in the context of machine learning.
> > > > > > > > > - We introduce the Geometric WL framework, a **novel theoretical tool** and upper bound on the expressive power of geometric GNNs. GWL **fills a key research gap**, as the standard theoretical toolkit for normal non-geometric GNNs is not applicable to the special case of geometric graphs.
> > > > > > > > > - We use GWL to provide **novel insights into geometric GNN's** workings and theoretical limitations. Most notably, we formalise the advantages of equivariant GNNs over invariant GNNs, and provide practical implications for geometric GNN design choices.
> > > > > > > > > - We connect the discrimination based perspective of GWL to universal approximation properties of geometric GNNs. We further study the minimal number of aggregators that are required to distinguish geometric neighbourhoods and thus **generalise results from PNA** to a geometric setting.
> > > > > > > > >
> > > > > > > > > If you still feel our contributions are incremental, it would help us improve the paper in future editions if you could substantiate this.
> > > > > > > > >
> > > > > > > > > If you concern is related to R1's concern that our results building upon Xu et al., Morris et al., Chen et al. -- we have acknowledged these papers multiple times, and do generalise ideas from these seminal papers on normal GNNs to the realm of geometric GNNs. **We feel that such generalisations are useful and help build connections to prior work.** As we study the connection between geometric graph isomorphism and geometric GNNs, it is inevitable that we will build upon seminal ideas connecting normal graph isomorphism to normal GNNs. We feel this should not be held against us.
> > > > > > > > >
> > > > > > > > > ---
> > > > > > > > >
> > > > > > > > > > Concerns raised via comparison with SVD - theoretically and empirically [appears to have concerns about steer-ability not possible]
> > > > > > > > >
> > > > > > > > > We feel that this concern is completely unsubstantiated by R4:
> > > > > > > > > - The SVD-followed-by-GNN approach cannot handle steerable features, as noted by R3.
> > > > > > > > > - The SVD-based approach does not enable us to study the expressive power of geometric GNNs in the same way as GWL. GWL mirrors this message passing procedure of geometric GNNs. This alignment is critical as it allows us to precisely upper bound the expressive power of geometric GNNs via GWL.
> > > > > > > > > - We tried implementing a version of SVD-GNN on QM9, and found that this approach has issues generalising.
> > > > > > > > >
> > > > > > > > > Please note that, to the best of our knowledge, nobody is using such SVD-based models in place of geometric GNNs. **If no such models are in used in practice, it certainly does not make sense to study their expressive power.**
> > > > > > > > >
> > > > > > > > > The goal of this paper is to study the expressive power of geometric GNNs based on invariant/equivariant message passing. **As long as there is a community using geometric GNNs, we feel that our paper is useful to that community**, regardless of whether an SVD-GNN approach may or may not be more suitable. It is not our goal to invent new GNNs for geometric graphs.
> > > > > > > > >
> > > > > > > > > Lastly, please consider the fact that this concern was raised in the final days and was not part of the original concerns of R4 (which we addressed fully). **R4 has not responded, elaborated on this matter, or provided any references to papers using the approach described, either.** Without substantial details from R4 and without being able to update our manuscript, we feel that it is rather unfair that this unsubstantiated point is used as a grounds for rejecting our work.
> > > > > > > > >
> > > > > > > > > Please reconsider your position.
> > > > > > > > >
> > > > > > > > > Best regards,
> > > > > > > > >
> > > > > > > > > Authors

---

### Official Review · Reviewer_Vdrv · 2022-11-02

**Confidence:** 4
**Correctness:** 4
**Technical Novelty And Significance:** 2
**Empirical Novelty And Significance:** 2
**Recommendation:** 5

**Clarity, Quality, Novelty And Reproducibility:**

The clarity is good.

The quality and novelty are limited as discussed above.

**Strength And Weaknesses:**

Strengths:
1. Learning on geometric graphs becomes more and more significant these days, and theoretical analysis of geometric GNNs is somehow limited. This work picks a good field.
2. The presentation is clear and easy to follow. The examples provided in Figures 1, 2 are good to show the gap between equivariant and invariant networks.
3. The results about numbers of aggregators in Theorem 21-23 are good, as an extension of previous results in PNA.

Concerns:
1. The major concern is a relative lack of novelty and contribution. Since geometric graphs are generic graphs equipped with additional node coordinates, the definitions of GWL and IGWL are very straightforward (which is not negative). But results in Section 4.1 on characterizing the gap between GWL and IGWL are too easy to obtain, including Prop 2, 3, 4, 7 and Theorem 6. Also, the results in Section 4.2 on the correspondence between GNNs and (I)GWL are too straightforward.
2. Theorem 21-23 build results on lower bounds of numbers of aggregators for injective functions. It would be great if it discusses whether or not such numbers are linked with the number of layers. In my mind, oversquashing [1]  might be interesting if its discussion can be generalized to geometric graphs. From a broader view, since this work is to generalize theoretical discussion on generic graphs to geometric ones, it would be better to develop a bigger story including richer contents, such as oversmoothing and oversquashing, rather than expressive power only.

Reference:
[1] Understanding over-squashing and bottlenecks on graphs via curvature.

**Summary Of The Paper:**

This work conducts an analysis of expressive power of geometric graph neural networks. Analogous to WL test for generic GNNs, this work proposes Geometric WL (GWL) test and its constraint version IGWL for geometric graphs. It characterizes an expressive-power gap between GWL and IGWL, which is then accordingly extended to a gap between equivariant and invariant GNNs.

**Summary Of The Review:**

I would like to recommend rejection. I am willing to raise my score if the above concerns are resolved.


---

The score is raised from 3 to 5.

---

> ### Author Response · Authors · 2022-11-13
> **(Part 1/2) Point-by-point response to Reviewer 1**
>
> > R1.1. The major concern is a relative lack of novelty and contribution.
>
> Thank you for raising this concern. We would like to reiterate our main novelty and contributions briefly. We have also revised the Abstract and Introduction to better highlight why our work is significant and relevant to the community:
> 1. This is the **first work to study the expressive power of geometric GNNs** from the perspective of graph isomorphism. We believe this is non-trivial because geometric graphs come with both relational structure and geometric attributes. Our work fills a key research gap as theoretical tools for standard GNNs are not applicable to the increasingly relevant special case of geometric graphs due to the stronger notion of physical symmetries and geometric isomorphism.
> 2. Our main contribution is a simple yet precise definition of a Geometric WL test, which is a theoretical upper bound on the expressive power of geometric GNNs. We use GWL to provide novel insights into the workings and theoretical limitations of geometric GNNs.
> 3. We study the design space of geometric GNNs through the lens of GWL and highlight practical challenges in building probably powerful models. We supplement our theory with new synthetic experiments which can be a pedagogical tool for the community.
> 4. Finally, we connect our discrimination-based perspective in GWL to previously studied universal approximation capabilities of geometric GNNs. This is important because discrimination is a more granular and practically insightful lens into geometric GNNs than universality. A model may either be universal or not. On the other hand, we show how there can be multiple degrees of discrimination.
>
> We agree with the reviewer’s implicit sentiment that there is space for deeper studies, and it is something we are excited to continue working on. We would like to highlight that the purpose of our paper was not to solve all questions, but to point the community into a (we hope) useful direction towards theoretically principled GNN models for applications in biochemistry, material science, and multiagent robotics.
>
> > R1.2. Since geometric graphs are generic graphs equipped with additional node coordinates, the definitions of GWL and IGWL are very straightforward (which is not negative). But results in Section 4.1 on characterizing the gap between GWL and IGWL are too easy to obtain, including Prop 2, 3, 4, 7 and Theorem 6.
>
> Thank you for this critical assessment. While GWL adds geometric features to WL and extends it in this sense, we would like to respectfully disagree that such an extension is simple for the following reasons:
> - **Geometric features are more than simple feature vectors**: We agree that simply adding feature vectors to each node would constitute a straightforward extension to WL. However, in contrast to adding generic feature vectors (which are essentially lists of scalars), geometric features must satisfy a well-defined transformation behaviour under spatial symmetries to retain their physical meaning. This introduces a notion of “geometric isomorphism” to the problem, which interplays with the “graph isomorphism” notion in the standard WL setting.
> - **Standard WL is not enough to study the increasingly relevant geometric graph setting**: The standard WL framework has [recently been applied](https://arxiv.org/abs/2201.07136) to study geometric GNNs which are purely based on distance cut-offs (e.g. SchNet like models). Yet, it would not be possible to perform this analysis for more recent equivariant GNNs such as MACE or PaiNN, because they contain feature types with symmetries, something not treated in standard WL. Our framework extends WL in a way that allows us to reason about the full range of geometric GNN models from invariant SchNet and DimeNet to equivariant MACE and PaiNN, and therefore constitutes a useful extension.
>
> We agree that the results in Section 3.1 (revised presentation) seem easy to derive from the definitions. We believe these are the **fruits of proposing a suitable formalisation** and **framework**, including the GWL test itself as well as the notion of $k$-hop distinct and $k$-hop identical geometric sub-graphs. We have tried to strike a balance between presenting theoretical results while remaining approachable and relevant to practitioners.
>
> These simple-to-grasp results in Section 3.1 can be insightful for understanding the workings and theoretical limitations of the two classes of geometric GNNs. We have expanded on this in Section 4 on understanding the geometric GNN design space via GWL, as well as synthetic experiments in Appendix E.

---

> > ### Author Response · Authors · 2022-11-13
> > **(Part 2/2) Point-by-point response to Reviewer 1**
> >
> > >R1.3. Also, the results in Section 4.2 on the correspondence between GNNs and (I)GWL are too straightforward.
> >
> > While we agree that the statements and proof techniques in Section 3.2 (revised presentation) largely follow Xu et al., Morris et al., we would like to reiterate that this generalisation is not trivial. This is due to the interplay between permutation and physical symmetries, as well as scalar and geometric attributes which come with additional constraints. Our main contribution is a simple yet precise definition of a geometric WL test. GWL then allows us to characterise the maximum expressive power of geometric GNNs, as stated in the results in Section 3.2.
> >
> > In that sense, the results in Section 3.2 are critical for **building the bridge** between GWL, an abstract mathematical tool, and practical geometric GNNs for real-world applications. We acknowledge that providing very similar statements for both GWL and IGWL is repetitive. We have revised the presentation to defer the IGWL – invariant GNN statements to the appendix.
> >
> > >R1.4. Theorem 21-23 build results on lower bounds of numbers of aggregators for injective functions. It would be great if it discusses whether or not such numbers are linked with the number of layers. In my mind, oversquashing might be interesting if its discussion can be generalized to geometric graphs. From a broader view, since this work is to generalize theoretical discussion on generic graphs to geometric ones, it would be better to develop a bigger story including richer contents, such as oversmoothing and oversquashing, rather than expressive power only.
> >
> > Thank you very much for encouraging us to better highlight the connection between the GWL framework (a theoretical tool) and its practical implications in building and training current geometric GNNs.
> >
> > Firstly, regarding the connection between injective functions and the number of layers: GWL assumes perfect injective aggregation as well as propogation of $\mathfrak{G}$-equivariant geometric information from local neighbourhoods. Thus, the test itself can be run for any number of iterations without any loss of information. This follows from the fact that the **composition of multiple injective functions** is an **injective function**. This is also the central premise of WL.
> >
> > On the other hand, in real-world geometric GNNs, $\mathfrak{G}$-equivariant information is propogated via summing features from multiple layers in **fixed dimensional spaces**, which may lead to distortion or loss of information from distant nodes, i.e. oversmoothing and oversquashing (as you have noted in your comment). Thank you very much for pointing us towards this direction!
> >
> > Towards this end, we have discussed in **Section 4** how GWL can be used to understand the design space of geometric GNNs. We highlight theoretical limitations of current geometric GNNs in terms of depth (new results proving invariant GNNs cannot compute global geometric properties) and body order of local aggregation (updated presentation). Additionally, in **Appendix C**, we include synthetic experiments supplementing our results from Section 4.
> >
> > Most notably, the synthetic experiments in Table 2 and Table 3 highlight the oversmoothing and oversquashing phenomenon in geometric GNNs with increased depth and reduced tensor order. It was interesting to notice that these issues are **most evident for E-GNN**, which uses a single vector feature to aggregate and propagate geometric information. Both theory and synthetic experiments point to how this is clearly not optimal. E-GNN is a [very](https://openreview.net/forum?id=GQjaI9mLet) [popular](https://proceedings.mlr.press/v162/stark22b.html) backbone for modelling macromolecules where long-range interactions often play important roles – our preliminary findings may be highly relevant to practitioners in this community. Studying these issues are exciting avenues for future work towards building provably powerful, practical geometric GNNs.
> >
> > >R1.5. I would like to recommend rejection. I am willing to raise my score if the above concerns are resolved.
> >
> > Thank you for your actionable feedback which helped significantly improve the manuscript. We believe we have addressed all the concerns raised. Please let us know if you are satisfied with our responses and have any further queries that we can address.

---

> ### Author Response · Authors · 2022-11-16
> **We are happy to discuss further**
>
> Dear reviewer, thank you very much once again for taking the time to review our manuscript and for your actionable feedback so far.
>
> We have worked hard to address the concerns raised by the original review in the revised manuscript, including better highlighting our novelty and improving our contributions, as well as drawing connections to practical challenges for geometric GNNs such as oversmoothing and oversquashing through synthetic experiments and code.
>
> As we are nearing the Reviewer-Author Discussions deadline (18 November), please let us know if you are satisfied with our responses and have any further queries that we can address. If we have addressed your concerns, please consider revising your score.

---

> ### Author Response · Authors · 2022-12-05
> **Discussion phase 2 ending soon; please let us know if your concerns have been addressed**
>
> Dear reviewer,
>
> Apologies for the repeated message, and thank you once again for your insightful review and actionable feedback.
>
> We have worked very hard to improve our manuscript and address the concerns raised by yourself and other reviewers during the revision period. As a quick summary, we addressed your main concerns by:
> - Better highlighting our novelty and improving our contributions.
> - Drawing connections to practical challenges for geometric GNNs such as oversmoothing and oversquashing through new results, synthetic experiments, as well as code.
>
> We would like to re-iterate that this is the **first work to study the expressive power of geometric GNNs** from the perspective of graph isomorphism. We have highlighted how this is non-trivial and fills a key research gap, as theoretical tools for standard GNNs are not applicable to the increasingly relevant special case of geometric graphs in drug discovery, biochemistry, material science, robotics, etc.
>
> As there is less than one week till the end of Discussions Phase 2 (**12 December**), please let us know if you are satisfied with our revised manuscript and have any further queries that we would be happy to address. If we have addressed your concerns, please consider revising your score.
>
> Soliciting actionable feedback from expert reviewers such as yourself to craft the best version of our paper is our top priority here.
>
> Best regards,
>
> Authors

---

> > ### Comment · Reviewer_Vdrv · 2022-12-05
> > **Reviewer Response**
> >
> > Dear authors,
> >
> > I appreciate the experiments in Table 2 and 3 to inspire any rigorous study of geometric graph networks in the future. So I would like to raise the score from 3 to 5.
> >
> > But in my opinion theorems and proof in Section 3 are still incremental tech contributions to this domain.

---

> > > ### Author Response · Authors · 2022-12-06
> > > **Post-rebuttal response to Reviewer 1**
> > >
> > > > I appreciate the experiments in Table 2 and 3 to inspire any rigorous study of geometric graph networks in the future. So I would like to raise the score from 3 to 5.
> > >
> > > Thank you very much for your response and for upgrading your score. We are excited to hear that you liked the synthetic experiments and felt that the directions they identify can lead to follow up work in the future!
> > >
> > > > But in my opinion theorems and proof in Section 3 are still incremental tech contributions to this domain.
> > >
> > > We acknowledge your opinion - thank you for the honest feedback and for your time.
> > >
> > > As a final request, we would appreciate if you could point us to any key references that you believe this work builds upon or extends known results from. This would help us better understand how our contributions are incremental and address this matter as best as possible.
> > >
> > > For completeness and for aiding the other reviewers in making their assessment, we respectfully disagree that our framework and results in Section 3 are incremental contributions to the domain of geometric graph neural networks:
> > > 1. We proposed to study geometric GNN expressivity through the lens of graph isomorphism for geometric graphs. To the best of our knowledge, this is **the first definition of geometric graph isomorphism**, at least in the context of machine learning.
> > > 2. We introduce the Geometric WL framework, a theoretical tool and upper bound on the expressive power of geometric GNNs. GWL is **novel** and **fills a key research gap**, as the standard theoretical toolkit for normal non-geometric GNNs is not applicable to the special case of geometric graphs.
> > > 3. In Section 3.1 and 3.2, we precisely characterise the classes of geometric graphs that can and cannot be distinguished by GWL and the corresponding class of geometric GNNs. These results are **new** and **practically relevant**, as they formalise our understanding of geometric GNNs, which are now the go-to for modelling systems in biochemistry, material science, robotics, etc. We provide practical implications of these results in Section 4.

---

> > > > ### Comment · Reviewer_Vdrv · 2022-12-06
> > > > **Response about contributions of theorems and their proof**
> > > >
> > > > Dear authors,
> > > >
> > > > Thank you for your prompt response! I would try to support my claim as following:
> > > >
> > > > 1. Proofs of Prop 1-4 are simply following two sentences: 1) $k$-step GWL is a perfect I-hash (i.e., ''identifies'') of k-hop subgraphs,  and 2) $1$-step IGWL is a perfect I-hash of $1$-hop subgraph. These two sentences are enough to prove these four propositions, but they selves are somehow not perfectly proven (if the authors think they are necessary to prove). However, these two sentences are directly used to prove the propositions, and the remaining is only copy of definitions.
> > > >
> > > > 2. Proof of Prop 7 is simply merging the two arguments in the above point by setting $k=1$.
> > > >
> > > > 3. Proof of Theorem 8 follows proof of Lemma 2 of Xu et al. They share a sentence-to-sentence level of bijection. The only difference is including $v, g$, whose impact is not significant for the proof.
> > > >
> > > > 4. Proof of Theorem 9 follows proof of Theorem 3 of Xu et al. They are very similar with the only difference again from definitions of GWL vs WL.
> > > >
> > > > 5. Results of Appendix G.1(Theorem 16-19) are following Chen et al., remarked by the authors as ''with minor adaptations''.
> > > >
> > > > ---
> > > >
> > > > Reference:
> > > >
> > > > Xu et al. How powerful are graph neural networks? ICLR 2019.
> > > >
> > > > Chen et al.  On the equivalence between graph isomorphism testing and function approximation with gnns. NeurIPS, 2019.
> > > >
> > > > ---
> > > >
> > > > Minor issue:
> > > >
> > > > in proof of Prop 1-4, it might be ''for all graph isomorphism $b$'' instead of ''there exists some bijection $b$''?

---

> > > > > ### Author Response · Authors · 2022-12-08
> > > > > **Response to Reviewer 1 on the significance of theorems and proofs**
> > > > >
> > > > > Thank you for supporting your claim and for taking the time to review our paper, as always.
> > > > >
> > > > > Overall, it is our opinion that it is highly desirable that a mathematical formalism makes it simple and easy to derive results, rather than a weakness. Additionally, we have stated clearly whenever we have followed proof techniques from seminal papers. We have also highlighted why each of our results are important for the overall message of the paper as well as practically relevant. Let us elaborate.
> > > > >
> > > > > > - Proofs of Prop 1-4 are simply following two sentences: 1) k-step GWL is a perfect I-hash (i.e., ''identifies'') of k-hop subgraphs, and 2) 1-step IGWL is a perfect I-hash of 1-hop subgraph. These two sentences are enough to prove these four propositions, but they selves are somehow not perfectly proven (if the authors think they are necessary to prove). However, these two sentences are directly used to prove the propositions, and the remaining is only copy of definitions.
> > > > > > - Proof of Prop 7 is simply merging the two arguments in the above point by setting k=1.
> > > > >
> > > > > This may be subjective, but we feel that our theoretical results being easy to understand and simple to derive is a **virtue**. As previously stated, we believe these are the fruits of proposing a suitable formalisation and framework from which results can be derived easily. In the revised manuscript and point-by-point responses, we have highlighted how these simple results have important practical implications in the design of geometric GNNs, and have additionally included synthetic experiments to further reinforce this.
> > > > >
> > > > > Note that the formalisms cited by the reviewer follow from the axiomatic properties of GWL/IGWL within our framework:
> > > > >
> > > > > > 1-step IGWL is a perfect I-hash of 1-hop subgraph.
> > > > >
> > > > > This follows from **Property 1**, stating that I-Hash is a G-orbit injective function that will assign unique colours to unique 1-hop subgraphs, up to global symmetries.
> > > > >
> > > > > > k-step GWL is a perfect I-hash (i.e., ''identifies'') of k-hop subgraphs
> > > > >
> > > > > This follows from Property 1 as well as **Property 2**, which states that the aggregation in each iteration of GWL is injective. Thus, k steps of GWL will injectively aggregate information from k-hop subgraphs, as the **composition of injective functions is an injective function**.
> > > > >
> > > > > Another alternative way to concisely understand this is via **geometric computation trees** (Figure 3). GWL colours geometric computation trees. We have elaborated on how computation trees are constructed in Section 4.
> > > > >
> > > > > > - Proof of Theorem 8 follows proof of Lemma 2 of Xu et al. They share a sentence-to-sentence level of bijection. The only difference is including , whose impact is not significant for the proof.
> > > > > > - Proof of Theorem 9 follows proof of Theorem 3 of Xu et al. They are very similar with the only difference again from definitions of GWL vs WL.
> > > > > > - Results of Appendix G.1(Theorem 16-19) are following Chen et al., remarked by the authors as ''with minor adaptations''.
> > > > >
> > > > > We have stated clearly in all these cases that we follow the proof techniques from the seminal papers of Xu et al., Morris et al., Chen et al. (we also reference these papers several times). E.g. pg.8, pg.19, pg.22, specifically before providing all the proofs. Our work naturally builds upon these seminal papers -- our goal was to generalise the study of GNN expressivity to geometric graphs.
> > > > >
> > > > > We have also tried to highlight why each of these results are important, much needed for the flow of the paper, as well as practically relevant. Without these results, the paper would be incomplete/disconnected. To briefly recap, these results enable us to build a bridge between the proposed GWL framework and geometric GNNs in order to study the expressive power of geometric GNNs.
> > > > >
> > > > > Finally, as you correctly noted, all of these proofs build upon our definition of the Geometric WL test. Proposing a suitable generalisation of WL which accounts for physical symmetries in geometric graphs is **our main technical contribution** and **novelty**. To the best of our knowledge, ours is the **first work** to study the expressive power of geometric GNNs, and we hope that our work points the community in a **useful direction** towards theoretically grounded geometric GNNs.

---

### Author Response · Authors · 2022-11-13
**Overall response**

# Overall response

We would like to express our sincere gratitude to the reviewers for taking the time to review this manuscript, and for their actionable suggestions which have significantly improved our paper.

We are glad to hear that the reviewers found our motivation to study the expressive power of Graph Neural Networks for geometric graphs to be **interesting**, **important**, and **significant** (R1, R2, R3). Thank you for acknowledging the **simplicity** of the proposed Geometric Weisfeiler-Leman framework (R1, R3) and the **interestingness** of its connections with universal approximation (R1, R3). Thank you for highlighting the **clarity of the presentation** (R1, R2, R3) and **contextualisation w.r.t. existing literature** (R2, R3). We are happy to know that R3 believes our framework will **serve to inform future work** in this growing area.

For your convenience, we would like to highlight the main novelty and contributions of our work:
1. We propose the **Geometric Weisfeiler-Leman test** (GWL), a novel theoretical tool to characterise the expressive power of geometric GNNs from the perspective of graph isomorphism. We believe our work fills a key research gap as geometric graphs come with both relational structure and geometric attributes, making theoretical tools for standard GNNs inapplicable in this increasingly relevant real-world setting.
2. We use GWL to provide novel insights into the workings and theoretical limitations of geometric GNNs. Most notably, we formalise the **advantages of equivariant GNNs** over **invariant GNNs**: invariant models have limited expressive power as they only reason locally via scalar quantities, while equivariant models can distinguish a larger class of graphs by propagating geometric vector quantities beyond local neighbourhoods.
3. We connect geometric GNNs’ ability to **discriminate geometric graphs** (as in GWL) and their **universal approximation** properties studied in past work. This is significant because discrimination is a more granular and practically insightful lens into geometric GNNs than universality. A model may either be universal or not. On the other hand, we show how there can be multiple degrees of discrimination.

---

# Improvements based on our reviewer's comments

## Summary

Below we summarise the major changes from the initial version of the paper that address the reviewers’ concerns:
1. In **Section 4**, we elaborate on how the GWL framework can be used to understand the design space of geometric GNNs. We highlight theoretical limitations of current geometric GNNs in terms of depth (new results proving invariant GNNs cannot compute global geometric properties) and body order of local aggregation (updated presentation), as well as their practical implications. This addresses R1.2, R1.3, R2.3, R2.7, R3.1, R4.4.
2. In **Appendix C**, we supplement our theory from Section 4 with new synthetic experiments to highlight the practical challenges in building powerful geometric GNNs: oversmoothing and oversquashing with increased depth, as well as designing efficient higher order aggregators. This addresses R1.4, R2.5, R3.1, R4.1. Our codebase, titled `geometric-gnn-dojo`, implements several popular geometric GNN architectures in a unified and pedagogical manner, and is available as supplementary material.
3. Additionally, in **Appendix B**, we provide more detailed background on the two classes of geometric GNNs. We include unified equations for invariant GNNs (SchNet and DimeNet), equivariant GNNs using cartesian vectors (PaiNN), and equivariant GNNs using spherical tensors (TFN). This addresses R2.2, R4.1, R4.3.

For clarity, we use two highlighting mechanisms in the revised manuscript to present updates:
- Blue font:, Indicates new ideas, new content, major changes.
- Blue box: Indicates re-writing/clarity changes or more minor presentation-related changes. The box label “Addressed R2.3” indicates that these changes mean to address reviewer 2’s request number 3.

## Point-by-point responses

In addition to the revised manuscript, we would also like to provide point-by-point responses to each reviewer, especially for aspects that are not included in the above-mentioned summary of changes.

---

> ### Author Response · Authors · 2022-11-16
> **Quick update on code for synthetic experiments**
>
> Our codebase, titled `geometric-gnn-dojo`, is now available as supplementary material. We hope to publicly release it on GitHub in the future. As a reminder, the key outcomes included identifying the challenges of oversmoothing and oversquashing with increased depth, as well as that of designing efficient higher order aggregators.
>
> We have implemented several popular geometric GNN architectures in a unified and pedagogical manner:
> 1. $\frak{G}$-equivariant GNNs using cartesian vectors: E-GNN, GVP-GNN
> 2. $\frak{G}$-equivariant GNNs using spherical tensors: TFN, MACE
> 3. $\frak{G}$-invariant GNNs: SchNet, DimeNet
>
> The codebase also provides jupyter notebooks to perform all the synthetic experiments proposed in the revised manuscript's Appendix C:
> 1. The counterexamples from Pozdnyakov et al. which test a layer's ability to identify **neighbourhood fingerprints**.
> 2. The rotationally symmetric structures which test a layer's ability to identify **neighbourhood orientation**.
> 3. The $k$-chains which test a model's ability to **propagate geometric information**.
>
> Our long-term goal for  `geometric-gnn-dojo` is to provide a pedagogical resource for beginners and experts to explore the geometric GNN design space in a principled and theoretically grounded manner. We believe this is important as most recent architectural advances in this area have been driven by domain-specific datasets -- `geometric-gnn-dojo` fills a gap by allowing practitioners to stress test ideas on general-purpose geometric graphs.

---

> ### Comment · Reviewer_Vdrv · 2022-12-11
> **Final score**
>
> Thanks to reviewers for these responses. I appreciate the motivation for investigating Geometric GNNs, and stay negative about the extent of tech contributions.
>
> I will keep my score as 5.

---

> > ### Author Response · Authors · 2022-12-11
> > **Thank you for your consideration**
> >
> > Thank you very much for your consideration, and for taking the time to review, once again. Our reminders today were meant for the other reviewers who had not responded yet. Thank you for creating a response, nonetheless!
> >
> > Regarding your remaining concern, we have tried our best to address them, but we feel we are unable to convince you without writing a completely new paper altogether.
> >
> > This is the first work to formalise the experssive power of GNNs for geometic graphs. In doing so, we have built upon seminal work on connecting graph isomorphism to normal GNNs and generalised results from Xu et al, Morris et al, Chen et al. to the geometric graph setting, which is new and leads to interesting practical insights into geometic GNNs. Our framework allows us to formalise in a simple and easy-to-derive manner the classes of geometric graphs that can or cannot be distinguished by invariant and equivariant GNNs. We stand by our paper and would not like to change both of these aspects (connection to seminal work and simplicity of presentation) in order to not deviate from our vision for the paper.

---

### Author Response · Authors · 2022-11-18
**Less than 12 hours to go till discussion closes; please let us know if you have any remaining concerns**

Dear reviewers,

Thank you very much once again for taking the time to review our manuscript. Your actionable feedback has significantly improved our paper.

We have worked hard to address the concerns raised by all the reviewers in the revised manuscript. We have summarised our revision in the comment titled 'Overall response' as well as via the detailed point-by-point response to each reviewer.

As we are very close to the Reviewer-Author Discussions deadline (**18 November** -- **less than 12 hours to go**), please let us know if you are satisfied with our responses, have any further queries that we can address, or any feedback on the proposed revisions.

If we have addressed your concerns, please consider revising your score.

Best regards,

Authors

---

### Decision · Program_Chairs · 2023-01-20

**Decision:**

Reject

**Justification For Why Not Higher Score:**

N/A

**Justification For Why Not Lower Score:**

N/A

**Metareview: Summary, Strengths And Weaknesses:**

Thank you for submitting you work to ICLR 2023. This paper was discussed extensively both by reviewers and authors that were very responsive and provided both detailed rebuttal, discussion and a paper revision. All reviewers agreed the goal of developing GNN algorithms and expressiveness theory for geometric graphs is a worthy one. The main concerns about this paper are two fold: (a) is the development of geometric-WL sufficiently motivated and important step towards for this goal? and (b) are its technical contributions significant enough for a theoretical paper?
The discussion among reviewer converged to one positive reviewer (with a score of 8) and three negative reviewers (with scores 3,5,5). All the reviewers (including the positive one) agreed the answer to (b) is that the mathematical proofs are overall incremental on their own. The reviewers were split on (a), however 3/4 of the reviewers were not convinced the answer is positive. Overall, it is hard to recommend acceptance in this state of things, however we do recommend the authors to better establish the motivation and benefit of their method over simple alternatives.